# Unraveling the functional role of DNA demethylation at specific promoters by targeted steric blockage of DNA methyltransferase with CRISPR/dCas9

Daniel M. Sapozhnikov ⬦ [1] & Moshe Szyf ⬦ [1✉]

Despite four decades of research to support the association between DNA methylation and gene expression, the causality of this relationship remains unresolved. Here, we reaffirm that experimental confounds preclude resolution of this question with existing strategies, including recently developed CRISPR/dCas9 and TET-based epigenetic editors. Instead, we demonstrate a highly effective method using only nuclease-dead Cas9 and guide RNA to physically block DNA methylation at specific targets in the absence of a confounding flexibly-tethered enzyme, thereby enabling the examination of the role of DNA demethylation per se in living cells, with no evidence of off-target activity. Using this method, we probe a small number of inducible promoters and find the effect of DNA demethylation to be small, while demethylation of CpG-rich *FMR1* produces larger changes in gene expression. This method could be used to reveal the extent and nature of the contribution of DNA methylation to gene regulation.

[1] Department of Pharmacology and Therapeutics, Faculty of Medicine, McGill University, Montreal, QC, Canada.  ✉email: moshe.szyf@mcgill.ca

DNA methylation is broadly involved in transcriptional regulation across a vast number of physiological and pathological conditions[1]. For nearly half a century, it has been widely documented that the presence of methyl groups on the fifth carbon of cytosines in the context of CpG dinucleotides within promoters is associated with transcriptional repression[2]. This is considered to be a crucial epigenetic mark as deviations from the tightly-regulated and tissue-specific developmental patterns have been implicated in conditions as diverse as cancers[3], suicidal behavior[4], and autoimmune diseases[5]. Yet, these studies also exemplify a fundamental challenge in the field: the persistent inability to attribute causality to a particular instance of aberrant DNA methylation. The issue of whether DNA demethylation is the driver of relevant transcriptional changes continues to be a source of controversy and is magnified by multiple studies suggesting that changes in gene expression and transcription factor binding can in some cases precede DNA demethylation[6–11]. The answers to this set of questions would reveal whether a particular DNA methylation state is only a marker for a particular condition, or whether it is plays a critical role in the pathophysiological mechanism.

In the case of DNA methylation, unconfounded manipulation of the methylation state of a CpG or region of CpGs in isolation remains a challenge: genetic (DNA methyltransferase knockdown) and pharmacological (5-aza-2′-deoxycytidine and S-Adenosyl methionine) hypo-methylating or hyper-methylating agents cause genome-wide changes in methylation[12–16], confounding conclusions by countless concurrent changes throughout the genome in addition to any region under study. A more specific approach to assessing causality involves comparing the abilities of in vitro methylated and unmethylated regulatory sequences to drive reporter gene expression in transient transfection assays. However, this is an artificial system and a simplification of the complex chromatin architecture at the endogenous locus and, therefore, the effects of methylation in the context of an artificial promoter-reporter plasmid may misrepresent those that would occur under physiological conditions.

More recently, the TET dioxygenases—which oxidize the methyl moiety in cytosine and can lead to passive loss of methylation by either inhibiting methylation during replication, or through repair of the oxidized methylcytosine and its replacement with an unmethylated cytosine—were targeted to specific sites using a fusion of TET dioxygenase domains to catalytically inactive CRISPR/Cas9 (dCas9)[17–20]. However, this method still introduces several confounding factors that preclude causational inferences, such as the fact that oxidized methylcytosines are new epigenetic modifications that are not unmethylated cytosines[21–27], and the fact that TET has methylation-independent transcriptional activation activity[28,29].

We propose and optimize instead an enzyme-free CRISPR/dCas9-based system for targeted methylation editing, which we show is able to achieve selective methylation in vitro and passive demethylation in cells through steric interference with DNA methyltransferase activity. We map the size of the region of interference, optimize the system for nearly complete demethylation of targeted CpGs without detectable off-target effects, and analyze the transcriptional consequences of demethylation of genetically dissimilar regions across several human and mouse genes. In doing so, we provide evidence that DNA demethylation at proximal promoters increases gene expression in some instances but not others, that it does so to varying degrees depending on the genomic context, and that demethylation may facilitate responses to other factors. Most importantly, we report a simple tool for investigations into the effects of DNA demethylation that can be applied with ease and in multiplexed formats to examine the vast existing and forthcoming correlational literature in order to distinguish causational instances of DNA methylation

and begin to develop a fundamental understanding of this biological phenomenon on a foundation of causality.

## Results

### CRISPR/TET-based approaches confound the causal relationship of DNA methylation and transcription.
To develop a tool for site-specific DNA methylation editing, we elected to study the murine interleukin-33 (Il33) gene. The distance between individual CpGs and sets of CpGs within its canonical CpG-poor promoter provides a simple starting point that enables specific CpG targeting in order to evaluate the impact of discrete methylation events on gene transcription (Fig. 1a). The promoter is highly methylated in NIH-3T3 cells (Supplementary Fig. 1A) and upon treatment of cells with the demethylating agent 5-aza-2′-deoxycytidine, CpGs adjacent to the transcription start site (TSS) are demethylated (Fig. 1b) and gene expression is moderately induced (Fig. 1c). However, this classical response to 5-aza-2′-deoxycytidine also emphasizes the shortcomings of this common approach in DNA methylation research: (1) multiple CpGs in the promoter are demethylated, so it remains unclear which sites of methylation contribute to transcriptional inhibition and (2) the global genomic consequences of 5-aza-2′-deoxycytidine treatment result in the induction of expression of several putative and experimentally validated Il33 transcription factors (Fig. 1d), exemplifying the possibility that demethylation of the Il33 promoter may not be the event responsible for upregulation of the gene. This demonstrates a need for an accurate and specific targeted methylation editing technology that can properly interrogate the fundamental question of the causal relationship between DNA methylation at specific sites and gene expression in cis.

To first assess the efficacy and specificity of the available targeted DNA methylation editing technology, we examined the lentiviral system created by Liu et al.[30] consisting of a catalytically inactive Cas9 (dCas9) fused to the catalytic domain of TET1 (dCas9-TET or a catalytic mutant, dCas9-deadTET), which is thought to promote active DNA demethylation by oxidation of the methyl moiety and eventual replacement of the modified cytosine with unmethylated cytosine by the base excision repair pathway[17]. We developed a set of 20 base-pair (bp) CRISPR guide RNAs (gRNAs) targeting distinct regions in the promoter of the Il33-002 transcript, the inducible variant[31,32] (Fig. 1a and Supplementary Table 1). The system was effective in partially demethylating the Il33 promoter; however, we noted several shortcomings of this method.

First, it was immediately apparent that even in the absence of targeting, NIH-3T3 cells expressing only a scrambled, non-targeting guide (gRNAscr) and dCas9-TET were significantly more demethylated than those expressing the same gRNAscr and dCas9-deadTET (Fig. 1e–g). While dCas9-TET triggered a 22–26% demethylation as compared with dCas9-deadTET at CpGs 5 ($P < 0.0001$), 10 ($P < 0.0001$), and 11 ($P < 0.0001$), dCas9-TET:gRNAscr that was not targeted to these sites also caused demethylation at these sites as well as all remaining evaluated CpGs. This is indicative of a potential ubiquitous and dCas9-independent activity of the fused, over-expressed TET domain, that we provide further evidence for with whole-genome methylation analysis in a subsequent section.

Second, the demethylation caused by dCas9-TET spanned a substantial genetic distance. For example, in gRNA1:dCas9-TET cells, while the protein complex was positioned at and significantly demethylated CpGs 1, 2, and 3 ($P < 0.0001$), the remaining CpGs were all significantly demethylated as well, including CpG 11 ($P = 0.00014$), which is nearly 700 bp away from gRNA1 (Fig. 1e). Similar significant long-distance demethylation effects could be observed in cells expressing gRNA2 and gRNA3 (Fig. 1f, g). The potential for long-distance effects is

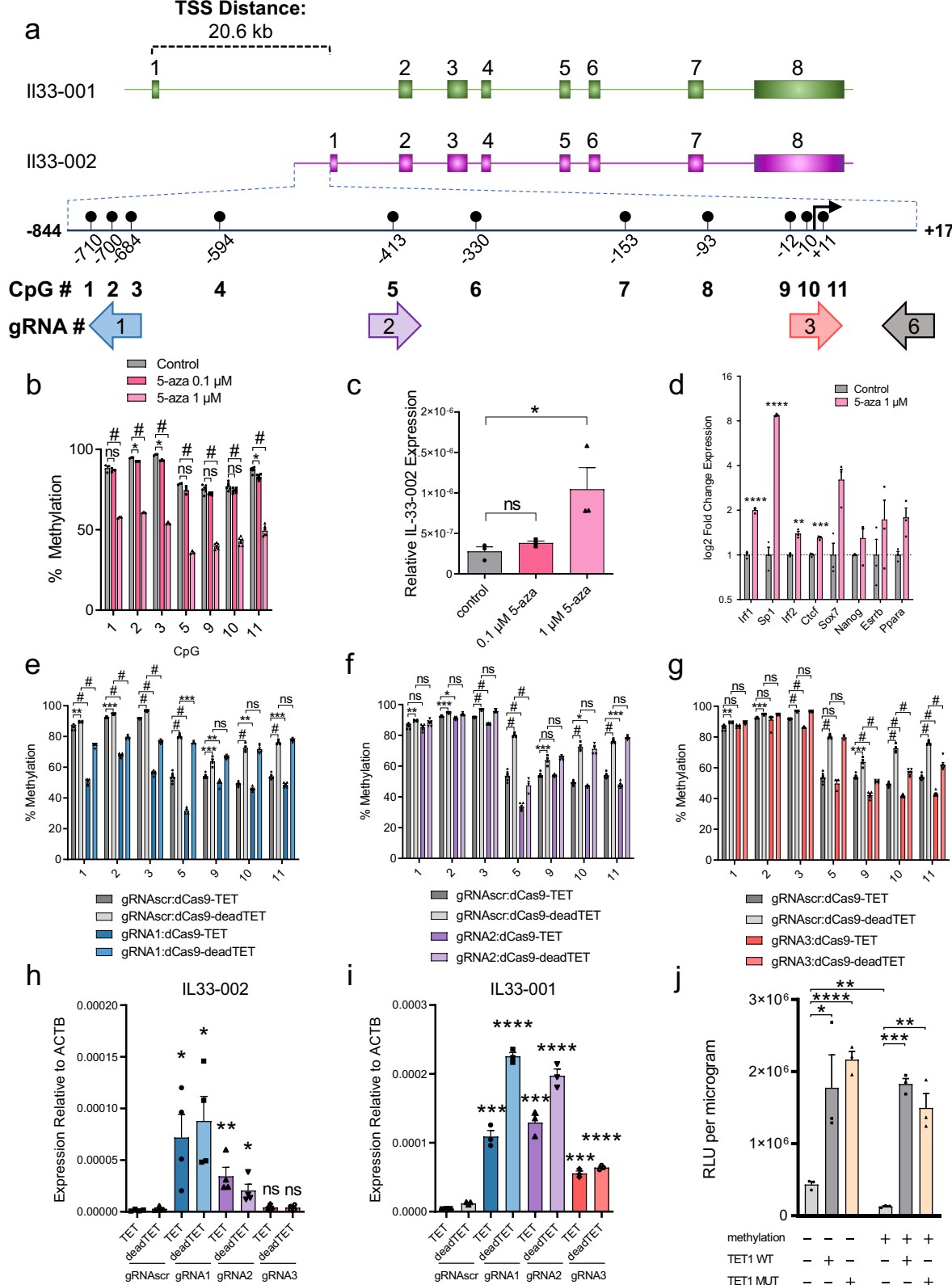

further exemplified at the mRNA level in the strong transactivation effects of dCas9-TET positioned at the *Il33-002* promoter on the distant *Il33-001* promoter, approximately 21 kb away (Fig. 1l).

Third, when evaluating the transcriptional effects of the epigenetic editing system, we were surprised to discover that dCas9-deadTET paired with gRNA 1 or 2 (gRNA3 blocks the TSS and likely interferes with RNA polymerase binding[33]) resulted in strong

demethylation and transactivation of the *Il33-002* transcript to levels comparable to dCas9-TET (Fig. 1h), despite lacking any catalytic capacity to initiate the active DNA demethylation process. To ensure that this unexpected result was not a consequence of erroneous sample switches, we amplified the region containing the catalytic mutations of the TET1 domain in the DNA samples used for methylation analysis, and in the cDNA samples used for expression

**Fig. 1 Targeting the Il33 promoter with dCas9-TET. a** Schematic of the murine *Il33* genomic locus depicting the two transcriptional isoforms with a highlighted 800 bp region of the *Il33-002* promoter and the locations of the 11 CpGs as well as four gRNAs targeting specific CpGs. The 11 CpGs are numbered sequentially in the 5′ to 3′ direction. The promoter-targeting gRNAs used in these experiments are shown relative to the CpGs and are approximately to scale such that CpGs 1, 2, and 3 are targeted by gRNA1, CpG 5 by gRNA 2, and gRNA 3 targets CpGs 9, 10, and 11—which overlap the transcription start site (TSS), marked by a black arrow. The orientation of the gRNAs is indicated by the direction of the arrow labeled with the respective gRNA, where an arrow pointing to the left indicates a gRNA that binds the plus strand. The fragment cloned into the luciferase vector (pCpGl) is marked at either end, spanning from −844 to +171 relative to the TSS. **b** Percent of DNA methylation (mean ± SEM) assayed by bisulfite-pyrosequencing at the three TSS CpGs (labeled 9–11) following treatment of NIH-3T3 cells with indicated concentrations of 5-aza-2′-deoxycytidine (5-aza) or water control ($n = 3$ independent experiments for CpGs 1, 2, 3, and 5; $n = 6$ for CpGs 9, 10, and 11). **c** Expression of *Il33-002* (mean ± SEM) quantified by RT-qPCR and normalized to beta actin (*Actb*) expression following treatment of NIH-3T3 cells with indicated concentrations of 5-aza-2′-deoxycytidine (5-aza) or water control ($n = 3$ biologically independent samples) (Student's *t*-test, two sided, control vs. 0.1 μM 5-aza; $P = 0.1636$, control vs. 1 μM 5-aza; $P = 0.0482$). **d** Expression (mean ± SEM) of predicted (Transfac) and experimentally validated (Qiagen, ENCODE, Gene Transcription Regulation Database) *Il33-002* transcription factors quantified by RT-qPCR and normalized to *Actb* expression following treatment of NIH-3T3 cells with indicated concentrations of 5-aza-2′-deoxycytidine (5-aza) or water control ($n = 3$ biologically independent samples). **e–g** Percent of DNA methylation (mean ± SEM) assayed by bisulfite-pyrosequencing at seven targeted CpGs in the *Il33-002* promoter following transduction with lentiviruses and antibiotic selection of virally infected cells (gRNAs) or selection by flow cytometry (BFP; dCas9 constructs) of NIH-3T3 cells with dCas9-Tet/dCas9-deadTET (BFP) and gRNA1 (**e**), gRNA2 (**f**), or gRNA3 (**g**) compared to gRNAscr (light and dark gray, gRNAscr data identical in **e–g** and shown for comparison) ($n = 4$–8 biologically independent experiments, depending on specific condition and CpG; see Source Data file for specific n of interest). **h, i** Expression of *Il33-002* (**h**) and *Il33-001* (**i**) (mean ± SEM) quantified by RT-qPCR and normalized to *Actb* expression in NIH-3T3 stably expressing one of 4 gRNAs and dCas9-TET or dCas9-deadTET ($n = 3$–4 biologically independent samples; statistical comparisons are between each gRNA and gRNAscr bearing the same dCas9 construct (dCas9-TET or dCas9-deadTET)). All data shown as (mean ± SEM). **j** Relative light units normalized to protein quantity (mean ± SEM) in transfected HEK293 cells. Cells were transiently transfected with methylated or unmethylated SV40-luciferase vector along with mammalian wild-type or mutant human TET1 expression plasmid or empty vector (pEF1A) control ($n = 3$). * indicates statistically significant difference of $P < 0.05$, **$P < 0.01$, ***$P < 0.001$, **** or # of $P < 0.0001$, and ns not significant (Student's *t*-test, two-sided, with Holm-Sidak correction if number of tests is greater than 3). Source data are provided as a Source Data file.

quantification and confirmed by Sanger sequencing that all dCas9-deadTET samples bore the two point mutations that render it catalytically inactive (Supplementary Fig. 1B). Equally surprising was the fact that that dCas9-deadTET was also effective in transactivation of *Il33-001* (Fig. 1i) ($P < 0.0001$ for all targeting gRNAs). The *Il33-001* transcript was also significantly more expressed in dCas9-deadTET cells under gRNA1 ($P = 0.0091$) and gRNA3 ($P = 0.0033$) as compared to catalytically active dCas9-TET, though it may be caused by different level of expression of the constructs; dCas9-deadTET expression levels were moderately higher than dCas9-TET by RT-qPCR (Supplementary Fig. 1C), whereas the protein levels as determined by a western blot analysis were not significantly different (Supplementary Fig. 1D–E).

To assess by a secondary measure the DNA methylation independent transactivation capacity of TET proteins, we performed transient co-transfections of in vitro methylated or unmethylated promoter-reporter plasmids—luciferase driven by the SV40 promoter/enhancer—in combination with a mammalian expression vector expressing human TET1 (TET1 WT), mutant TET1 (TET1 MUT), or an empty vector control (pEF1A). We found that TET1 induces the activity of completely unmethylated promoters (Fig. 1j), as does TET2 (Supplementary Fig. 1M), reaffirming the notion that TET proteins produce transcriptional changes independently of any DNA demethylation and thus confound correlational assessments. SV40-pCpGl copy number in cells is equivalent upon transfection of fully methylated or fully unmethylated DNA (Supplementary Fig. 1N).

Additionally, we combined our three targeting gRNAs with the well-characterized dCas9-VP64 fusion; VP64 is a potent transcriptional activator originating from the herpes simplex virus[34]. The tetramer of the herpes simplex VP16 protein acts to activate transcription primarily through recruitment of basal transcription machinery, including TFIID/TFIIB, and has no known catalytic capacity for DNA demethylation[35]. Yet, we found that dCas9-VP64 co-expressed with all three *Il33-002* gRNAs resulted in dramatic and broad demethylation of the *Il33-002* promoter in stably infected cells (Supplementary Fig. 1F–H). This suggests that DNA demethylation can in particular instances be secondary to transcription factor recruitment and transcriptional activation, rather than causal (Supplementary Fig. 1I). To further test this,

we performed a time-course experiment in which we observed activation of transcription of *Il33-002* by dCas9-VP64:gRNA2 24 h after transient transfection and prior to initiation of any detectable demethylation at this time point nor at any time point up to 96 h (Supplementary Fig. 1J). We again found significant and robust activation of the distant *Il33-001* promoter (gRNA2, $P < 0.05$; gRNA3, $P < 0.001$), supporting the notion that enzymatic domains flexibly tethered to dCas9 can act across large genetic distances (Supplementary Fig. 1K).

Finally, we detected a significant increase of 5-hydroxymethylcytosine in the *Il33-002* promoter in the presence of dCas9-TET but not dCas9-deadTET (Supplementary Fig. 1L), demonstrating that demethylation is not the only epigenetic change conferred by dCas9-TET and, since dCas9-deadTET activates transcription (Supplementary Fig. 1C), that catalytic 5-hydroxymethylation is not necessary for the transcriptional induction.

**A method for site-specific DNA methylation in vitro.** A potential mechanism for producing specific demethylation in cells is through targeted physical interference with the DNA methyltransferase (DNMT) machinery that deposits methyl groups onto nascent post-replicative DNA. We reasoned that since dCas9 is able to interfere with transcriptional machinery to reduce gene expression[33], it may also be able to sterically obstruct DNMT activity at its binding position (Fig. 2b). dCas9 is a prokaryotic protein with no documented protein:protein interaction with eukaryotic gene transcription machinery, the protein has no homology to known eukaryotic protein:protein interaction domains and has no enzymatic activity epigenetic or other[36].

To test this hypothesis, we first investigated whether dCas9 could be applied as a tool to interfere with DNMT activity at targeted CpGs in a simplified in vitro system. The target DNA used for methylation was a 1015 bp fragment of the *Il33-002* promoter (Fig. 1a) inserted into an otherwise CpG-free luciferase reporter vector (pCpGl)[37] to enable the assessment of methylation changes on reporter gene activity in transient transfection assays. Standard methylation with recombinant bacterial CpG methyltransferase M.SssI protein resulted in 80–93% methylation at all CpGs as assayed by pyrosequencing (Fig. 2a) and a

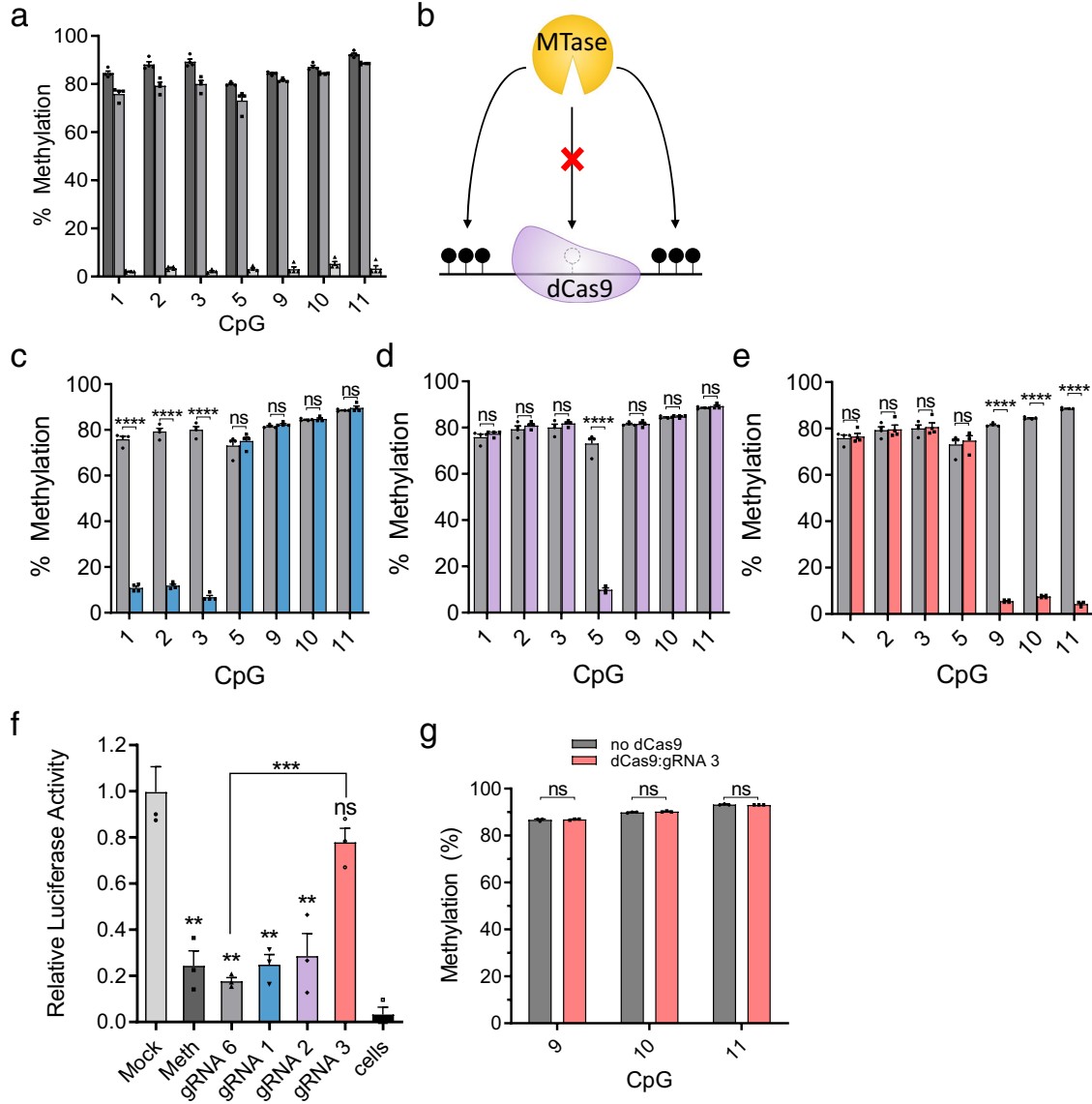

**Fig. 2 dCas9 blocks DNA methyltransferase in vitro. a** Pyrosequencing data (mean ± SEM, $n = 4$ biologically independent samples) for the methylation state of indicated CpGs in the *Il33*-pCpGl plasmid following standard methylation for 4 h by M.SssI (dark gray), methylation in the presence of dCas9 and gRNA 6 (distant binding) (gray), or a mock-methylated control reaction that lacked S-adenosyl methionine substrate (light gray). **b** Diagram illustrating the principle of site-specific methylation utilizing pre-incubation of DNA with dCas9 and selective CpG-targeting guide restricting M.SssI from binding and methylating the targeted region, while permitting methylation of remaining unobstructed CpGs. **c–e** Pyrosequencing data ($n = 4$ biologically independent samples, mean ± SEM)) for the methylation state of CpGs in the IL-33-pCpGl plasmid following pre-incubation with dCas9 and gRNA1 (**c**), gRNA2 (**d**), or gRNA3 (**e**) and methylation by M.SssI (colored bars). Gray bars are identical (**a**, **c–e**) and indicate methylation levels for the same treatment utilizing gRNA6. **f** Luciferase reporter activity of the plasmids (**a**, **c–e**), expressed as relative light units (mean ± SEM) normalized for protein content per sample, and then normalized to average value for mock methylated condition ($n = 3$ biologically independent experiments). All statistical comparisons are to mock methylated conditions unless otherwise indicated. **g** Percent of methylation (mean ± SEM) assayed by pyrosequencing when Il33-pCpGl is incubated with dCas9 and gRNA 3 or only gRNA 3 (no dCas9 control) after standard methylation, instead of before ($n = 3$ biologically independent samples). * indicates statistically significant difference of $P < 0.05$, **$P < 0.01$, ***$P < 0.001$, ****$P < 0.0001$, and ns not significant (Student's $t$-test, two-sided, with Holm-Sidak correction if number of tests is greater than 3). Source data are provided as a Source Data file.

significant 4-fold decrease in luciferase reporter activity in a transient transfection assay ($P = 0.0041$) (Fig. 2f). Incubation of *Il33*-pCpGl with recombinant dCas9 protein and an in vitro transcribed chimeric control gRNA (gRNA6 in Fig. 1a) targeting the CpG deficient region approximately 110–130 bp downstream of the TSS only slightly inhibited the efficiency of the M.SssI reaction at all CpGs (Fig. 2a in gray). The plasmid was still highly methylated and the treatment also significantly reduced luciferase activity ($P = 0.0018$) compared to mock treatment and to a similar extent as standard methylation ($P = 0.374$) (Fig. 2f). This

confirms that the reaction components (including dCas9 protein, non-CpG-targeting gRNA, buffer system, and incubation times) do not compromise DNA methyltransferase activity of M.SssI in vitro.

The DNA was then incubated with recombinant dCas9 protein and each of the three in vitro transcribed gRNAs—targeting CpGs in the proximal promoter region of *Il33-002*—in order to facilitate binding of the dCas9:gRNA complex to the DNA prior to the addition of M.SssI methyltransferase (Fig. 2b). Following M.SssI treatment, the methylation state of each target CpG was

assayed by bisulfite conversion and pyrosequencing and compared to treatment with control gRNA6. Pre-incubation of *Il33-pCpGl* with dCas9 and all CpG-targeting gRNAs resulted in a drastic, specific interference with DNA methylation at targeted sites (Fig. 2c–e). For example, in the case of gRNA3, the targeted CpGs (CpGs 9, 10, and 11) were methylated only to a mean ± SEM of $5.75 \pm 0.45\%$, whereas the control gRNA6 barely affected methylation and the sites were methylated at $84.79 \pm 0.88\%$ ($P < 0.00001$). Sites that were not directly within or adjacent to the binding site of dCas9:gRNA3 (CpGs 1, 2, 3, and 5) remained unaffected by the treatment (Fig. 2e) ($P = 0.752$, 0.878, 0.800, 0.618, respectively). The same levels of inhibition and specificity were achieved by two other CpG-targeting gRNAs (Fig. 2c, d). Notably, with gRNA2, we successfully prevented methylation of a single CpG while leaving all remaining assayed CpGs completely unaffected (Fig. 2d). We also reversed the order of the reaction, incubating the target DNA first with M.SssI and then with dCas9 and gRNA3 in order to ascertain that dCas9 is not able to catalytically remove methyl groups post hoc but rather inhibits methylation by competitive binding (Fig. 2g).

Now in possession of five *Il33-pCpGl* plasmids bearing unique methylation patterns (gRNA1, gRNA2, gRNA3, gRNA6, and mock), we sought to assay the impact of these patterns on transcription in live cells using a transient transfection reporter assay. We transfected each uniquely methylated plasmid into NIH-3T3 cells and performed a luciferase reporter assay (Fig. 2f). As mentioned previously, mock (unmethylated) plasmid drove luciferase activity to a significantly higher degree than both standard methylated and dCas9:gRNA6 treated plasmids. When CpGs 1, 2, 3 were unmethylated (by gRNA1 treatment) or CpG5 was unmethylated (by gRNA2), luciferase activity remained low and was not significantly different from gRNA6 control ($P = 0.202$, $P = 0.332$). However, in the case of unmethylated CpGs 9, 10, and 11 (by gRNA3) surrounding the *Il33-002* TSS, luciferase activity was significantly greater than gRNA6 ($P = 0.0007$) and not significantly different from mock-methylated DNA ($P = 0.157$), demonstrating that the methylation of these three TSS CpGs, but not the others, blocks *Il33-002* promoter activity. gRNA1 and gRNA3 both interfered with methylation of three CpGs and thus the overall promoter methylation levels were similar between these two treatments; yet, there was a stark difference in luciferase activity. These data demonstrate the exquisite impact of site-specific methylation rather than just methylation density, and thus this assay appears to capture the sequence specificity of inhibition of promoter function by DNA methylation.

In summary, we demonstrate that dCas9 specifically inhibits DNA methylation of targeted sites in vitro, enabling the analysis of the causal role of specific methylated sites per se. The only difference between our different transfected plasmids is the positions of the methyl moieties. No additional confounding enzyme is introduced. CpGs 9, 10, and 11 at the *Il33-002* TSS silence transcription; demethylation of these CpGs is sufficient for maximal activation of the promoter-reporter construct. In contrast, demethylation of CpGs 1, 2, 3, or 5 is insufficient for re-activation of the methylated promoter suggesting that methylation of these sites is not involved in silencing of transcription from the *Il33-002* promoter.

**Blocking of methylation by dCas9 is limited to its binding site and is affects both DNA strands**. In the preceding in vitro assays, we were able to prevent on-target DNA methylation with dCas9 without affecting the remaining target CpGs in the promoter. However, as the *Il33-002* promoter is CpG-poor and clusters of CpGs (e.g., 1, 2, 3 and 9, 10, 11) are separated by several hundreds

of base pairs, the precision of this approach needs to be determined. In order to delineate the DNA span that is protected from methylation by bound dCas9, we repeated the same in vitro assay using a canonical CpG-rich promoter. The human *CDKN2A* (p16) promoter contains a 310 bp fragment with 38 CpGs, which are frequently aberrantly hypermethylated in all common cancers[38]. We designed a gRNA overlapping a single CpG (CpG 17) within this promoter that was flanked on either side by CpGs 8 base pairs away from the 23-nucleotide gRNA and protospacer adjacent motif (PAM) sequence (Fig. 3a). We then applied bisulfite-cloning to map the methylation patterns of individual DNA molecules and assessed whether there was a difference in the methylation pattern of the CpGs in the strand bound by the dCas9:gRNA ribonucleoprotein and its complementary strand (as CpGs are palindromic).

In the control methylation reaction, M.SssI almost completely methylated all CpGs on both strands (Fig. 3b, d) with some sporadic unmethylated CpGs that are likely consequences of poor bisulfite conversion or Sanger sequencing errors; M.SssI is highly processive and it is unlikely that the sporadic demethylation resulted from inhibition of M.SssI[39]. In contrast, p16-targeting (CpG 17) gRNA completely inhibited methylation of the targeted CpG on the gRNA bound strand, while scrambled control gRNA did not block DNA methylation of CpG 17 (0% vs. 80% methylation, $P < 0.0001$, Fisher's exact test) (Fig. 3b, c). The CpG immediately downstream of the gRNA-PAM sequence was slightly but not significantly unmethylated (77% vs. 100% methylation, $P = 0.2292$, Fisher's exact test). Interestingly, the following CpG 19 was significantly unmethylated (38% vs. 100% methylation, $P = 0.0027$, Fisher's exact test), while the CpG only two additional base pairs downstream (CpG 20) was 100% methylated and unaffected. The distance between the unaffected CpG 20 and the 3′ end of the PAM is 14 bp and the upstream unaffected CpG 16 is 8 bp from the 5′ end of the gRNA (Fig. 3a). We thus define the range of dCas9 inhibition of M.SssI DNA methylation to be less than eight base pairs from the 5′ end and smaller than 14 base pairs from the 3′ end of the PAM adding to a total protection range of 45 bp. Nevertheless, peak inhibition is exactly at the binding site and any inhibition within the 45 bp is only partial.

It is interesting to also note that while the target CpG 17 is always protected from methylation in all of the molecules, CpG 18 and/or CpG 19 are protected only in certain DNA molecules. These data suggest that CpGs 3′ of the gRNA sequence are variably protected, possibly reflecting the dynamic orientation of the flexible gRNA scaffold[40]. It may thus be possible to refine this method to reduce or, conversely, target protection of neighboring CpGs. The results are in accordance with the crystal/cryo-EM structures of the dCas9:gRNA:DNA ternary complex, which reveals minimal 5′ protrusion of dCas9:gRNA beyond the 5′ end of the target DNA strand and more pronounced extension (and steric interference) of both dCas9 protein and gRNA scaffold beyond the 3′ end of the target DNA sequence, which still seats deep within the dCas9 binding pocket (Supplementary Fig. 2A)[40,41].

We also determined whether protection from methylation by dCas9 was symmetric on both DNA strands, and whether dCas9 preferably obstructed methylation of the targeted CpG only on the strand that was complementary to the gRNA. Given that bound dCas9 envelopes nearly the entire DNA double helix[40], we predicted that both CpG sites would be equally protected. Bisulfite-cloning of the opposite strand again revealed complete protection from M.SssI methylation of CpG 17 (0% vs. 100%, $P < 0.0001$) and the next CpG (8% vs. 90%, $P = 0.0003$) (Fig. 2d, e). Interestingly, the 3′ footprint is smaller by at least 2 bp (and at most 6 bp) than in the strand interacting with the gRNA, as CpG 19 is not affected

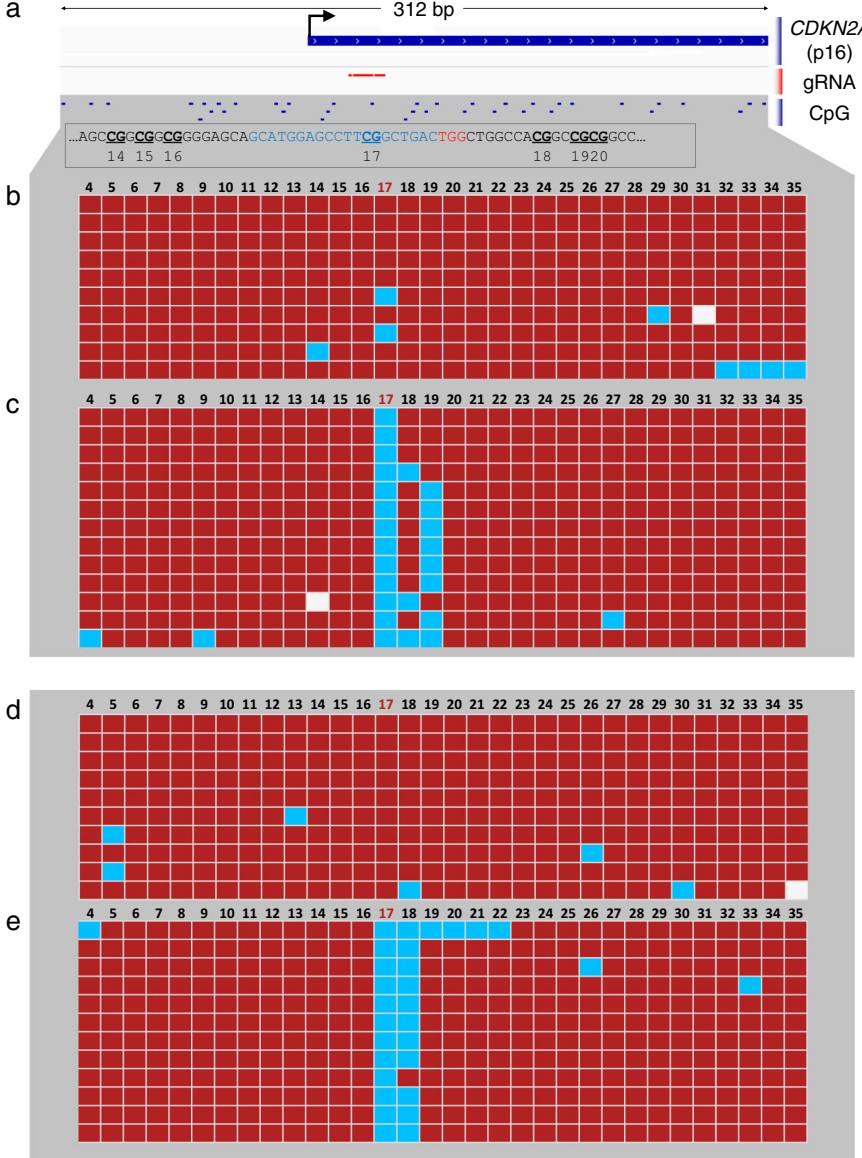

**Fig. 3 The footprint of dCas9. a** Genome browser diagram of the *CDKN2A* (p16) promoter region, which was used for the methylation assay, showing transcription start site (TSS, marked by black arrow), gRNA position overlapping CpG 17, and surrounding CpGs. Below, DNA sequence is shown in black, gRNA sequence in blue, and PAM site in red, with CpGs bolded, underlined, and numbered according to the figures that follow. **b–e** Methylation of individual strands of the *CDKN2A* promoter plasmid following standard methylation (**b**, **d**) or methylation preceded by incubation with dCas9 and p16 gRNA (**c**, **e**). Red squares indicate methylated CpGs and blue squares indicate unmethylated CpGs; white squares indicate no data. Figures **b** and **c** represent the forward strand whereas (**d** and **e**) represent the reverse strand. Figures generated by BISMA software (http://services.ibc.uni-stuttgart.de/BDPC/BISMA/). Regions below 80% methylation were filtered out as strands that were not effectively methylated by M.SssI. Source data are provided as a Source Data file.

on the antisense strand. Thus, dCas9:gRNA complex completely protected both the target and complementary CpG on the antisense strand.

We determined whether we could focus the range of protection using the smaller dCas9 protein from *Staphylococcus aureus* despite the fact that it classically requires a longer gRNA (21 bp instead of 20 bp) and a longer PAM sequence (NNGRR instead of NGG). We designed four *S.aureus* gRNAs (SAgRNAs1–4) that also overlapped with potential gRNAs for the hitherto utilized *Streptococcus pyogenes* dCas9 (SPgRNAs1–4) (Supplementary Fig. 2B). The first three gRNAs assayed the 5′ protrusion and were shifted by one base pair each in order to refine the 5′ distance for both dCas9 variants; three 5′ CpGs were 4, 7, and 11 bp away from the 5′ end of SAgRNA1; 3, 6, and 10 bp away for

SAgRNA2; and 2, 5, and 9 bp away for SAgRNA3. Each CpG was 1 bp further away for the corresponding SPgRNAs as these were 1 bp shorter at the 5′ end (20 bp vs. 21 bp). We determined that *S. aureus* dCas9 is equally capable of complete interference with M.SssI at sites within the bound region (CpGs 20–22), with a gradual 5′ fall-off in protection; 90–100% protection of CpG 2–4 bp away, 80% protection of CpG 5 bp away, 50–60% at CpG 6 or 7 bp away and 0–10% at 9–11 bp away from the target (Supplementary Fig. 2C and S2D). 5′ interference of SP-dCas9 was consistently less than SA-dCas9 at all distances in a manner that was not sufficiently explained by the additional single 5′ bp of the *S. aureus* gRNAs (Supplementary Fig. 2D). The 3′ distance for SP-dCas9 could not be refined further because of a lack of efficacy of SPgRNA 4 (Supplementary Fig. 2C and S2E); only four strands

appeared to have been protected from dCas9 (of 17 sequenced), and the interference was interestingly limited to CpGs in the PAM site and not within the gRNA binding site, likely indicative of a poor-quality gRNA. However, SAgRNA4 was efficient and we could calculate that SAdCas9 interfered with a minimum of 11 bp and a maximum of 13 bp from the 3′ end, including its 5 bp PAM sequence. Therefore, we demonstrate that despite its smaller protein size, SA-dCas9 has a 3′ footprint comparable to but possibly smaller than SP-dCas9 (likely due to similar gRNA scaffolds) and a definitively larger 5′ footprint, drawing the conclusion that the original SP-dCas9 allows more precise interference with DNMTs, however it is also useful to note that the equivalent efficacy of SA-dCas9 presents a secondary option for combinational approaches, and for a more diverse selection of target sequences by addition of a second PAM option.

**The dCas9 system directs robust site-specific demethylation in living cells.** dCas9 is obviously not an active demethylase; nevertheless, we hypothesized that we could use it to demethylate specific CpGs in dividing cells. As nascent post-replicative DNA is unmodified and must be methylated by the maintenance methyltransferase Dnmt1 in order to preserve parental cell methylation patterns[42], we postulated that dCas9 would interfere with Dnmt1 methylation similar to its blockage of M.SssI methylation and thereby cause passive demethylation of targeted sites through successive rounds of cell division and DNA replication. Therefore, we used the gRNAs characterized above to demethylate the endogenous *Il33-002* promoter in NIH-3T3 cells. We established by lentiviral transduction cell lines stably expressing SP-dCas9 and each *Il33* gRNA or a scrambled, non-targeting control gRNA (gRNAscr) and collected DNA for methylation analysis by bisulfite conversion and pyrosequencing 1 week after complete antibiotic marker selection (1 µg/mL puromycin). We demonstrate that the dCas9:gRNA complex is sufficient to produce robust demethylation of targeted CpGs (Fig. 4a–c). dCas9 in combination with gRNA1 (Fig. 4a) reduced absolute methylation levels by an average of 27.0% ($P < 0.0001$), 28.3% ($P < 0.0001$), and 34.6% ($P < 0.0001$) at CpGs 1, 2, and 3, respectively; gRNA2 (Fig. 4b) reduced CpG 5 methylation by 52.0% ($P < 0.0001$); gRNA3 (Fig. 4c) reduced CpG 9, 10, and 11 methylation by 30.2, 31.4, and 38.4% ($P < 0.0001$ for all). Demethylation with dCas9, unlike dCas9-TET (Fig. 1f) was highly specific to targeted CpGs, as in the case of gRNA2, no other assayed CpGs were demethylated. gRNA3 caused significant demethylation of off-target CpG 3 ($P = 0.002$) but the extent of demethylation was only 0.6%. gRNA1 caused a slightly larger, significant demethylation of the distant CpGs 9, 10, and 11 (5.3, 4.5, and 3.9%) but still to a lower level than the target CpGs 1, 2, and 3, and less than that of dCas9-TET:gRNA1. These data also clarify that the binding site demethylation in dCas9-TET and in dCas9-deadTET cells (Fig. 1e–g) likely stems from the same mechanism of steric interference with Dnmt1 rather than a catalytic TET activity, as the tightly bound dCas9 domain likely makes it impossible for the fused TET domain to access this bound DNA.

We were also able to demonstrate similar levels of demethylation and specificity by a second gRNA targeting CpGs 9, 10, and 11 which was shifted two base pairs in the 3′ direction (Supplementary Fig. 3A) relative to gRNA3, demonstrating that altering the exact CpG positioning relative to the gRNA, whether within the gRNA target sequence, PAM site, or immediately adjacent to either, does not impact demethylation efficiency in cells. All these positions were predicted to be completely protected from DNMT activity by both gRNAs based on the in vitro footprint assays (Fig. 3).

Though these experiments demonstrated a higher specificity of dCas9 than dCas9-TET across adjacent CpGs in the *Il33-002*

promoter, we also sought to determine if the same off-target effects seen with dCas9-TET could be found in equivalent dCas9 treated cells. Unlike in dCas9-TET cells, the distant *Il33-001* transcript was not upregulated by dCas9 combined with any of the three targeting gRNAs (Supplementary Fig. 3B); however, there was detectable significant downregulation of *Il33-001* under gRNA1. We found no potential off-target site for gRNA1 (no less than eight mismatches) within ± 10 kb from the *Il33-001* TSS.

Next, we wished to evaluate if the dCas9 demethylation approach could be optimized to yield higher demethylation. Passive demethylation by Dnmt1 interference would require cell division and, if fully efficient, methylation levels would halve with every round of replication. We therefore hypothesized that passaging the cells in culture would increase the extent of demethylation. dCas9:gRNA3 and dCas9:gRNAscr cell lines were passaged for an additional 30 days after the original DNA collection. This approach increased the extent of demethylation of only target CpGs 9 (14.3%, $P = 0.0009$), 10 (10.2%, $P = 0.003$), and 11 (15.5%, $P = 0.002$) (Fig. 4d). Passaged dCas9:gRNAscr cell lines were demethylated at several non-target *Il33-002* promoter CpGs compared to original unpassaged cells, but none of these differences were significant after correction for multiple testing.

Another common approach to improve the efficiency of CRISPR/Cas9 editing is cloning[43]. Despite the fact that we could achieve robust demethylation of a target CpG in a population of cells, as a particular strand of DNA only exists in a methylated or unmethylated state, we reasoned that we could isolate clonal populations that are completely demethylated at the target sites (CpG 9, 10, 11). Therefore, we expanded ten clonal lines from each of the dCas9:gRNA3 and dCas9:gRNAscr cell lines and subjected these clones to pyrosequencing. The population of gRNAscr clones (gray circles) was not significantly demethylated relative to the original gRNAscr pool at any CpG except a significant 0.6% demethylation at CpG 3 and, with the lone exception of a single CpG in one clone that displayed 39.5% methylation, no CpG in any of the ten clones was methylated less than 50% (Fig. 4e). Therefore, even though some gRNAscr cells in a population that is not 100% methylated must have fully unmethylated CpGs, the clonal isolation process is unable to generate fully demethylated clones, perhaps due to an equilibrium between methylation and demethylation established by the nuclear DNA methylation machinery in the cells. On average, dCas9:gRNA3 clones were not significantly demethylated at target CpGs 9, 10, and 11, compared to both original and passaged dCas9:gRNA3 lines. However, 6 of 10 clones isolated from the dCas9:gRNA3 pool displayed methylation levels below 11% at CpGs 9, 10, and 11 and two of these clones were methylated at or below 5% at all targeted CpGs. We concluded that we were able to produce cell lines with almost completely demethylated target CpGs with this approach (the small level of methylation detected in these clones is around the standard error for unmethylated controls in our pyrosequencing assay).

The clonal analysis suggests a clonal variation in the extent of demethylation by dCas9:gRNA. A plausible cause could be variation in the level of expression of either dCas9 or the gRNA. dCas9 mRNA levels did not correlate with methylation levels ($r = 0.1982$, $P = 0.6091$, $n = 9$) (Supplementary Fig. 3D) whereas gRNA3 expression levels correlated negatively with methylation ($r = -0.7307$, $P < 0.05$, $n = 9$) (Supplementary Fig. 3E). Similar to several other studies that demonstrated that expression of gRNA is the rate limiting factor in Cas9 cleavage efficiency[44–47], our data suggest that gRNA is the limiting factor in targeted demethylation efficiency in our model.

Clonal isolation is tedious, involves long passaging times, and prone to producing bottleneck effects from a heterogenous cell line; we also found that unhealthy morphologies were common to

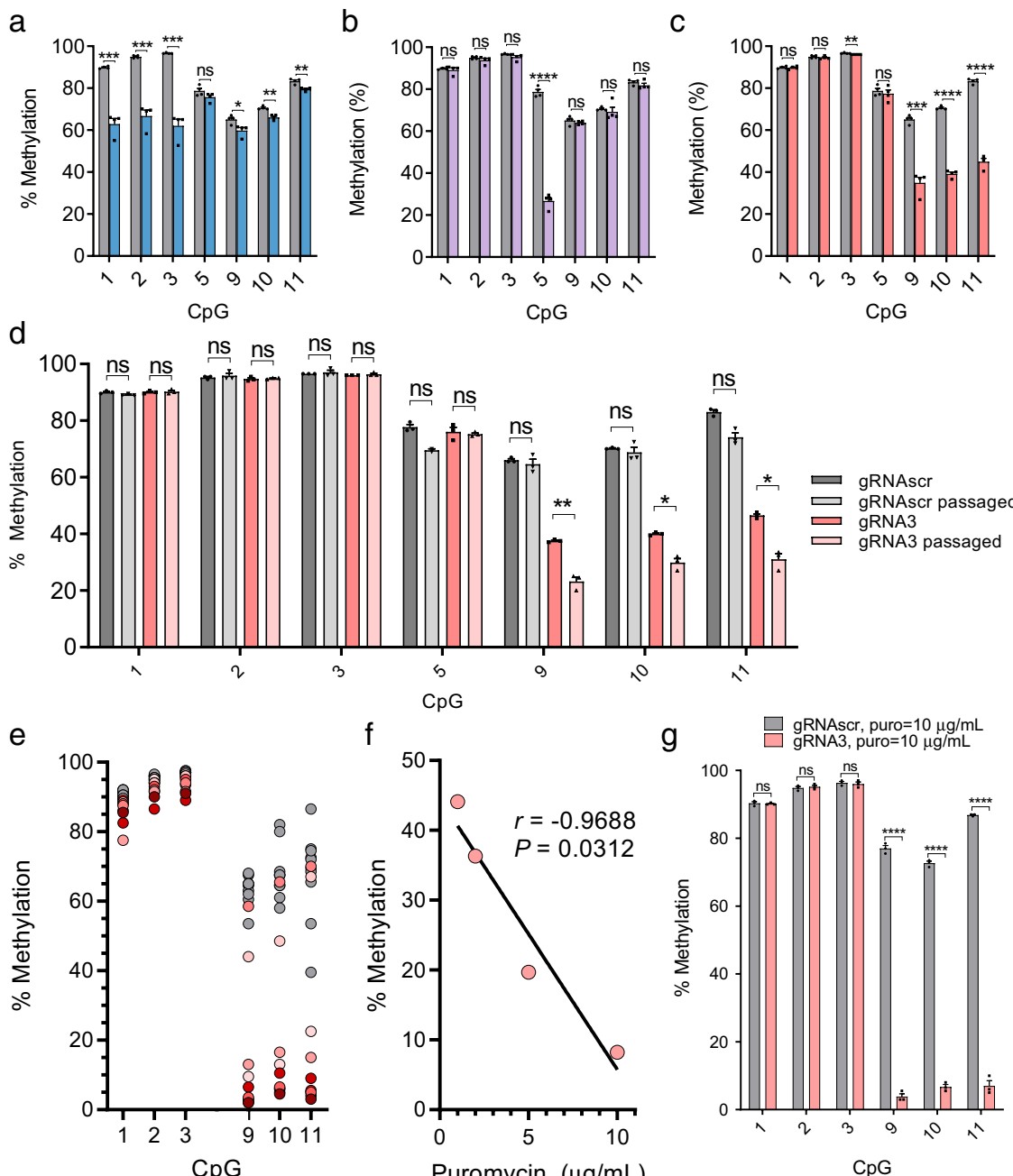

**Fig. 4 dCas9 causes demethylation in mammalian cells. a–c** Methylation levels (mean ± SEM) assayed by bisulfite-pyrosequencing at CpGs 1, 2, 3, 5, 9, 10, and 11 of the *Il33-002* promoter in NIH-3T3 cells stably expressing dCas9 and gRNA1 (**a** blue), gRNA2 (**b** purple), gRNA3 (**c** pink) or scrambled gRNA (**a–c**, gray; identical in all) ($n = 4$ biologically independent samples). **d** Cells from **c** were passaged for an additional 30 days and methylation percentage was assayed as previously ($n = 3$ biologically independent samples, mean ± SEM). **e** Cells from **c** were subjected to clonal isolation and expansion. Gray circles represent methylation levels of clones containing dCas9 and scrambled gRNA and various red circles represent methylation levels of randomly selected clones stably expressing dCas9 and gRNA 3 ($n = 10$ independent clones per condition). **f** Average DNA methylation at CpGs 9–11, assayed by bisulfite-pyrosequencing, as a function of increasing the selection antibiotic puromycin (lentivirus is expressing puromycin resistance gene) concentration in cell lines (pools) stably expressing dCas9 and gRNA3 ($n = 1$ cell line per puromycin concentration) fitted with a line of best fit. **g** DNA methylation at CpGs 1, 2, 3, 9, 10, and 11 in ($n = 3$ biologically independent samples, mean ± SEM) NIH-3T3 cells stably expressing dCas9 and gRNA3 (pink) or control gRNAscr (gray) and treated with 10 µg/mL puromycin until no antibiotic-associated cell death could be observed and surviving cells were of sufficient quantity for DNA extraction and other procedures (approximately 2 weeks). * indicates statistically significant difference of $P < 0.05$, **$P < 0.01$, ***$P < 0.001$, ****$P < 0.0001$, and ns not significant (Student's *t*-test, two-sided, with Holm-Sidak correction if number of tests is greater than 3). Source data are provided as a Source Data file.

these clonal populations (Supplementary Fig. 4). In order to increase gRNA transgene expression in the clonal population, we increased the quantity of puromycin, which we hypothesized would select for cells with higher copy numbers of virally-inserted transgenes and increase gRNA expression. We noted a stepwise increase in demethylation as puromycin concentrations were increased from the standard 1 µg/mL concentration to 2, 5, or 10 µg/mL (Fig. 4f) with a significant correlation ($P < 0.05$), and a large difference of 36% in extent of demethylation of the target sequences between minimal and maximal concentrations. Settling on 10 µg/mL, we produced high-puromycin selected populations of gRNAs1–3 and gRNAscr and verified the extent of demethylation. We found that dCas9:gRNA3-treated cells were highly demethylated at CpGs 9–11 with 3–10% residual methylation, compared to 71–87% in dCas9:gRNAscr cells with 10 µg/mL puromycin ($P < 0.000001$ for all), while off-target CpGs 1–3 were still highly methylated and unaffected by the treatment ($P = 0.742$, $0.621$, and $0.670$, respectively) (Fig. 4g). In summary, we successfully developed a protocol to produce near-complete, specific targeted DNA demethylation in cell lines and selected this optimized approach for future experiments.

**The effect of site-specific demethylation on *Il33* gene expression.** The next step was to assess the utility of our demethylation strategy in exploring the causal links between DNA demethylation at a specific region and transcriptional changes. We predicted that demethylation in this context would not be sufficient to activate transcription because dCas9 remains bound to the TSS and obstructs binding of transcriptional machinery, which is in itself an established technique to inhibit gene expression[33]. Accordingly, despite robust demethylation, high-puromycin dCas9:gRNA3 cell lines expressed significantly less *Il33-002* transcript than even scrambled cells (Supplementary Fig. 3F), whereas the *Il33-001* isoform was not significantly impacted in the same cells (Supplementary Fig. 3B). In fact, in contrast to the typical negative correlation between expression and DNA methylation, *Il33-002* expression was positively correlated with CpG 9–11 methylation level across dCas9:gRNA3 clones ($r = 0.74$, $P = 0.02$) (Supplementary Fig. 3G). This unique relationship likely originates from the fact that increased dCas9 on-target binding not only obstructs Dnmt1 activity, but also concurrently blocks access to RNApolII complex, inhibiting transcription.

To study the transcriptional consequences of promoter demethylation, dCas9 would need to be removed following demethylation to expose the newly unmethylated DNA to the nuclear environment. We tested transient gRNA expression with the aim that following several rounds of cell division, having caused demethylation of target DNA, gRNAs will be diluted and will not block binding of RNApolII. However, transient transfection of guide RNA molecules in a stably expressed dCas9 background resulted in only 15% on-target demethylation (Supplementary Fig. 5), and we determined to forego optimization of this strategy in favor of one compatible with the optimized high-puromycin protocol we had established. We implemented the Cre-lox system (Fig. 5a) that would allow complete dCas9 removal by Cre-mediated recombination only after demethylation is maximized.

We established high-puromycin selected NIH-3T3 cell lines expressing each lentiviral *Il33* gRNA and a lentiviral loxP-flanked dCas9 variant and validated successful demethylation (Supplementary Fig. 6A–C). One of the two base substitutions to render this dCas9 variant nuclease-dead (D10A, H840A) is different than the dCas9 used in previous experiments (D10A, N863A). We then used lentivirus-mediated gene transfer to introduce either

Cre recombinase or an empty control vector and verified successful dCas9 removal by Cre at the DNA level by PCR, using primers that produced a 500 bp fragment upon recombination (Supplementary Fig. 6D, E) and at the protein level by chromatin immunoprecipitation followed by quantitative PCR (ChIP-qPCR). ChIP-qPCR demonstrated elevated dCas9 binding to the *Il33-002* promoter region only in cells stably expressing dCas9 and gRNA3 but not in dCas9:gRNAscr cells regardless of Cre treatment (Supplementary Fig. 6F); in dCas9:gRNA3 cells, Cre recombination eliminated dCas9 binding to the *Il33-002* promoter to background levels with no significant difference from dCas9:gRNAscr cells. Interestingly, low levels of methylation persisted for at least 75 days after removal of dCas9 by Cre recombinase (Fig. 5b), indicating a lack of de novo methylation of this locus in these cells and the ability of this approach to modify DNA methylation in a stable manner despite elimination of dCas9.

Having generated targeted demethylation without bound dCas9 to hinder RNApolII binding to the TSS, we were then able to interrogate whether demethylation of the proximal promoter causes changes in expression of the gene. Expression levels of *Il33-002* transcript were measured by RT-qPCR. We detected a small but significant ($P = 0.0312$) increase in *Il33-002* expression in NIH-3T3 cells treated with dCas9:gRNA3 and Cre recombinase as compared to dCas9:gRNAscr, but not in dCas9:gRNA1 or dCas9:gRNA2 cells (Fig. 5c). This is consistent with our in vitro/transient transfection luciferase assays findings (Fig. 2f); both approaches suggest that methylation of TSS CpGs 9, 10, and 11 silence the basal *Il33-002* promoter.

It is possible that the small magnitude of induction of expression by demethylation of the TSS region can be explained by the presence of other methylated regulatory regions, or other required *trans*-acting factors that need to be demethylated to facilitate larger changes in expression. We used 5-aza-2′-deoxycytidine, a global demethylation agent, to assess whether demethylation of other sites would further induce the expression of TSS-demethylated *Il33-002*. Our results show that gRNAscr-bearing, gRNA1-bearing, and gRNA2-bearing cells, which were still methylated at the TSS, were still induced by the drug, while gRNA3 treated cells that were demethylated at the TSS were no longer responsive (Fig. 5d), suggesting that no further demethylation is required beyond demethylation of TSS sites 9, 10, and 11 for the activity of the basal promoter. To further corroborate that the lack of further induction by 5-aza-2′-deoxycytidine in cells with demethylated CpG sites 9, 10, and 11 was not a consequence of some other resistance to demethylation of dCas9:gRNA3 cells, we demonstrate that, in these dCas9:gRNA3 cells, the induction of the *Il33-001* isoform, driven by an untargeted upstream promoter, continued to be responsive to 5-aza-2′-deoxycytidine (Fig. 5e).

We verified that lack of further induction of gRNA3 demethylated *Il33-002* by a demethylating agent was not a result of an upper threshold of expression or our detection method, because treatment of cells with 1 µg/mL polyinosinic:polycytidylic acid (poly(I:C)) activated expression of *Il33-002* several hundred-fold after 4 and 8 h (Fig. 5f). Equally surprising was the fact that that dCas9:gRNA3 induced a 1.48X higher level of *Il33-002* expression than dCas9:gRNAscr counterparts at 8 h ($P = 0.0097$). However, the overall induction within each treatment group (poly(I:C) vs. control) was lower in gRNA3 cells ($401\times$) than in gRNAscr cells ($451\times$), because control-treated gRNA3 cells already had a higher baseline *Il33-002* expression as demonstrated here ($1.67\times$, $P = 0.1354$) and in Fig. 5c, d. Interestingly, this strong induction in response to poly(I:C) occurred in the complete absence of any detectable demethylation of the three TSS CpGs after 8 h (in gRNAscr cells), and even when incubation

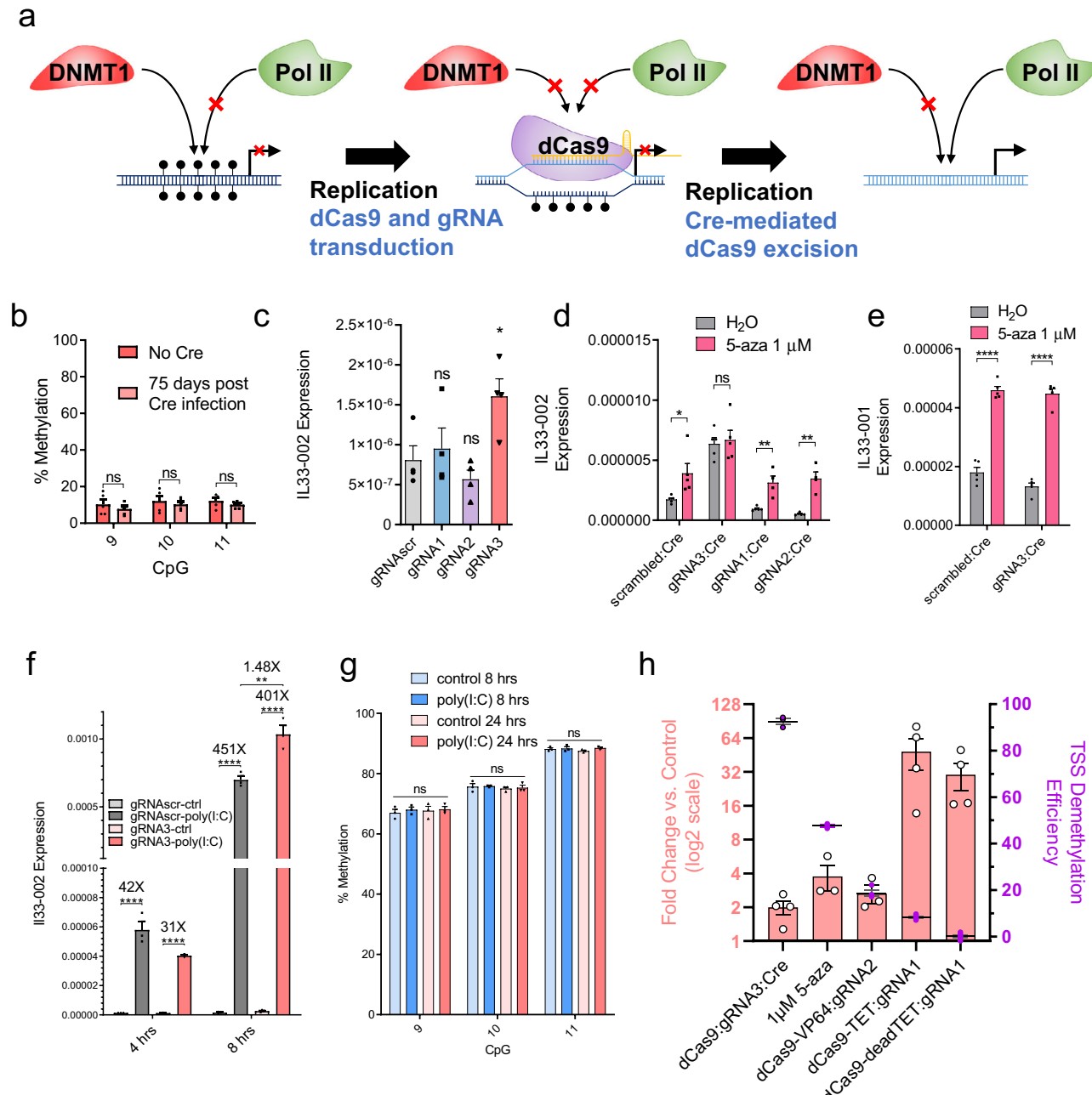

was extended to 24 h (Fig. 5g) nor of any other CpGs in the promoter (Supplementary Fig. 7A). These data suggest that DNA methylation suppresses basal activity of the *Il33-002* promoter but does not dramatically affect its inducibility, which can be independent of DNA methylation in the promoter region.

Histone deacetylase inhibition has been previously reported to act in combination with DNA demethylation to activate gene expression[48]. Activation of gene expression might require both demethylation and histone acetylation. We tested whether we can achieve a robust activation of the demethylated *Il33-002* with the histone deacetylase inhibitor trichostatin A (TSA, 50 nM). However, we only noticed a minor difference in the responses to treatment with TSA: in gRNAscr, gRNA1, and gRNA2 cells, TSA slightly reduced expression, and in gRNA3 cells, expression was not affected by TSA (Supplementary Fig. 7B). Thus, TSA inhibition of histone deacetylase activity does not add to the transcription activity of *Il33-002*. Finally, we determined whether

demethylation poises *Il33-002* to activation by other inducers. Lipopolysaccharide (LPS) has previously been shown to induce *Il33*[49]. In a pattern nearly identical to poly(I:C), treatment of NIH-3T3 cells with 100 ng/mL LPS induced the overall expression levels of *Il33-002* in gRNA3 cells, where the TSS is demethylated, to a larger extent (2.03×, $P < 0.01$) than in cells where *Il33-002* TSS is methylated (gRNAscr); however the fold change within each condition relative to phosphate-buffered saline (PBS) control was similar (2.43× in gRNA3 and 2.74× in gRNAscr) as a consequence of 2.28× higher baseline *Il33-002* expression in gRNA3 cells treated with PBS than gRNAscr treated with PBS, consistent with our observations in Fig. 5 (Supplementary Fig. 7C). This suggests that LPS can activate both the unmethylated and methylated *Il33-002*, but the total output increases once the promoter is demethylated. Alternatively, since *Il33-002* is not 100% methylated in control cells and in some cells the promoter is unmethylated (~20%), LPS might have induced

**Fig. 5 The effect of targeted promoter DNA demethylation on *Il33* expression. a** Diagram illustrating the principle of site-specific demethylation with dCas9 removal in order to facilitate transcription factor binding to the newly demethylated region. First, DNA is endogenously methylated by Dnmt1 with every round of replication and RNA polymerase II (RNA-polII) is not recruited to the promoter. After the introduction of dCas9 and a promoter-targeting gRNA, Dnmt1 is physically occluded from the locus and nascent strands of DNA are unmethylated, facilitating passive demethylation of the bound region. However, RNA-polII is also physically occluded by dCas9. If dCas9 is successfully removed, the unmethylated DNA no longer serves as a substrate for Dnmt1 and continues to remain unmethylated and RNA-polII may now be recruited. **b** Methylation of CpGs 9, 10, and 11 (mean ± SEM) which had been previously demethylated by high-puromycin gRNA3:dCas9 in NIH-3T3 cells, after 75 days of passaging following the lentiviral transduction of Cre recombinase (pink) or empty-vector control (red) ($n = 5$ biologically independent samples). **c** *Il33-002* expression (mean ± SEM) in NIH-3T3 cell lines stably expressing gRNAscr (gray) or gRNA1 (blue), gRNA2 (purple), or gRNA3 (pink) under high-puromycin conditions in combination with dCas9, followed by dCas9 removal by Cre recombinase as assayed by RT-qPCR and normalized to *Actb* expression. Statistical comparisons are to gRNAscr condition ($n = 4$ biologically independent samples). **d, e** *Il33-002* expression (**d** $n = 4$–5 biologically independent samples) or *Il33-001* expression (**e** $n = 5$ biologically independent samples) in NIH-3T3 cells from (**c**) following treatment wither water control or 1 μM 5-aza-2'-deoxycytidine, measured by RT-qPCR and normalized to *Actb* expression (mean ± SEM). **f** *Il33-002* expression (mean ± SEM) measured by RT-qPCR and normalized to *Actb* expression, in dCas9:gRNAscr (gray) or dCas9:gRNA3 (pink) NIH-3T3 cells following Cre recombinase treatment and then treated with poly(I:C) (1 μg/mL) or water control for 4 or 8 h ($n = 3$ biologically independent experiments). **g** DNA methylation (mean ± SEM) assayed by bisulfite-pyrosequencing in NIH-3T3 cells expressing dCas9, gRNAscr, and Cre treated with 1 μg/mL poly(I:C) or water control for 8 and 24 h ($n = 3$ biologically independent experiments). **h** Summary of maximal *Il33-002* induction (mean ± SEM) (left *y*-axis, pink bars; data in log2 scale but axis numbering is not transformed) and maximal promoter demethylation (purple, right *y*-axis, calculated as percent unmethylated divided by control methylation) under different treatments presented thus far (*x*-axis: dCas9, 5-aza-2'-deoxycytidine, dCas9-VP64, dCas9-TET, and dCas9-deadTET). Where relevant, data for maximally inducing/demethylating gRNA is shown. $n = 3$–6 biologically independent cell cultures * indicates statistically significant difference of $P < 0.05$, **$P < 0.01$, ***$P < 0.001$, ****$P < 0.0001$, and ns not significant (Student's *t*-test, two-sided, with Holm-Sidak correction if number of tests is greater than 3). Source data are provided as a Source Data file.

the unmethylated copies in the control cells explaining the lower total output in the control cells. However, the ratio of unmethylated Il33 promoter (20%) in the untreated cells relative to the demethylated cells (90%) (0.22) is lower than the ratio of expression in control and demethylated cells following LPS induction (0.5). The data are consistent with the hypothesis that at this locus methylation can silence basal promoter activity but not affect inducibility.

In summary, we show that near-complete demethylation of *Il33-002* TSS using an enzyme-free approach results in only a mild two-fold induction of basal gene expression, whereas other approaches that cause smaller degrees of demethylation can produce larger changes in gene expression, such as dCas9-TET:gRNA1, which produces only a 10% demethylation but a 50-fold gene induction (Fig. 5h).

**dCas9 off-target demethylation events and comparison to dCas9-TET.** In order to determine the specificity of dCas9-targeted demethylation and compare it to that of the dCas9-TET method, we performed whole-genome bisulfite sequencing (WGBS) of control (untreated) NIH-3T3 cells, dCas9-TET:gRNA3 and dCas9-TET:gRNAscr NIH-3T3 cells (from Fig. 1), as well as dCas9:gRNA3 and dCas9:gRNAscr NIH-3T3 cells subsequently treated with lentiviral Cre recombinase (Cre rationale provided in the previous section) ($n = 3$). To understand the changes imposed by these treatments at a global level, we first performed an analysis of CpG methylation clustering of high-coverage (≥10×) CpGs genome-wide, from which it was apparent that cells modified with the dCas9 method clustered with untreated control cells, whereas dCas9-TET cells were (1) more divergent from control cells and (2) unable to be clustered within gRNAscr vs. gRNA3, reaffirming the global effects of TET despite locus-specific targeting by dCas9:gRNA (Fig. 6a). dCas9-TET cells with both gRNAs were also significantly less methylated genome-wide than untreated cells ($P < 0.01$) and dCas9 counterparts ($P < 0.01$ for gRNAscr, $P < 0.01$ for gRNA3) to such an extent that they failed to demonstrate the typical genomic hypermethylation in response to lentiviral integration[50,51] that dCas9 cells demonstrated (Fig. 6b).

For all significantly differentially hypomethylated CpGs (differentially methylated CpGs, dmCpGs), we defined the

following thresholds: covered ≥5× in all replicates from treatments in question, *q*-value (SLIM-adjusted *p*-value) less than 0.01, and a difference in methylation ≥25%. When comparing all treatment conditions to untreated controls, there were 54 dmCpGs in dCas9:gRNAscr, 338 in dCas9:gRNA3, 3940 in dCas9-TET:gRNAscr, and 6286 in dCas9-TET:gRNA3. Due to differences in sample-level read depth (Supplementary Table 4), the direct comparison of the numbers of dmCpGs could suffer from coverage bias, therefore, the incidence of dmCpGs as a fraction of all CpGs assayed (≥5× covered in all six samples) under each comparison to untreated cells are as follows: dCas9:gRNAscr = 54/9,039,707 (0.0006%), dCas9:gRNA3 = 338/9,903,308 (0.0034%), dCas9-TET:gRNAscr = 3940/7,503,634 (0.0525%), dCas9-TET:gRNA3 = 6286/10,931,608 (0.0575%). Accordingly, after this normalization, dCas9-TET produces 16.9× (gRNA3) to 87.5× (gRNAscr) more demethylated CpGs. Furthermore, as dCas9:gRNA cells serve as better experimental controls (e.g., for lentiviral integration) than untreated controls, comparisons of dCas9-TET cells to dCas9 cells expressing the same gRNA are more appropriate and result in the following numbers of hypomethylated dmCpGs: dCas9-TET:gRNAscr = 26,860/8,216,634 (0.3269%), dCas9-TET:gRNA3 = 98,568/13,290,423 (0.7416%) (Fig. 6c right, D top and middle panel). These data emphasize the genomic hypomethylation burden of dCas9-TET, and establish that genomic hypomethylation of the dCas9 demethylation method to be far more limited.

Next, we defined off-targets of dCas9:gRNA3 by comparison to dCas9:scr under the same minimum coverage and statistical conditions described above (Fig. 6d, e bottom panel) and found a total of 643 dmCpGs (Supplementary Data 1). Interestingly, the top two dmCpGs in terms of statistical significance were the target *Il33-002* CpGs 10 ($q = 2.53 \times 10^{-5}$) and 11 ($q = 3.03 \times 10^{-6}$) (Fig. 6e, circled in red). Upon further inspection, the third target CpG, CpG 9, failed to be identified in this analysis because it was 4× covered in one sample (Supplementary Table 5) but was significantly demethylated in this dataset (87.45% vs. 8.15% $P = 0.0011$, *t*-test). The highest ranked off-target dmCpG was chr8:4802686 (Fig. 6e, circled in black), yet the highest scoring sequence match to the gRNA3 target within ±50 bp from this CpG (using CCTop[52], an algorithm that identifies and ranks off-targets on both DNA strands by position and number of mismatches) had 13 mismatches to

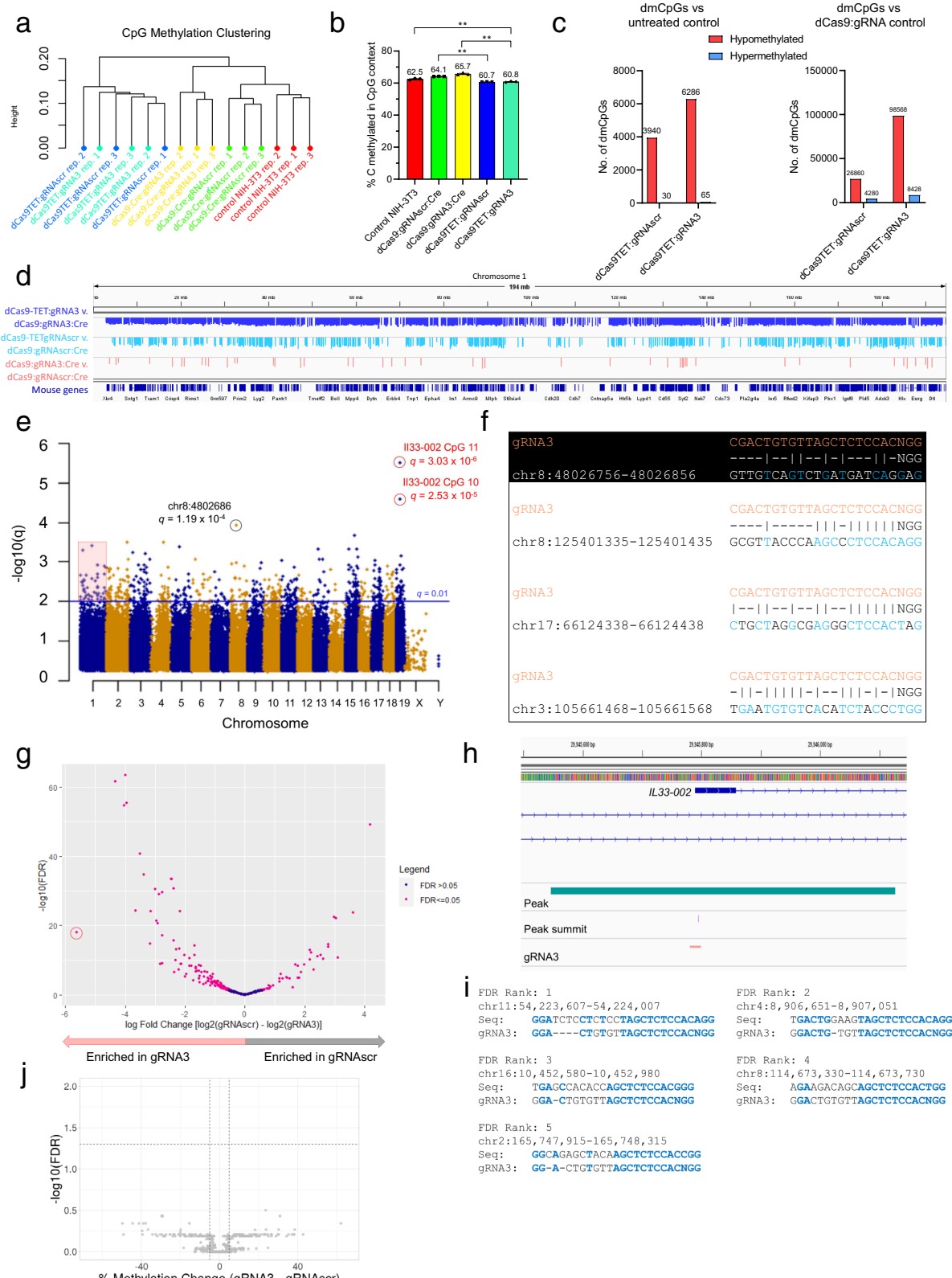

gRNA3 (Fig. 6f, black panel), including one in the most deleterious position, immediately adjacent to the PAM, and a non-standard NAG PAM, making it unlikely to be demethylated as a consequence of off-target dCas9 binding.

To see if any of the 641 off-target dmCpGs had any sequence similarity to gRNA3, we first compiled a comprehensive list of 100 bp regions surrounding all possible gRNA3 off-targets in the murine genome of up to four mismatches and one gap (4436 total), representing what is typically accepted to be the maximum number of tolerated mismatches by CRISPR/Cas9[52] (generated by combining lists from four online tools CRISPR DESIGN (crispr.mit.edu; deprecated), OFF-Spotter[53], CCTop[52], and OFF-Finder[54]) and searched this list for the presence of the dCas9 off-target dmCpGs. None of the 4436 potential off-target sites

**Fig. 6 WGBS and ChIP-seq analyses of dCas9 and dCas9-TET approaches to targeted demethylation. a** Clustering of NIH-3T3 samples with indicated lentiviral treatment and replicate number by CpG methylation, based on highly covered (≥10×) CpGs common to all samples with the cluster using Samples function in the methylKit package for R (ward.D2 method). **b** Fraction of total sequenced CpGs (mean ± SEM) that read as methylated (after bisulfite conversion) in each treatment type, aligned and calculated with Bismark default parameters ($n = 3$, biological replicates; **$P < 0.01$ with two-sided Student's $t$-test). **c** Number of significantly differentially methylated CpGs (dmCpGs) (red = hypomethylated, blue = hypermethylated) determined by methylKit calculateDiffMeth function (≥5× coverage, $n = 3$, $q$-value ($p$-value adjusted for multiple testing by SLIM method) <0.01, 25% methylation difference) of dCas9TET:gRNAscr and dCas9TET:gRNA3 NIH-3T3 cell lines compared to untreated control NIH-3T3 cells (left) or compared to dCas9:gRNAscr:Cre or dCas9:gRNA3:Cre, respectively. **d** Genome browser view of mouse (mm10 genome) chromosome 1, with bedGraphs containing hypomethylated dmCpGs (and amount of hypomethylation in %) in dCas9-TET:gRNA3 (top, blue) and dCas9-TET:gRNAscr (middle, light blue) from (**c**, right, chromosome 1 only) and dCas9:gRNA3:Cre hypomethylated dmCpGs compared to dCas9:gRNAscr (bottom, pink), which are the same as pink inset in **e**. Range is 0 to −100. Gene structures are densely mapped and sparsely labeled at the bottom (dark blue). **e** Manhattan plot of all hypomethylated (>25% change in methylation) sites in dCas9:gRNA3:Cre cells compared to dCas9:gRNAscr:Cre. Significantly differentially methylated sites were considered under default methylKit conditions ($q < 0.01$, above horizontal blue line). CpGs circled in red and labeled with q-values represent two target *Il33-002* CpGs (10 and 11) and the CpG circled in black represents the third-highest (top non-target) dmCpG ranked by $q$-value. Pink box highlights significant dmCpGs in chromosome 1, which are displayed in **d**. **f** Best alignments to gRNA3 and PAM sequence of four 100 bp regions surrounding (±50 bp) four selected off-target dmCpGs (of 641 total) from **e**. Dashes indicate mismatches. Vertical lines indicate matching base pairs. Matched base pairs in off-target region are also shown in blue. For the purposes of this representation, the adenine in the NAG PAM in considered a mismatch to the more active guanine in the NGG PAM. Top alignment (in black) shows the region containing the top off-target dmCpG by $q$-value, chr8:4802686, circled in black in **e** (14/23 mismatches). Second from top alignment displays the 100 bp off-target region containing a sequence with the most similarity to gRNA3 of all 641 100 bp off-target regions, as calculated by position-weighted mismatch algorithm CCTop (ten mismatches). Third from top alignment displays off-target region ranked second for similarity to gRNA3 by CCTop (eight mismatches but one is closer to 3′/PAM end than above). Final alignment shows the off-target region with the lowest mismatches overall to gRNA3, regardless of position, which is 7. **g** Volcano plot of ChIP-seq significantly differentially enriched regions in gRNA3 and gRNAscr conditions ($n = 3$) for anti-FLAG ChIP-seq against FLAG-dCas9, using input as control. Log2 fold change (log2(gRNAscr) − log2(gRNA3)) is plotted on the x-axis and −log10(False Discovery Rate) is plotted on the y-axis. The locus corresponding to the *Il33-002* transcription start site is circled in red. **h** Genome browser view (mm10) of *Il33-002* (blue). Statistically significant peak (turquoise) (circled in red in **g**), peak summit (purple) and gRNA3 sequence (pink) are labeled. **i** Manually curated sequence alignments of top five DERs from ChIP-seq data to gRNA3 and PAM sequence. Identical sequence matches are marked in blue and bolded. Potential gaps are indicated with dashes. Chromosomal locations are given for the mm10 genome build. **j** Volcano plot depicting changes in methylation and associated statistical probabilities in dCas9:gRNA3 NIH-3T3 cells (from WGBS data) for CpGs that are within 150 differentially enriched off-target regions bound by dCas9:gRNA3 (*Il33-002* is excluded). Change in methylation (x-axis) is expressed as mean percent methylation in dCas9:gRNAscr subtracted from mean percent methylation in dCas9:gRNA3. Statistical probabilities are provided as the −log10 of the p-value derived by the independent $t$-test as corrected for multiple testing by the False Discovery Rate method. Five hundred and forty-nine CpGs were located in DERs after filtering for CpGs that were 5× covered in all six samples; however, 132 CpGs with exactly 0% methylation in all six samples and 13 CpGs with exactly 100% methylation in all six samples are not depicted as $P$-values cannot be mathematically calculated. Therefore, 404 CpGs are shown. Underlying WGBS data was presented above and reflects $n = 3$ independent stable cell lines. Source data are provided as a Source Data file.

overlapped with equal-sized 100 bp regions containing the significantly hypomethylated off-target CpGs. In an effort to find any similarity of dmCpGs to gRNA3, we again invoked CCTop to search the list of 100 bp regions surrounding the dmCpGs to identify the highest sequence similarity to gRNA3 and found the following top off-candidate regions (Fig. 6F): chr8:125401335–125401435 with ten mismatches but a complete 6-bp match to the 3′ (seed) region and 9 of 10 matches to the ten most 3′ nucleotides and chr17:66124338–66124438 with a similar 6 bp complete 3′ match, only eight mismatches, but an NAG pam instead of NGG PAM. The fewest possible mismatches to any sequence within the 100 bp dmCpG-containing regions was 7, but this scored lower as it included two mismatches 2 and 4 bp from the PAM, which is not compatible with dCas9 binding. The complete list of CCTop-generated mismatched (up to 18 mismatches) off-targets is available in Supplementary Data 2.

Given the facts that dCas9 is much less tolerant to mismatches in the seed region (5–12 bp nearest to the PAM), NAG PAMs display an estimated one-fifth to one-tenth of NGG PAM activity, and most importantly, that although there is more tolerance for mismatches in the 5′ region there seem to be no reports in the literature of activity with more than five mismatches anywhere in the sequence[55,56], we hypothesized that none of the dmCpGs are genuine off-targets of gRNA3. To test this hypothesis, we performed chromatin immunoprecipitation sequencing (ChIP-seq) with an anti-FLAG antibody in cells expressing FLAG-tagged dCas9 and either gRNA3 or gRNAscr. We found 151 significantly differentially enriched regions (DERs) of dCas9:gRNA3 (FDR <

0.05) and 44 DERs in dCas9:gRNAscr (Fig. 6g and Supplementary Data 3). The most enriched locus (by fold change) was the targeted *Il33* promoter and the summit of the peak (highest fragment pileup and predicted binding spot) was within the gRNA3 target sequence (Fig. 6h). Other DERs included six of the 4436 off-target sites predicted above. Manual analysis of the top five gRNA3 DERs (sorted by FDR) revealed considerable sequence similarity to gRNA3, with 100% alignment of a 10–11 bp seed region and PAM (Fig. 6i). Importantly, none of the 150 (excluding the *Il33-002* DER) gRNA3 DERs overlapped with the 641 DMGs from the WGBS data: the only region demethylated and bound by dCas9 was *Il33-002*, reinforcing the high specificity of this approach. Moreover, restricting differential methylation analysis to only the 549 CpGs (with a minimum coverage of 5× in all six samples) located within the 150 off-target DERs bound by dCas9:gRNA3 revealed no statistically significant differentially methylated sites, and no otherwise apparent trend that favors nonsignificant hypomethylation over hypermethylation (Fig. 6j).

These data suggest that the dmCpGs may originate from an activity that is not the off-target binding of dCas9:gRNA, such as differential epigenetic drift during cell passaging[57], global epigenetic change as a response to lentiviral integration[50,51], technical variability in WGBS, or by insertional mutagenesis and lentiviral integration into gene-regulatory elements that could also lead to modified expression of epigenetic editing enzymes[58]. To address some of these potential factors, we analyzed split sequencing reads from our WGBS data (see "Methods" section

and Supplementary Software 1) to identify 2792 possible lentiviral insertion points across all six replicates and found that of 641 off-target dmCpGs, 13 are within ±5 kb from viral insertion sites ($P = 0.0322$, hypergeometric) and 97 are within ±50 kb ($P = 0.00729$, hypergeometric). Additionally, as it is known that lentivirus integration predominantly results in genomic hypermethylation[50,51], we wondered if dCas9 off-target dmCpGs were identified as hypomethylated in dCas9:gRNA3:Cre because these sites were aberrantly hypermethylated in dCas9:gRNAscr:Cre, rather than by direct demethylation in dCas:gRNA3:Cre cells. Indeed, three dmCpGs, including the top hypomethylated by $q$-value—chr8:4802686 (Fig. 6e, circled in black)—were significantly hypermethylated dmCpGs in dCas9:gRNAscr:Cre as compared to untreated control cells. As a lack of statistical significance in these sites compared to untreated does not discount a lack of statistical significance compared to dCas9:gRNA3:Cre (for example, these sites can show less variability in dCas9:gRNA3:Cre than untreated), we were prompted to see what fraction of the 641 off-target dmCpGs were generally hypermethylated in dCas9:gRNAscr:Cre as compared to untreated control cells. Of the 641 dmCpGs, 424 were sufficiently covered (≥5×) in all six dCas9:gRNAscr:Cre and untreated samples. Of these 424 sites, 379 (89%) were generally hypermethylated. 246 of these (65%) were nominally significant ($P < 0.05$, one-sided $t$-test) and 179 were still significant after correction for multiple testing (false discovery rate) (Supplementary Data 4).

We also used targeted-bisulfite pyrosequencing to assess whether dCas9:gRNA3 caused demethylation of the top five predicted candidate off-target CpGs for gRNA3 and found that there was no observable change in methylation of any of the top-predicted off-targets (Supplementary Fig. 3C and Supplementary Table 6).

Interestingly, under the same analysis conditions, there were no significantly differentially methylated CpGs between dCas9-TET:gRNA3 and dCas9-TET:gRNAscr, further emphasizing the non-specific activity of the TET domain even when it is targeted by the CRISPR system. To provide further evidence that the dCas9-TET hypomethylated dmCpGs (Fig. 6c, right) might originate as a consequence of TET-directed (rather than dCas9-directed) interaction with DNA of the dCas9-TET fusion protein, we analyzed whether these dmCpGs are enriched in established sites of TET action: enhancers[59–64]. In dCas9TET:gRNA3 cells, 815 of 106,966 dmCpGs (0.76%) could be found in mouse enhancers (FANTOM5 project[65], mouse_permissive_enhancers_phase_1_and_2.bed.gz) compared to 89,922 of all 13,290,423 ≥5× covered CpGs (0.68%) ($P = 2.93 \times 10^{-5}$, hypergeometric). There was also a significant enrichment of dmCpGs in enhancers in dCas9TET:gRNAscr cells, where 244 of 26,860 of dmCpGs were in enhancers while 56,417 of all 8,216,634 ≥5× covered CpGs (0.91% vs. 0.69%) were in enhancers ($P = 3.03 \times 10^{-6}$, hypergeometric). Importantly, an even greater fraction (46 of 4174 or 1.1%) of shared dmCpGs between dCas9TET:gRNA3 and dCas9TET:gRNAscr were found in enhancers. Of all regions containing predicted gRNA3 off-targets of up to four mismatches and one gap (100 bp around cut site), 21 and five were within 100 bp of dmCpGs in dCas9-TET:gRNA3 and dCas9-TET:gRNAscr, respectively. 45 and 24 of these gRNA3 and gRNAscr dmCpGs, respectively, were bound by dCas9 in the ChIP-seq data.

**dCas9-based demethylation analysis of the role of TSS methylation in *SERPINB5*, *Tnf*, and *FMR1* genes**. Our previous results show that methylation of *Il33-002* TSS silences basal promoter activity but that demethylation does not result in robust activation of the gene. Induction of this gene could occur independently of methylation of the promoter. We therefore examined whether TSS (de)methylation might play similar or different roles in other genes.

We next examined the *SERPINB5* gene, which encodes the tumor suppressor maspin and is methylated and transcriptionally silenced in human MDA-MB-231 breast cancer cells. Reactivation of this gene has been reported to increase cell adhesion and therefore decrease growth, invasion, and angiogenesis[66–70]. Several studies have reported that DNA methylation of the *SERPINB5* promoter negatively correlated with gene expression in human cancer and that 5-aza-2′-deoxycytidine treatment is sufficient to restore *SERPINB5* expression[71–75].

We designed a single gRNA targeting six CpGs (three within the gRNA binding site and three within 11 bp of the 3′ end of the gRNA, as predicted to be completely affected by our in vitro footprint assays in Fig. 3) in the core *SERPINB5* promoter and specifically in the transcription-regulatory GC-box (Fig. 7a). In this case, increasing puromycin had a mild effect in increasing the frequency of unmethylated promoters and even the highest puromycin concentrations (40 µg/mL) resulted in demethylation of only 20% (Supplementary Fig. 8). We reasoned that perhaps there is a strong selection against cells expressing *SERPINB5*—which is a known tumor suppressor—resulting in overgrowth of cells bearing highly methylated *SERPINB5*. Therefore, we turned to the previously described clonal isolation strategy. We picked approximately 20 clones from each of the two treatments (gRNAscr and gRNASERPINB5) and evaluated methylation by pyrosequencing, which revealed a significant demethylation in gRNASERPINB5 MDA-MB-231 clones on average comparted to gRNAscr clones (Fig. 7b). We found that numerous clones were completely demethylated (Fig. 7c) and we selected five gRNA-SERPINB5 clones with methylation levels below 5% at all six CpGs as well as five representative gRNAscr clones for dCas9 removal by Cre-mediated recombination and subsequent analyses. Methylation levels of gRNAscr clones and gRNASERPINB5 clones ($n = 3$) remained constant for at least 45 additional days of passaging after dCas9 removal, though there appeared to be a small non-significant trend of increasing methylation in demethylated gRNASERPINB5 clones (Supplementary Fig. 8b, c). Surprisingly, despite the large change in methylation, *SERPINB5* expression after Cre-mediated dCas9 removal remained unchanged between the two sets of clones, though there was a small insignificant ($P = 0.105$) increase in the variance of expression levels in the demethylated clones (Fig. 7d). The difference in *SERPINB5* expression was increased when these cells were further subcloned in order to reduce the potential of selection against cells with activated *SERPINB5* expression (Fig. 7E), but not to a statistically significant degree ($P = 0.0767$), suggesting that demethylation of the *SERPINB5* promoter is insufficient to activate the gene. Since 5-aza-2′-deoxycytidine was shown to induce the gene, we tested whether induction of the gene requires additional demethylation beyond the gene TSS by analyzing whether 5-aza-2′-deoxycytidine would induce the methylated and unmethylated *SERPINB5* promoter to the same extent. In contrast to *Il33-002*, which was not further induced by 5-aza-2′-deoxycytidine after TSS demethylation, expression of *SERPINB5* with a demethylated TSS region was significantly increased by 5-aza-2′-deoxycytidine treatment as compared to gRNAscr cells treated with 5-aza-2′-deoxycytidine (Fig. 7f) ($P = 0.0184$) (4.85× in gRNASERPINB5 vs. 2.59× in gRNAscr). This is consistent with the conclusion that demethylation of the promoter is insufficient for its expression and demethylation of other regions, such as the depicted enhancer regions (Fig. 7a), is required for induction of *SERPINB5*; however, basal promoter demethylation contributes to

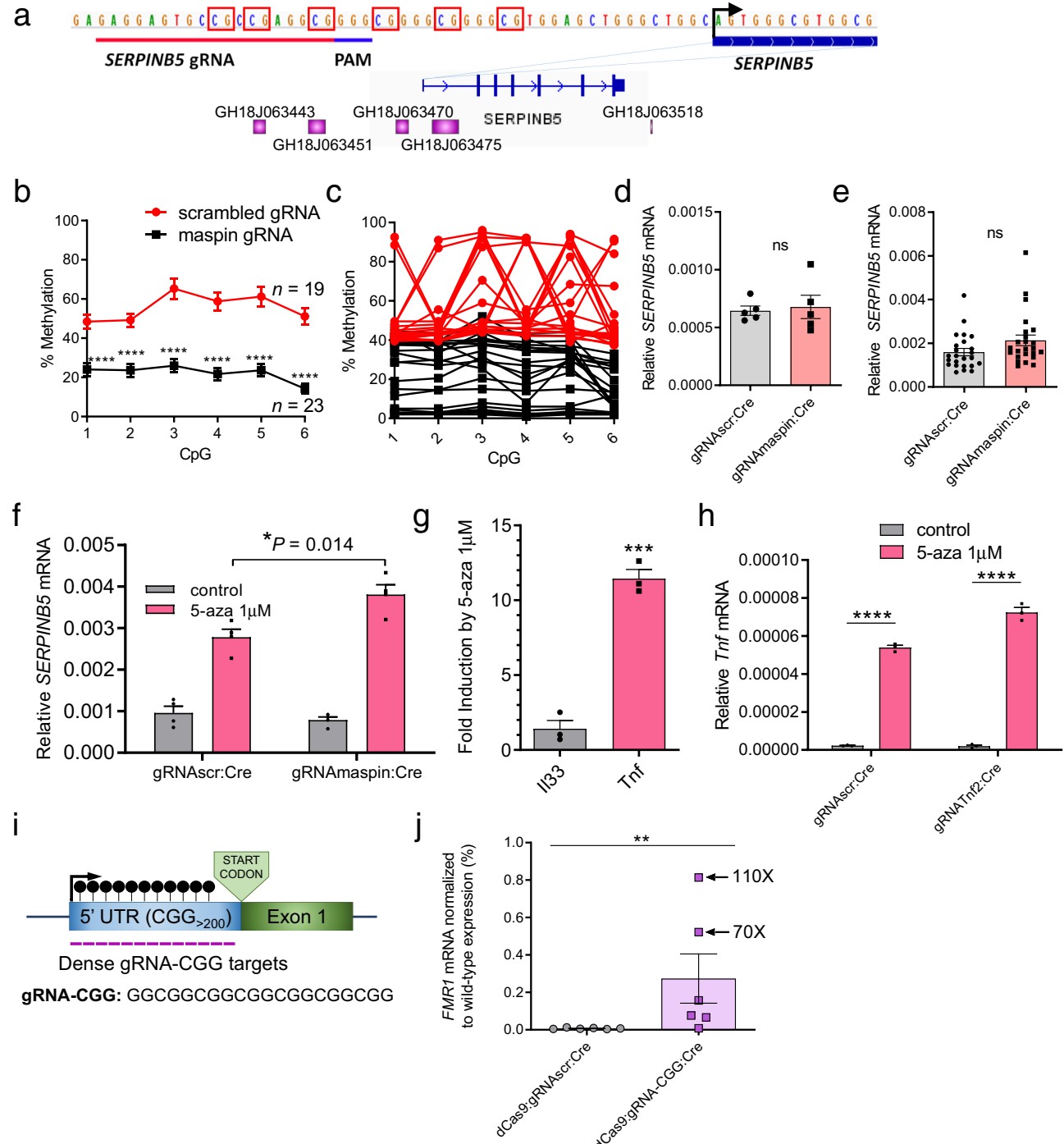

the overall expression level following demethylation of other regions.

We then questioned whether larger changes in expression could follow demethylation of proximal promoters in other genes. To identify genes that may potentially display such changes, we selected 17 candidate genes in NIH-3T3 cells with large expression fold changes in response to 5-aza-2′-deoxycytidine in a publicly available microarray dataset (GEO GSE8374) and analyzed their expression changes by RT-qPCR following 1 μM 5-aza-2′-deoxycytidine treatment (Supplementary Table 7). We selected the *Tnf* gene which was heavily methylated at the proximal promoter region and the expression of which was increased by more than ten-fold by 5-aza-2′-deoxycytidine treatment (Fig. 7h). We tested six gRNAs under high-

puromycin selection (20 μg/mL) conditions and identified a gRNA that demethylated all ten CpGs in approximately 200 bp upstream of the *Tnf* TSS (Supplementary Fig. 9A–C). We chose this gRNA (gRNATnf2) for Cre recombinase removal of dCas9. Surprisingly, complete *Tnf* promoter demethylation did not result in a significant difference in *Tnf* expression compared to gRNAscr (Fig. 7h) nor could we observe any difference in expression in subclones from these cell pools (Supplementary Fig. 8D). However, when these cells were treated with 5-aza-2′-deoxycytidine, the demethylated gRNATnf2 cells were induced to a larger extent than the methylated gRNAscr pools (36-fold vs. 24-fold) (*P* = 0.0008) (Fig. 7h). Therefore, we conclude that, similar to demethylation of *SERPINB5* TSS, demethylation of *Tnf* basal promoter contributes to expression but is insufficient to

**Fig. 7 The effect of dCas9-based demethylation of TSS on expression of _SerpinB5_, _Tnf_, and _FMR1_ genes. a** (Top) Schematic of the human _SERPINB5_ promoter region, including the start site of transcription (marked by black arrow) and the binding site and PAM of the _SERPINB5_ gRNA. CpG sequences are boxed in red. (Bottom) _SERPINB5_ gene with purple boxes indicating enhancer positions relative to gene body: enhancer IDs correspond to the GeneHancer database. **b** DNA methylation level of each CpG averaged over $n = 19$ gRNAscr (red) and $n = 23$ gRNASERPINB5 (black) independent MDA-MB-231 clones isolated from three independent treatments of cell cultures as assessed by pyrosequencing (mean ± SEM). **c** Same data as (**b**) except now shown as the calculated methylation fraction for each of the 19 gRNAscr (red) and 23 gRNASERPINB5 (black) clones, rather than the average of all clones. **d** _SERPINB5_ expression levels measured by RT-qPCR and normalized to _GAPDH_ expression levels for five gRNAscr and five lowly-methylated gRNASERPINB5 clones (mean ± SEM, $n = 5$ biologically independent clones). **e** _SERPINB5_ expression levels (mean ± SEM) measured by RT-qPCR and normalized to _GAPDH_ expression levels for 48 ($n = 24$ for each treatment) MDA-MB-231 clones subcloned from the clones in (**d**). **f** _SERPINB5_ expression levels (mean ± SEM) measured by RT-qPCR and normalized to _GAPDH_ expression levels for clones from (**d**) following treatment with 1 μM 5-aza-2′-deoxycytidine or water control ($n = 5$ biologically independent experiments). **g** Expression fold change of murine _Il33-002_ (gray) and _Tnf_ (pink), normalized to _Actb_ and water control (mean ± SEM), following treatment of control NIH-3T3 cells with 1 μM 5-aza-2′-deoxycytidine ($n = 3$ biologically independent experiments). **h** _Tnf_ expression (mean ± SEM) in NIH-3T3 cell lines in control (water); gray bars) or 1 μM 5-aza-2′-deoxycytidine (pink bars) stably expressing either gRNAscr or gRNATnf2 under high-puromycin conditions in combination with dCas9, followed by dCas9 removal by Cre recombinase, as assayed by RT-qPCR and normalized to _Actb_ expression ($n = 3$ biologically independent experiments). **i** Schematic of the human _FMR1_ repeat region showing the 5′ untranslated region (UTR) that is prone to CGG repeat expansion and methylation in Fragile X syndrome. Sequence of the gRNA targeting this region is shown (gRNA-CGG) and the extent of the available binding sites for this gRNA is represented by purple lines which indicate binding sites: the 13 presented here represent less than 15% of the available binding site in the Fragile X syndrome patient primary fibroblasts used in this study, which have approximately 700 CGG repeats. **j** _FMR1_ expression quantified by RT-qPCR and normalized to _GAPDH_ expression levels in Fragile X syndrome patient primary fibroblasts that had stably expressed dCas9 (later removed with Cre) and either gRNAscr (gray) or gRNA-CGG (purple) under high-puromycin selection ($n = 6$ biologically independent experiments, mean ± SEM). Data is represented as a percent of the expression of _FMR1_ in wild-type age-matched primary fibroblasts (Mann–Whitney test, two-sided). * indicates statistically significant difference of $P < 0.05$, ** $P < 0.01$, *** $P < 0.001$, **** $P < 0.0001$, and ns not significant (Student's _t_-test, two-sided, with Holm-Sidak correction if number of tests is greater than 3). Exceptionally, for (**j**) Mann–Whitney test was used due to unequal variance. Source data are provided as a Source Data file.

induce expression and that expression necessitates demethylation of a different region either in _cis_, such as the two murine proximal _Tnf_ enhancers (Fig. S9A)[76], or in _trans_ through activation of putative transcription factors.

Our final demethylation target was the _FMR1_ gene which, in patients with Fragile X syndrome, undergoes a CGG repeat expansion (>200 repeats) in its 5′ UTR that becomes aberrantly hypermethylated and results in silencing of _FMR1_ transcription[77]. This region has repeatedly been shown to be reactivated by 5-aza-2′-deoxycytidine[78–81] and we validated it herein (Supplementary Fig. 10A). The CGG repeat expansion is a unique target for a guide RNA with the sequence GGCGGCGGCGGCGGCGGCGG and PAM motif CGG since it should bind sequentially to the entire large repeat region and—under sufficient expression levels—shield the entire region from methyltransferase activity (Fig. 7i). We obtained publicly available primary fibroblasts from a patient with Fragile X syndrome with approximately 700 CGG repeats exhibiting high methylation, as determined previously[82], and a lentiviral vector bearing the CGG-targeting gRNA sequence (gRNA-CGG)[83]. After application of our optimized dCas9-demethylation protocol using gRNA-CGG or gRNAscr (20 μg/mL puromycin) we observed a reduction in the methylated CGG repeat fraction in the gRNA-CGG condition (Supplementary Fig. 10B–D) and significant upregulation of _FMR1_ gene expression ($P = 0.0087$) (Fig. 7j), characterized by an increase from a mean 0.7% of wild-type (control primary fibroblast) expression in gRNAscr cells to a mean of 27% in gRNA-CGG cells and as much as a 110-fold induction in one cell line corresponding to 81% of wild-type _FMR1_ levels. The magnitudes of induction of _FMR1_ gene expression are vastly larger than the induction following TSS demethylation observed in _Il33-002_ and are suggestive of the fact that, in this case, DNA methylation of the repeat region has a large effect on gene expression.

In summary, we demonstrate that the dCas9 demethylation method can be effectively applied in several different cell types: a murine fibroblast cell line, a human breast cancer cell line, and primary patient fibroblasts and across different genetic contexts. This method could be used to assess the relative contribution of DNA methylation in specific sites to modulation of gene expression and to delineate positions whose demethylation would

have the largest effect on expression. Since our method physically targets DNA methylation without confounding enzymatic activities it provides an unconfounded and at times surprising assessment of the role of DNA methylation.

**CRISPR/Cas9-induced demethylation confounds mutational studies with Cas9.** The catalytically active CRISPR/Cas9 system has become the gold standard technique for generating gene knockouts in functional studies. A common technical consideration in these approaches is to target 5′ constitutive exons such that frameshift mutations are more likely to take effect early and render the translated protein nonfunctional[84]. This inevitably results in the positioning of the Cas9:gRNA ribonucleoprotein complex near the TSS and proximal promoter of the targeted gene. Based on the results describe here, we hypothesized that the residence time of DNMT-interfering Cas9, in addition to the drastic epigenetic changes that occur during post-mutagenesis repair[85], may, in certain cell subpopulations, result in DNA demethylation and gene induction that would confound the interpretation of the results.

We had in a previous study used Cas9 and an _HNF4A_-targeting gRNA from the commonly used GECKO gRNA library[84] to generate _HNF4A_ gene knockouts in primary human hepatocytes[86]. The gRNA target site is located in the first exon of several _HNF4A_ isoforms, the _HNF4A_ TSS is only 2 bp from the 3′ end of the PAM, and there are three CpGs directly within the site, with additional CpGs in close proximity (Fig. 8a). We analyzed one mixed _HNF4A_ CRISPR:Cas9 targeted cell population and mapped by Sanger sequencing different _HNF4A_ alleles, which were primarily bearing a T → C missense mutation as well as in-frame and out-of-frame deletions (Fig. 8a–c), indicating that a considerable fraction of cells in this population were likely to produce a protein that retained some degree of functionality. To our surprise, we found that this highly methylated region was completely demethylated in this cell population, irrespective of the mutation induced by Cas9 (Fig. 8d). This demethylation was both substantial and broad, covering not just a 311 bp fragment with 15 CpGs highly methylated in gRNAscr cells to over 90% on average, but also continued to a slightly smaller degree into

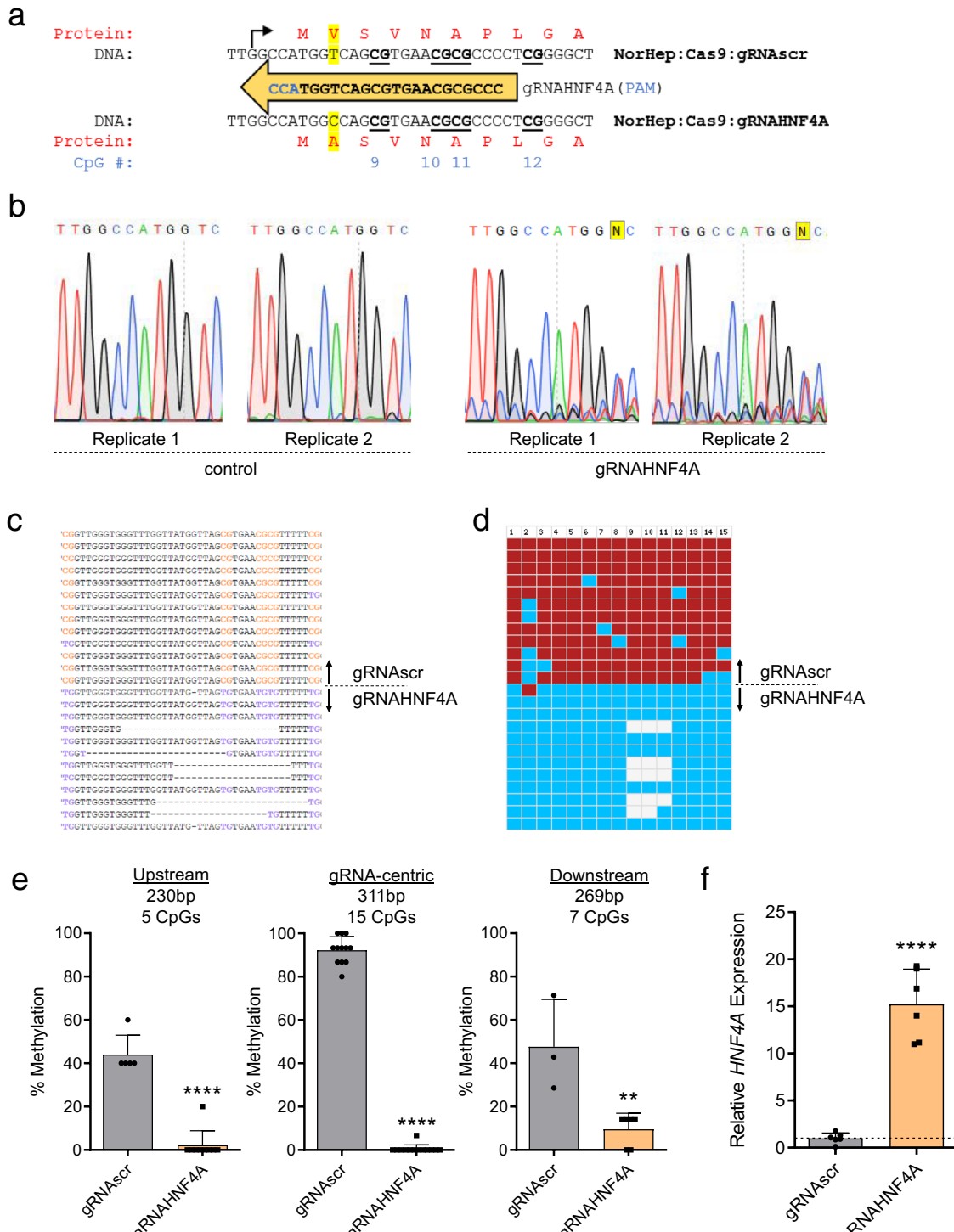

originally less methylated regions immediately upstream (230 bp with five CpGs) and downstream (269 bp with seven CpGs) (Fig. 8e). We also found that this demethylated gHNF4A population expressed approximately 15-fold more *HNF4A* mRNA than gRNAscr controls (Fig. 8f). Thus, standard CRISPR/Cas9 gene depletion studies might be confounded by the effects of extensive demethylation.

## Discussion

The developmental profiles of DNA methylation across human tissues[87] combined with the fact that deviations from these patterns are associated with disease[1,88,89] suggest that DNA methylation has an important role in physiological processes. Importantly, it has been suggested that DNA methylation plays a functional role in the molecular pathology of cancer[1,3,73,89–91] and other common diseases, including mental health disorders[92–94].

Correlation studies since the early 1980s have suggested that DNA methylation in promoters and other transcriptional regulatory regions is negatively correlated with gene expression[95–98]. In the last three decades, several lines of evidence have provided support to the causal role of DNA methylation in the modulation

**Fig. 8 Demethylation is a confound of Cas9 knockout gene deletion. a** The sequence of the lentiviral gRNAHNF4A and its PAM site in blue. Above is the reference sequence of the *HNF4A* gene near the gRNA target site, as validated by Sanger sequencing in primary human hepatocytes expressing via lentivirus Cas9 and gRNAscr, with the TSS indicated by a black arrow and the reference protein sequence in red. CGs are bolded and underlined. Below is the dominant Sanger sequence profile of a primary human hepatocyte population expressing lentiviral Cas9 and gRNAHNF4A. This mutation and the resulting difference in the amino acid sequence, as well as the reference sequences at this location, are highlighted in yellow. **b** Two technical replicates each of the Sanger sequencing chromatograms from the primary human hepatocytes expressing dCas9 and gRNAscr (left) or dCas9 and gRNAHNF4A (right) at the targeted *HNF4A* locus. **c** Sanger sequencing results of 13 gRNAscr and 12 gRNAHNF4A DNA strands following bisulfite conversion from the cell populations in (**b**), demonstrating both the methylation levels and the variety of mutations induced by Cas9 in gHNF4A-treated cells. **d** Same as (**c**) except data expanded is expanded to a larger (>300 bp) region, and simplified such that only CpGs are shown, where blue squares indicate unmethylated CpGs, red squares indicate methylated CpGs, and white squares indicate missing information due to Cas9-induced deletions. CpGs are numbered in accordance with (**a**). **e** Bisulfite-sequencing data from (**d**) (center) as well as five CpGs immediately upstream (left) and seven CpGs immediately downstream (right), displayed as percent DNA methylation over all sequenced DNA strands in primary human hepatocytes expressing Cas9 and either gRNAscr (gray) or gRNAHNF4A (orange) and as mean ± SD as it is summary data from one mutated cell line. Individual dots represent individual strands of DNA from this clonal cell line. **f** *HNF4A* expression in primary human hepatocytes expressing Cas9 and either gRNAscr (gray) or gRNAHNF4A (orange) quantified by RT-qPCR and normalized to *GAPDH* expression, followed by normalization to average expression in gRNAscr cells, with a dashed line at 1 ($n = 6$ independent clones, mean ± SD). * indicates statistically significant difference of $P < 0.05$, **$P < 0.01$, ***$P < 0.001$, ****$P < 0.0001$, and ns not significant (Student's *t*-test, two-sided, with Holm-Sidak correction if number of tests is greater than 3). Source data are provided as a Source Data file.

of gene expression. First, in vitro methylation of reporter plasmids was shown to silence transcriptional activity when these plasmids were transfected into cell lines[97]. Later studies used different methods to limit in vitro methylation to specific regions. Although these studies provide the most direct evidence that there are cellular mechanisms to recognize DNA methylation in particular regions and translate this into silencing of gene activity, the main limitation of these studies is that silencing of ectopically methylated DNA might not reflect on genomic methylated sites and might instead represent a defense mechanism to silence invading viral and retroviral DNA[99] rather than a mechanism for cell-type-specific differential gene expression. Second, DNA methylation inhibitors 5-aza-2′-deoxycytidine and 5-azacytidine provided early evidence for a causal role for DNA methylation in defining cellular identity and cell-type-specific gene expression[100]. However, these inhibitors act on DNA methylation across the genome and do not provide evidence for the causal role of methylation in specific regions or specific genes. Moreover 5-azacytidine was reported to have toxic effects unrelated to DNA methylation[12]. Antisense[13], siRNA[14], and gene knockout[15] depletions of DNA methyltransferases (DNMTs) provided further evidence for the role of DNA methylation in cellular differentiation and development, however DNMT depletion similarly reduces methylation in a general manner, leaving unanswered questions as to the relative role of DNA methylation at specific regions. Furthermore, all DNMTs form complexes with chromatin silencing proteins and might control gene expression by DNA methylation independent mechanisms[90,101–103].

A study examining the state of methylation of TSS regions that are physically engaged in transcription using ChIP-sequencing with antibody against RNAPolII-PS5, the form of RNApolII that is engaged at transcription turn on, showed that promoters that are actively engaged in transcription onset are devoid of methylation[104]. Although these data show that transcription initiation is inconsistent with DNA methylation, the question of causality remains: Is DNA demethylation a cause or effect of transcription onset? Similarly, enhancers are demethylated at transcription factor binding sites; Is demethylation a cause or effect of transcription factor binding[105–107].

To address this longstanding question, CRISPR/Cas9 fusion constructs with TET catalytic domains were generated to target demethylation to specific regions and to determine whether demethylation of particular regions alters transcription activity[19,20,108].

Here, we show that while dCas9-TET induces only modest demethylation of the TSS, it induces robust activation of the *Il33-*

*002* gene (Fig. 1), but the results leave us with unanswered questions on whether DNA demethylation of the basal promoter was causal to this activation. First, TET enzymes are not enzymatically demethylases but monooxygenases which oxidize 5-methylcytosine to 5-hydroxymethylcytosine, 5-formylcytosine, and 5-carboxylcytosine, which have demonstrated stability[109,110], demonstrated differential protein interactors[21–26], and demonstrated structural effects on DNA[27], suggesting that each derivative may be a unique epigenetic mark that confounds conclusions concerning the causality of DNA demethylation events[21–23,25,109,110].

We show here that dCas9-TET causes hydroxymethylation of the *Il33-002* promoter that is maintained in culture (Supplementary Fig. 1I). Moreover, TET proteins are also able to oxidize thymine to 5-hydroxymethyluracil, thereby introducing another confounding epigenetic mark that produces a unique spectrum of modifications on chromatin structure and transcription factor activity[111]. A recent candidate for the improvement of such a strategy is the fusion of dCas9 to the *Arabidopsis* ROS1 glycosylase that directly removes 5-methylcytosine by direct base excision repair, foregoing the intermediate oxidized derivatives with epigenetic potential[112]; yet the issue of the overexpression of an enzyme with a capacity for unwanted and non-targeted effects is not solved by this approach.

Moreover, our data suggest that TET activation of *Il33-002* is independent of DNA demethylation since a dCas9-deadTET mutant with inhibited catalytic monoxygenase activity does not trigger demethylation but also activates *Il33-002* to a similar extent as the catalytically active dCas9-TET (Fig. 1h). We also find that TET1 is capable of inducing unmethylated DNA (Fig. 1j), clearly indicating a demethylation-independent transactivation capacity. It is indeed known that even the restricted catalytic TET domains used in dCas9-TET fusions retain a protein interaction domain that binds O-linked N-acetylglucosamine transferase (OGT)[28,29] and TET proteins and OGT have been shown to co-localize across the genome[113]. The recruited OGT regulates gene expression by glycosylating and modulating the activity of transcription factors such as HCFC1, SP1, OCT4, MYC, p53, and RNA polymerase II as well as histones to directly increase local H2B mono-ubiquitination and trimethylation of histone 3 on lysine 4, both of which are associated with increased gene expression[28,113,114]. This mechanism as well as other potential mechanisms of catalytic-independent transcriptional activation by TET may explain our observation that dCas9-deadTET led to substantial gene induction despite an apparent lack of catalytic activity. The fact that catalytically dead TET

protein activates transcription is consistent with previous reports[115], and confounds the interpretation of the causal role of TET induced demethylation in gene activation.

Third, the fact that an enzyme with such a potential for transcriptional modulation is being overexpressed as a dCas9-TET1 fusion introduces capacity for unwanted transcriptional changes and more recent attempts to use the SunTag system to amplify TET binding at a desired locus[20] only aggravate this issue by overexpression of large numbers of antibody-fused TET1. These undesirable effects would only be negligible in a scenario where a cell expresses a single copy of dCas9-TET that is bound at the intended locus, with highly effective oxidation and base excision repair, an impossible situation given that these lowly-active fusions must be highly expressed to facilitate robust demethylation, and thus inevitably leaving many unbound copies of dCas9-TET free to affect the genome in a TET-dependent—rather than dCas9-dependent—binding manner. Indeed, our data suggest that dCas9-TET demethylates the Il33-002 promoter with a scrambled, non-targeting guide (gRNAscr) (Fig. 1e–g). We also show global genomic hypomethylation in response to dCas9-TET expression and see that a significant fraction of this demethylation localizes to enhancers, a well-established target of TET proteins[59–64]. Though these data represent only a small fraction of dmCpGs caused by dCas9-TET, enhancers are also likely a fraction of TET targets, an issue likely aggravated by both the fact that these experiments involved TET of human origin expressed in mouse cells, and that the database of mouse enhancers does not necessarily reflect those to which TET is recruited in NIH-3T3 cells. This is indicative of a potential ubiquitous and dCas9-independent activity of the fused, over-expressed TET domain in a behavior similar to the demonstrated global methylation by DNMT3A in dCas9-methyltransferase fusions[116].

A second category of confounding off-target effects that are introduced by a targeting strategy that employs a flexibly tethered enzyme, which can modify genetic regions in close physical proximity—despite large genetic distances—particularly those in ubiquitous self-interacting topologically associating domains (TADs), such as the one that the two Il33 promoters belong to[117]. This is aggravated by the fact that the TET family of proteins is known to participate in enhancer regions and facilitate long-range chromatin interactions[118,119]. However, as mentioned above, this may also not be a long-range interaction, but rather a direct interaction of dCas9-TET through the TET domain.

Thus, whenever a flexibly tethered enzyme is employed for epigenetic editing, it will be difficult to dissociate effects of targeted and nontargeted DNA demethylation on transcription activity.

Finally, the demethylation that is observed with dCas9-TET fusions might be secondary to transcription activation. When we combined our three targeting gRNAs with the well-characterized dCas9-VP64 fusion (VP64 is a potent transcriptional activator originating from the herpes simplex virus[34]) we observed broad demethylation of the Il33-002 promoter (Supplementary Fig. 1F–H). This phenomenon suggests that DNA demethylation can in particular instances be secondary to transcription factor recruitment and transcriptional activation (Supplementary Fig. S1I-J) as has been previously reported[105,106].

Lastly, there are examples in which dCas9 or another targeting protein either bears a catalytically inactive form of TET or the domain is altogether missing, and mild demethylation is still observed[19,20,30,120]. We propose that, in some cases, this demethylation stems from the lingering transactivation capacity of the mutated TET domain (discussed above) followed by demethylation as a consequence of activation, such as the demethylation caused by VP64 activation (Supplementary Fig. S1F–H). Alternatively, as we demonstrate here (Fig. 2) binding of dCas9 blocks

DNA methyltransferase catalyzed methylation. This therefore obscures the true contribution of TET proteins to demethylation. It is in fact possible that most of the demethylation triggered by dCas9-TET fusions seen in dividing cells stems from the simple steric interference with DNA methyltransferase activity, as we demonstrate in this study.

Taken together, these data reveal that while dCas9-TET may be a valid tool for producing epigenetic perturbations that may further understanding of TET dynamics, it introduces a number of confounds inherent to the properties of the TET protein that prohibit conclusions as to the causal relationship of changes in DNA methylation at particular sites and gene expression.

We instead propose and demonstrate here a previously unrecognized capacity of dCas9 to prevent DNA methylation with high efficacy at fairly small, precise regions and, more importantly, free from any fused eukaryotic enzyme that may act independently of the dCas9:gRNA binding activity. We first show that this approach can be implemented to map the individual methylated CpGs within a regulatory region, which silence transcription using an in vitro methylation promoter-reporter transient-transfection assay. This method has advantage over earlier methods that protected individual CpGs from methylation by mutagenesis to non-CpG sequences[107], since mutagenesis can disrupt protein:DNA interactions by the sequence change rather than by the methylation difference[121]. Our method alters the methylation per se without disrupting the genetic sequence. Our results demonstrate that three CpG sites within 22 bp of the TSS are sufficient to silence the Il33-002 promoter, while other CpG sites do not contribute to methylation-dependent silencing of promoter activity.

We further show that this approach can be applied to trigger site-specific demethylation in dividing cells, and that it can be optimized for near-complete removal of DNA methylation from sites that had previously been fully methylated, without perturbing the methylation states of adjacent CpGs in the same promoter to any substantial degree. Thus, this method could interrogate the causal role of DNA methylation in silencing gene expression. Since inhibition of DNA methylation is dependent on tight binding of dCas9 which is also dependent on gRNA target and quality, the risk for nontargeted demethylation is low. Accordingly, we find that there appear to be no off-target DNA demethylation events as a consequence of gRNA:dCas9 off-target binding in WGBS/ChIP-seq data and in targeted sequencing of five candidate off-target regions. However, further work is needed to identify the biological origin of the dmCpGs that are not a consequence of dCas9:gRNA off-target binding that were detected in WGBS analysis. Potential off-target effects of a larger number of gRNAs across multiple cell lines and species need to be evaluated as well.

We used our method of demethylation to define the role of TSS and proximal promoter methylation of the Il33-002 gene in its cogent genomic context. We found that demethylation of the Il33-002 TSS produces a small but significant increase in its expression. Our results confirm what was observed in the transient transfection assay: CpG sites 9–11 at the TSS suppress promoter activity. However, dCas9-TET induced 25-fold higher Il33 expression compared to dCas9 alone when targeted to the same promoter, even though it caused significantly lower demethylation than dCas9 (Fig. 5h). There are several possible explanations for this discrepancy between the fold induction achieved by demethylation and by TET recruitment. First, the fusion of TET to dCas9 is flexible and may allow access to DNA in a wider region, perhaps inducing demethylation in other regulatory regions or methylated transcription factors that are required for more robust expression (Fig. 1e–g). However, treating cells that have been demethylated at the Il33-002 TSS CpG sites 9–11 with

5-2′ deoxy-azacytidine doesn't further induce the gene, while cells that were methylated at 9–11 sites are induced to a level like the levels achieved by dCas9. This suggests that the main regulation by DNA methylation occurs at CpGs 9–11 but that the gene is further induced by DNA methylation independent mechanisms that are partially triggered by TET. This illustrates that the results of TET targeting could not be automatically understood as being driven by demethylation and highlights the need for enzyme independent targeted DNA demethylation for understanding the role of DNA methylation.

We then determined whether demethylation of the TSS poises the *Il33-002* promoter for induction by known inducers of this gene. poly(I:C) induces *Il33-002* approximately 300-fold without detectable DNA demethylation, and does not induce the demethylated *Il33-002* to an appreciably higher level than the methylated *Il33-002* promoter. Thus, induction of *Il33-002* expression is independent of DNA demethylation in the basal promoter. It is possible however that poly(I:C) triggers demethylation in a remote enhancer that wasn't examined in our study. In contrast, induction by LPS is higher when the basal promoter is demethylated, however LPS induces the promoter whether it is methylated or not, suggesting an additive but nonessential effect of demethylation of TSS for LPS induction. What is the role of *Il33-002* promoter methylation? The data is consistent with the idea that this gene is mainly regulated by extracellular signals irrespective of DNA methylation. DNA methylation only silences the residual basal activity of the promoter, perhaps to prevent leaky expression and transcriptional noise in the absence of the appropriate signal. This is consistent with the observation that the ectopically transfected promoter is silenced by DNA methylation (Fig. 2). Therefore, both targeted demethylation or 5-aza-2′-deoxycytidine achieve only a small elevation in expression.

A different paradigm is represented by the *SERPINB5* promoter. Demethylation of the basal promoter on its own has no effect on expression, which remains low (Fig. 7e, F) even when six CpGs in the proximal promoter region become completely demethylated (Fig. 7b–e). However, global demethylation by 5-aza-2′-deoxycytidine induces the activity of this demethylated promoter further than the naturally methylated promoter in control MDA-MB-231 cells, suggesting that expression of this gene is regulated by methylation in the promoter region as well as other regions in *cis* or *trans*. Demethylation of the proximal promoter on its own is insufficient to induce transcription. A possible explanation is that activity of this gene requires activation of a transcription factor that is silenced in these cells and induced by demethylation as we have recently shown[122]. The tumor necrosis factor (*Tnf*) gene exhibits a proximal TSS promoter region that is highly methylated in NIH-3T3 cells. The gene is highly induced and its proximal promoter region is demethylated by 5-aza-2′-deoxycytidine (Fig. 7g). In contrast to the large induction of expression of by 5-aza-2′-deoxycytidine, demethylation of ten CGs proximal to the TSS (Fig. S9) using the targeted dCas9 method did not turn on the gene (Fig. 7h, gray bars). Here, as was the case with the *SERPINB5* promoter, 5-aza-2′-deoxycytidine treatment of cells bearing dCas9-demethylated *Tnf* TSS region resulted in higher induction of expression than treated control cells bearing a methylated *Tnf* TSS (Fig. 7h). These experiments illustrate the importance of studying demethylation of specific sites per se to truly understand their contribution to gene expression control.

Finally, in a manner dissimilar to the other genes examined in this study, targeted demethylation of the large, highly-methylated *FMR1* repeat region in Fragile X syndrome patient fibroblasts did induce basal transcription of the *FMR1* gene up to a 110-fold in one cell pool suggesting that, in this case, methylation of the repeat element plays a large role in silencing of the gene.

As the larger magnitude of demethylation observed in the dCas9 approach does not produce transcriptional changes as substantial as those observed by dCas9 tethered to TET1, it is clear that promiscuous mammalian enzymatic domains do not exclusively demethylate, have other methylation independent activities, and cannot be suitably applied to investigate the causal relationship between DNA methylation at specific sites and gene expression.

The ability of newly demethylated sites to stay demethylated can vary; we detected no increase in *Il33-002* TSS methylation 75 days after removing dCas9 by Cre-mediated recombination in NIH-3T3 cells, but saw a small non-significant increase in SER-PINB5 demethylated CpGs 45 days after Cre-mediated dCas9 removal. It is useful for research timescales that sites stay demethylated, but the small variation between the two genes in the two cell lines suggests that the retention of unmethylated CpGs may vary as a factor of cell line (e.g., how much de novo methyltransferase activity a cell line has) or by specific CpG sites (e.g., in a growing cell population, how detrimental to cell growth is demethylation of a specific CpG and will it be selected against?), and thus it will be important in future studies that use this technique to assay how long demethylation persists in the CpG and cell line contexts under examination to ensure that demethylation persists for the duration of the experiments. It is important to note that in stem cells where de novo DNMTs are expressed to a higher level[123], methylation might be regained after removal of dCas9.

In summary, we developed a tool that allows site-specific demethylation of a narrow region of DNA by physical blocking of DNMTs without using confounding epigenetic enzymatic activities. This tool enables the examination of causal relationships between demethylation of specific sites and gene expression in genes at their native positions in the chromatin. Comparing the results obtained using this tool and results obtained using general DNA methylation inhibitors reveals that the role of DNA demethylation at specific sites might have been previously overestimated by confounded techniques and thus is part of a growing body of evidence in support of this notion[10,124]. Our study demonstrates the need for the careful causational investigation of the role of DNA demethylation of different regions per se by an unconfounded tool. We hope that this tool can be used to attribute causality to DNA methylation changes not only in fundamental physiological gene transcription, but also under different specific physiological and pathological conditions mediated by changes in extracellular signals and changes in the milieu of cellular transcription factors in order to begin to reveal the true extent, nature, and diverse contribution of DNA methylation at different regions to gene regulation.

## Methods

**gRNA design and synthesis.** To maximize likelihood of on-target efficiency and minimize off-target binding, gRNAs were designed using three online tools with distinct scoring algorithms: Off-Spotter, CCTOP, and CRISPR Design[43,52,53]. Final gRNAs were chosen based on highest cumulative rank and location in the promoter. The scrambled gRNA sequence was obtained from pCas-Scramble (Origene). For in vitro assays, gRNAs were in vitro transcribed with the GeneArt™ Precision gRNA Synthesis Kit (Thermo Fisher Scientific) according to manufacturer protocol and using primers listed in Supplementary Table 2. Due to a lack of available kit compatible with *S. aureus* gRNAs, SA-gRNA1–4, and SP-gRNA1–4 were generated by a custom T7 in vitro transcription protocol (https://doi.org/10.17504/protocols.io.dwr7d5) modified to replace the *S. pyogenes* scaffold sequence with that of *S. aureus.* (primers in Supplementary Table 3). Lentiviral gRNAs were first produced according to the protocol by Prashant Mali[125]. Briefly, 455 bp double stranded DNAs containing the human U6 promoter, gRNA sequence, gRNA scaffold, and termination signal were ordered as gBlock Gene Fragments (IDT). These were re-suspended, amplified with Taq Polymerase (Thermo Fisher Scientific) according to manufacturer protocol and using primers listed in Supplementary Table 2, extracted from an agarose gel with the QIAEX II Gel Extraction Kit (QIAGEN), and inserted into pCR®2.1-TOPO (Thermo Fisher

Scientific) by incubating for 30 min at room temperature. The gRNA scaffold was now flanked by EcoRI sites from the vector. A lentiviral backbone was obtained from Addgene (pLenti-puro, Addgene #39481) and the CMV promoter was removed to prevent aberrant transcription by digesting the plasmid with ClaI-HF and BamHI-HF (NEB), gel extracting, removing DNA overhangs with the Quick Blunting™ Kit (NEB), and circularization with T4 Ligase (Thermo Fisher Scientific) for 1 h at 22 °C. The resulting promoterless pLenti-puro plasmid was then digested with EcoRI and the 5′ phosphates were removed with Calf Intestinal Alkaline Phosphatase (Thermo Fisher Scientific) to facilitate efficient ligation of the EcoRI-flanked gRNA scaffold. Resulting clones were Sanger sequenced with pBABE 3′ sequencing primer to ensure proper gRNA sequence and orientation (Génome Québec). The gRNAs targeting SERPINB5 and Tnf were created by site-directed mutagenesis of pLenti-Il33gRNA6-puro using primers listed in Supplementary Table 2, and the Q5® Site-Directed Mutagenesis Kit (NEB) according to manufacturer protocol. The HNF4A-targeting gRNA is from the genome-scale CRISPR knock-out (GeCKO) v2 library[126] (purchased as lentiviral plasmid from Genscript) and the FMR1-targeting gRNA from the Jaenisch lab was obtained from Addgene (pgRNA-CGG, Addgene #108248).

**Site-specific in vitro DNA methylation**. First, a dCas9:gRNA ribonucleoprotein complex was formed with the following mixture: 14 μL nuclease-free water, 3 μL Cas9 Reaction Buffer (Applied Biological Materials Inc.), 7.5 μL 300 nM CpG-targeting in vitro transcribed gRNA or non-CpG-targeting control gRNA, and dCas9 recombinant protein (Applied Biological Materials Inc.). After 10 min at room temperature, 3 μL of 30 nM Il33-pCpGl was added to the reaction, which was then transferred to 37 °C to allow dCas9:gRNA complex binding to DNA. After 1 h, the following mixture was added to the reaction: 145 μL nuclease-free water, 17 μL NEBuffer™ 2, 5 μL 32mM S-Adenosyl methionine (NEB) (final concentration 0.8 mM), and 3 μL (12 units) M.SssI methyltransferase (NEB). This solution was pre-warmed to 37 °C before addition to prevent interference with dCas9:gRNA binding to the DNA. After 4 h of incubation at 37 °C, 1 μL of 20 mg/mL Proteinase K (Roche) was added and the temperature was raised to 64 °C for an additional 4 h.

**DNA Isolation, bisulfite conversion, bisulfite-cloning, and pyrosequencing**. Plasmid DNA was recovered by phenol-chloroform extraction and precipitation in ethanol overnight. DNA was washed one time with 70% ethanol, dried, and re-suspended in 30 μL nuclease-free water. Genomic DNA was extracted from cells by resuspension in 400 μL DNA lysis buffer (100 mM Tris, pH 7.5, 150 mM NaCl, 0.5% SDS, 10 mM EDTA) and treatment with 2 μL 20 mg/mL RNAse A (NEB) for 30 min at 37 °C and 5 μL 20 mg/mL Proteinase K (Sigma) for 4 h at 55 °C. This was followed by phenol-chloroform extraction by addition of 200 μL phenol solution and 200 μL of chloroform, vortexing for 10 s, and centrifugation at $16,000 \times g$ for 5 min at 4 °C. The aqueous phase was then transferred to a new 1.5 mL tube, mixed with 400 μL chloroform, and centrifuged again at $16,000 \times g$ for 5 min at 4 °C. The aqueous phase was again transferred to a new tube and DNA was precipitated by the addition of 1 mL 95% ethanol and 1 μL glycogen overnight at −80 °C, centrifugation at $16,000 \times g$ for 30 min at 4 °C. DNA was washed a single time with 1 mL 70% ethanol, centrifuged at $16,000 \times g$ for 15 min at 4 °C, air dried for 5 min, and resuspended in 50 μL nuclease-free water. Following DNA extraction, bisulfite conversion was conducted according to manufacturer protocol with the EZ DNA Methylation-Gold Kit (Zymo Research) using 5 μL of in vitro methylated plasmid DNA or 1.5 μg genomic DNA measured with the Qubit dsDNA BR Assay (Thermo Fisher Scientific). One microliter of bisulfite-converted DNA was amplified with HotStar Taq DNA polymerase (QIAGEN) in a 25 μL reaction using the primers designed with MethPrimer[127] and listed in Supplementary Table 2. Pyrosequencing samples were processed in the PyroMark Q24 instrument according to protocols designed by the PyroMark Q24 software (QIAGEN). Sequencing primers were designed with Primer3[128]. Alternatively, amplicons were cloned into pCR®4-TOPO (Thermo Fisher Scientific) for 30 min at room temperature and transformed into TOP10 competent cells (Thermo Fisher Scientific) prior to plasmid isolation with the High-Speed Plasmid Mini Kit (Geneaid) and Sanger sequencing (Eurofins Genomics) using the M13R sequencing primer. All oligonucleotides used in this study were purchased from Integrated DNA Technologies.

**Luciferase assay**. $8.0 \times 10^4$ NIH-3T3 cells (Il33 experiments) or $1.2 \times 10^5$ HEK293 cells (TET co-transfection) were plated in a 6-well plate (Corning) 24 h prior to transfection. One microgram (Il33) or 100 ng (SV40) plasmid DNA from the in vitro methylation reactions were transfected with 3 μL (Il33) or 1 μL (SV40) X-tremeGENE 9 transfection reagent (Roche) diluted in 50 μL of Opti-MEM medium (Gibco). Luciferase assays were performed 36 h after transfection using the Luciferase Reporter Gene Assay, high sensitivity (Roche). Briefly, cells were washed with 1 mL of phosphate-buffered saline (Wisent), detached with scrapers (Thermo Fisher Scientific) after the addition of 150 μL lysis buffer, and transferred to 1.5 mL tubes. After a 15-min incubation at room temperature, the mixtures were centrifuged for 5 s at maximum speed and the supernatant transferred to new 1.5 mL tubes. Two 50 μL volumes per condition were supplemented with 50 μL luciferase assay reagent in disposable glass tubes (Thermo Fisher Scientific), and light emission was measured immediately in the Monolight 3010 luminometer (Analytical Luminescence Laboratory). Sample protein concentration was determined

by Bradford Protein Assay (Bio-Rad) and A595 readings were measured in a DU 730 UV–Vis Spectrophotometer (Beckman Coulter). Protein concentration in cell lysate was determined by comparing to a bovine serum albumin standard curve and luciferase activity was normalized to concentration. We validated that our transfection method results in equal copy numbers transfected for both methylated and unmethylated DNA by measuring copy number of transfected pCpGl 36 h after transfection (Supplementary Fig. 1N).

**Plasmids**. The original dCas9 plasmid lacking loxP sites was obtained as a dCas9-VP64 fusion (lenti dCAS-VP64_Blast, Addgene #61425). The VP64 domain was removed by digestion with BamHI-HF and BsrGI-HF, blunting with the Quick Blunting™ Kit (NEB), and circularization with T4 Ligase (Thermo Fisher Scientific) for 1 h at 22 °C. Following transformation, plasmids were isolated from ampicillin-resistant clones (High-Speed Plasmid Mini Kit, Geneaid) and Sanger sequenced to identify plasmids that maintained the blasticidin resistance gene in-frame with dCas9. Floxed dCas9 was purchased as a ready plasmid (pLV hUbC-dCas9-T2A-GFP, Addgene #53191) and primers were designed to amplify a fragment of approximately 500 base pairs when dCas9 is removed with Cre recombinase (Supplementary Table 2). The Cre-containing plasmid was obtained from Addgene (pLM-CMV-R-Cre, Addgene #27546). A fragment encoding the CMV promoter and mCherry-T2A-Cre-WPRE was excised by NdeI and SacII (Thermo Fisher Scientific) and transferred to the pLenti6/V5-DEST™ Gateway™ Vector (Thermo Fisher Scientific) bearing a blasticidin resistance cassette (Thermo Fisher Scientific) to facilitate antibiotic selection. Lentiviral Fuw-dCas9-Tet1CD-P2A-BFP and Fuw-dCas9-dead Tet1CD-P2A-BFP were obtained from Addgene (Addgene #108245, #108246). Catalytically active Cas9 lentiviral vector was obtained from Genscript as pLentiCas9-Blast. TET1 plasmids were obtained from Addgene: #49792 (FH-TET1-pEF) and #124081 (pEF1a_FL MUT TET1) and control pEF1A was purchased from Thermo Fisher Scientific. pcDNA3-TET2 (Fig S1J) was generated by amplification of TET2 from human cDNA, TOPO-TA cloning and sequence validation by Sanger sequencing, followed by digestion and ligation into pcDNA3.1 (Thermo Fisher Scientific) using the restriction enzymes XhoI and ApaI. SV40-pCpGl (Fig. 1j) was generated by amplification of the SV40 promoter and enhancer region from lenti dCAS-VP64_Blast using primers that added a 5′ BamHI site and a 3′ HindIII site, which were then used for transfer into pCpGl[37] following sequence verification.

**Cell culture**. HEK293T and NIH-3T3 cells (ATCC) were thawed and maintained in DMEM medium (Gibco) supplemented with 10% Premium Fetal Bovine Serum (Wisent) and 1× Penicillin-Streptomycin-Glutamine (Gibco). Cells were grown in a humidified incubator of 5% carbon dioxide at 37 °C and cultured in 100 mm × 20 mm tissue culture dishes (Corning) and harvested or passaged by trypsinization (Gibco) upon reaching 80–90% confluency. Clones were isolated by limiting dilution and trypsinization with the aid of cloning rings. Fragile X syndrome fibroblasts (GM05848, Coriell Institute) and age-matched control fibroblasts (GM00357, Coriell Institute) were maintained as above. Flow cytometry to isolate dCas9-TET/dCas9-deadTET (BFP) and dCas9 (GFP) when antibiotic selection was not an option was performed by Julien Leconte of the Flow Cytometry Core Facility at McGill University Life Sciences complex. All replicates presented in this study are biological replicates. A technical replicate is performed for each assay and averaged per each biological replicate.

**Lentiviral production**. HEK293T cells were plated at a density of $3.8 \times 10^6$ per 100 mm dish 24 h prior to transfection. Cells were transfected using X-tremeGENE 9 transfection reagent (Roche). Briefly, individual lentiviral transfer plasmids were mixed with a packaging plasmid (pMDLg/pRRE, Addgene #12251), envelope protein plasmid (pMD2.G, Addgene #12259), REV-expressing plasmid (pRSV-Rev, Addgene #12253), and the transfection reagent in Opti-MEM medium (Gibco). The mixture was incubated for 30 min at room temperature and added in a drop-wise manner to HEK293T cells in 8 mL of fresh DMEM medium in a 100 mm dish. Lentiviral particles were harvested by filtering the supernatant through a 0.45 μm disk filter 72 h after transfection and either used immediately or stored at −80 °C. 5 μg/mL Blasticidin S HCl and 1–20 μg/mL Puromycin Dihydrochloride (Gibco) were used to select for stable transformants.

**Transient transfection for time-course experiments**. $8.0 \times 10^4$ NIH-3T3 cells stably expressing dCas9-VP64 (from lentiviral transfer and blasticidin selection, above) were plated in a 6-well plate (Corning) 24 h prior to transfection. One microgram per well pLenti-IL33_gRNA2 vector was transfected using X-tremeGENE 9 transfection reagent (Roche) and cells were harvested at 0, 24, 48, 72, and 96 h. RNA and DNA were extracted from separate wells and RNA expression and DNA methylation were measured as described in the relevant methodology sections.

**RT-qPCR**. RNA was isolated from approximately 80% confluent 100 mm dishes with 1 mL of Trizol reagent (Thermo Fisher Scientific) following harvest by tryp-sinization and washing with phosphate-buffered saline (Wisent). RNA extraction was performed according to Trizol manufacturer protocol. Briefly, 200 mL of chloroform was added to 1 mL of Trizol-RNA mixture. The samples were

thoroughly vortexed, incubated at room temperature for 2 min, and centrifuged for 15 min at 12,000×$g$ at 4 °C. The aqueous phase was transferred to a new 1.5 mL tube prior to the addition of 0.5 mL isopropanol and incubation at room temperature for 10 min. The samples were centrifuged for 10 min at 12,000×$g$ at 4 °C, and washed twice with 75% ethanol, discarding the supernatant each time. The pellets were air dried for 10 min and re-suspended in 50 μL DEPC-treated water (Ambion). Concentrations were measured with the Qubit RNA BR Assay (Thermo Fisher Scientific) and 1 μg RNA was used for each reverse transcriptase reaction using M-MuLV Reverse Transcriptase (NEB) according to manufacturer protocol. cDNA was diluted 1:2 (20 μL reverse transcription reaction to 40 μL water) and 2 μL of diluted cDNA was amplified in the LightCycler ® 480 Instrument II (Roche) in a 20 μL reaction containing 10 μL LightCycler ® 480 SYBR Green I Master Mix (Roche) and 0.8 μL each of 10 μM forward and reverse primer listed in Supplementary Table 2. Quantification was performed by Roche Lightcycler Software.

**Drug treatment**. 5-aza-2′-deoxycytidine (Sigma A3656) was dissolved to 10 mM in sterile water and frozen in one-time-use aliquots at −80 °C. Trichostatin A (TSA, Sigma T8552) was dissolved to 1 mM in dimethyl sulfoxide (DMSO, Sigma D8418) and frozen in one-time-use aliquots at −80 °C. Lipopolysaccharides from Escherichia coli O55:B5 (Sigma L6529) were diluted to 1 mM in phosphate-buffered saline. 5-aza-2′-deoxycytidine and TSA treatment regimen involved three treatments every other day with media replacement (5 days total) at specified concentrations and sample collection on the sixth day.

**Off-target prediction for pyrosequencing**. Potential off-target sites of $Il33$ gRNA3 in the mouse genome were predicted using Cas-OFFinder[54], a program that allows bulges in the RNA and DNA (which Cas9 is known to tolerate) to increase the number of possible off-target sites. Because we were interested in changes in methylation, results were filtered for the presence of a CG at a maximum of 10 bp from either end of the gRNA sequence. Of 15 results, two differed by three mismatches, nine by four mismatches, and four by two mismatches and a bulge. We developed functional pyrosequencing assays for four of these sites.

**Hydroxymethylation quantification**. DNA isolated from cells by phenol:chloroform isolation and ethanol precipitation was cleaned on Micro Bio-Spin P-6 SSC columns (Bio-Rad) according to manufacturer protocol. Fifteen millimolar of KRuO$_4$ (Sigma) was prepared by dissolving 0.153 g in 50 mL of 0.05 M NaOH and thawed freshly for each oxidation reaction. One microgram cleaned DNA was incubated in a 19 μL volume reaction in a PCR tube with 0.95 μL 1 M NaOH at 37 °C in a shaking incubator for 0.5 h. The sample was cooled immediately in an ice-water bath for 5 min prior to the addition of 1 μL ice-cold 15 mM KRuO$_4$ and incubation in an ice-water bath for 1 h with vortexing every 20 min. A second oxidation was performed by the addition of 4 μL 0.05 M NaOH, incubation at 37 °C in a shaking incubator for 0.5 h, following by cooling, addition of 1 μL ice-cold 15 mM KRuO4, and incubation in ice-water bath with occasional vortexing as before. Oxidized DNA was cleaned again on Micro Bio-Spin P-6 SSC columns and the DNA was subjected to bisulfite conversion and pyrosequencing. Control reactions were done in parallel in which 15 mM KRuO4 was replaced by 0.05 M NaOH and percent hydroxymethylation was quantified as the decrease in methylated fraction in oxidized DNA as compared to control DNA.

**Chromatin immunoprecipitation**. Hundred and fifty millimeter tissue culture dishes containing 90% confluent NIH-3T3 cells from each experimental condition were cross-linked by the direct addition of formaldehyde to a 1% final concentration. The dishes were incubated for 10 min at room temperature with constant agitation. The reaction was quenched by the addition of glycine to a final concentration of 0.125 M and incubated for an additional 5 min at room temperature with constant agitation. Cross-linking solution was aspirated and cross-linked cells were washed three times with 10 mL ice-cold phosphate-buffered saline (PBS). Ten milliliter of ice-cold PBS was added and cells were scraped into suspension by a rubber cell scraper. Cross-linked cells were pelleted at 800 × $g$ at 4 °C in 15 mL falcon tubes, the supernatant removed, and the cells were lysed in 300 μL ice-cold lysis buffer (1% SDS, 10 mM EDTA, 50 mM Tris), pipeted thoroughly, incubated for 15 min on ice, and immediately sonicated on the Bioruptor (Diagenode) in 1.5 mL Eppendorf tubes at high power for three 10-min cycles of 30 s on and 30 s off, replacing warmed water with ice-cold water and minimal ice between each cycle. Sonicated samples were centrifuged at 4 °C for 16,000 × $g$ for 5 min, and supernatant was transferred to a clean 1.5 mL Eppendorf tube, with 30 μL set aside for shearing efficiency analysis. The remaining supernatant was diluted with a 9× volume of dilution buffer (16.7 mM TrisHcl pH 8.0, 1.2 mM EDTA, 167 mM NaCl, 0.01% SDS, 1.1% Triton X-100) and precleared with washed Dynabeads Protein G (Thermo Fisher) for 2 h at 4 °C on a nutator. Using a magnetic rack, 1% of precleared chromatin was set aside for input and 5 μg Monoclonal ANTI-FLAG® M2 antibody (Sigma, F1804) (to capture 5′ 3×FLAG-tagged dCas9) or IgG (abcam) was added to the remaining volume and then incubated at 4 °C on a nutator overnight. Sixty microliter of washed (3× with Tris-EDTA—10 mM Tris pH = 8, 1 mM EDTA— and 3× with RIPA—20 mM Tris, 2 mM EDTA, 150 mM NaCl, 1% Triton X, 0.1% SDS, 0.5% deoxycholate) Dynabeads were added to each sample and incubated at 4 °C on a nutator for 4 h. Beads were then washed with 1 mL each as follows: 2× with low salt wash buffer (0.1% SDS, 1% Triton-X, 2 mM EDTA, 20 mM Tris, 150 mM NaCl), 2× with high salt wash buffer (same as low except 500 mM NaCl), 2× with LiCl wash

buffer (0.25 M LiCl, 1% NP-40, 1% deoxycholate, 1 mM EDTA, 10 mM Tris, pH 8.0), and 2× with Tris-EDTA. All buffers contained 1× cOmplete™ Protease Inhibitor Cocktail (Sigma). DNA was eluted by the addition of 100 μL elution buffer (1% SDS, 0.1 M NaHCO$_3$), vortexing vigorously, and 15-min incubation at room temperature with constant agitation before transferring to a clean 1.5 mL tube. This was repeated twice for a final volume of 200 μL and the input fraction was adjusted to the same volume with elution buffer. Reverse cross-linking (0.2 M final concentration of NaCl, 65 °C overnight) was performed for all samples, followed by standard treatment with RNAse A, proteinase K, and phenol:chloroform cleanup followed by ethanol precipitation. Clean DNA was then quantified by qPCR and enrichment in the immunoprecipitated samples was calculated as fraction of input. Nonspecific (IgG) antibody and qPCR primers of unbound regions were used as controls for effective immunoprecipitation.

**Chromatin immunoprecipitation sequencing and analysis**. For ChIP-seq experiments, cells were prepared as for ChIP and IP was performed with the same anti-FLAG antibody (above) on NIH-3T3 expressing FLAG-tagged dCas9-GFP (selected by FACS) and gRNA (selected under high puromycin); these are the same cells depicted in Fig. 5 (transduced with empty vector instead of Cre recombinase). All cross-linking and immunoprecipitation steps were performed with the ChIP-IT High Sensitivity® Kit (Active Motif) according to manufacturer's instructions using 30 μg input chromatin as quantified by NanoDrop. Sonication was performed as above. Successful ChIP with anti-FLAG antibody was validated by qPCR (as described above) with primers for $Il33$ (positive control) and $Actb$ (negative control) (Supplementary Fig. 11). Eluted DNAs were sent to Center d'expertize et de services Génome Québec at McGill University for library preparation and sequencing. Fragmented DNA from 12 samples (three replicates each of gRNAscr anti-FLAG, gRNA3 anti-FLAG, gRNAscr input, and gRNA3 input) was quantified using 2100 Bioanalyzer (Agilent Technologies). Libraries were generated robotically with fragmented DNA (range 100–300 bp) using the NEBNext Ultra II DNA Library Prep Kit for Illumina (New England BioLabs), as per the manufacturer's recommendations. Adapters and PCR primers were purchased from Integrated DNA Technologies (IDT). Size selection was carried out using SparQ beads (Qiagen) prior to PCR amplification (12 cycles). Libraries were quantified using the Kapa Illumina GA with Revised Primers-SYBR Fast Universal kit (Kapa Biosystems). Average size fragment was determined using a LabChip GX (PerkinElmer) instrument. The libraries were normalized and pooled and then denatured in 0.05 N NaOH and neutralized using HT1 buffer. The pool was loaded at 200pM on a Illumina NovaSeq S4 lane using Xp protocol as per the manufacturer's recommendations. The run was performed for 2 × 100 cycles (paired-end mode). A phiX library was used as a control and mixed with libraries at 1% level. Base calling was performed with RTA v3.4.4. Program bcl2fastq2 v2.20 was then used to demultiplex samples and generate fastq reads. Paired-end FastQ files were trimmed for adapters and quality scores using TrimGalore v0.6.4_dev[129] under default settings. Alignments to the mm10 genome were performed using bowtie2 v2.3.4.1[130] under default settings and peak calling for each sample was performed with the macs2 v2.2.7.1[131] callpeak function (--g mm --nomodel --extsize 204 --SPMR) after first running the predictd script and establishing --extsize 204 according to the macs2 manual. Alignments were passed to the DiffBind R package to identify significantly differentially enriched regions under default parameters.

**Western blot**. Control NIH-3T3 cells and cells expressing dCas9-TET or dCas9-deadTET were grown to 80% confluency on 100 mm tissue culture dishes. Cells were washed twice with 10 mL PBS and collected into 15 mL falcon tubes by scraping and then pelleted by centrifugation for 5 min at 300×$g$ at 4 °C. The supernatant was aspirated and cells were resuspended in 80 μL protein extraction buffer (150 mM NaCl, 0.1% SDS, 0.5% sodium deoxycholate, 50 mM Tris pH 7.5, and 1% NP-40) with 1× cOmplete™ Protease Inhibitor Cocktail (Sigma), incubated for 30 min on ice with vortexing every 5 min, centrifuged for 10 min at 16,000 × $g$ at 4 °C. The supernatant was retained and protein concentration was measured by Bradford assay. Twenty microgram protein in 2× Laemmli Sample Buffer (Bio-Rad) was prepared according to manufacturer protocol and loaded into a 5% acrylamide gel (for dCas9-TET/deadTET) or 10% acrylamide gel (for beta-actin loading control) with 5% upper stacking gel. Gels were run for 10 min at 110 V and then for 50 min at 170 V, followed by overnight transfer to nitrocellulose membrane at 30 V. Membranes were blocked with 1% milk in TBST and protein was detected with either mouse Anti-CRISPR-Cas9 primary antibody [7A9-3A3] (Abcam, ab191468) (1/2000 dilution) or monoclonal Anti-β-Actin primary antibody produced in mouse (Sigma, A2228) (1/5000 dilution) and goat Anti-Mouse IgG H&L (HRP) secondary antibody (Abcam, ab205719) (1/10,000 dilution). Each antibody incubation was performed for 1 h. After addition of Clarity Western ECL Substrate (BioRad), images were acquired with automatic exposure on the Amersham Imager 600.

**Whole-genome bisulfite sequencing (WGBS)**. WGBS was performed by the Center d'expertize et de services Génome Québec at McGill University. Genomic DNA was quantified using the Quant-iT™ PicoGreen® dsDNA Assay Kit (Life Technologies). 2 × 151 bp paired-end libraries were generated using the NEBNext® Enzymatic Methyl-seq Kit (New England BioLabs, NEB). Adapters were purchased from NEB. Libraries were quantified using the Kapa Illumina GA with Revised Primers-SYBR Fast Universal kit (Kapa Biosystems) and average size fragment was determined

using a LabChip GX (PerkinElmer) instrument. The libraries were normalized and pooled and then denatured in 0.05 N NaOH and neutralized using HT1 buffer. The pool was loaded at 225 pM on an Illumina NovaSeq S4 lane using Xp protocol as per the manufacturer's recommendations. The run was performed for $2 \times 100$ cycles (paired-end mode). A phiX library was used as a control and mixed with libraries at 5% level. Base calling was performed with RTA v3.4.4. Program bcl2fastq2 v2.20 was then used to demultiplex samples and generate FastQ reads.

**WGBS data analysis**. Paired-end FastQ files were trimmed for adapters and quality scores using TrimGalore v0.6.4_dev[129] under default settings. Alignments to the mouse mm10 genome, deduplication, and methylation calling were performed using Bismark v0.22.3[132] under default settings. All statistical analyses were performed with the R package methylKit v1.14.2[133]. For off-target analyses for dCas9:gRNA3:Cre, significantly differentially methylated ($q < 0.01$, methylation difference >25%) CpGs were determined by comparison to dCas9:gRNAscr:Cre with the calculateDiffMeth function after filtering for CpGs that were covered at least 5× in all samples. Off-target site manhattan plot generated with R package qqman[134].

**Quantification of *FMR1* CGG repeat methylation**. DNA from Fragile X patient fibroblasts treated with lentiviral dCas9 and either lentiviral gRNAscr or gRNA-CGG was isolated by the phenol-chloroform method as described above. Two microgram DNA from each condition was digested for 4 h at 37 °C in a thermocycler in a 20 μL reaction containing 2 μL rCutsmart buffer and 1 μL Fnu4HI restriction enzyme (NEB) or in a control reaction without enzyme. Methylation sensitivity of the enzyme was verified in parallel by digestion of unmethylated or in vitro (M.SssI) methylated plasmid DNA and agarose gel electrophoresis. Following restriction digest, DNAs were re-purified using Monarch® PCR & DNA Cleanup Kit (NEB). DNA concentration was measured by NanoDrop and DNAs were diluted to 20 ng/μL for use with the AmplideX® mPCR FMR1 assay (Asuragen). Note that the AmplideX® mPCR FMR1 assay involves restriction digest with methylation sensitive enzyme HpaII that is directly outside the CGG repeat region, and is not informative for the methylation status of the CpG dinucleotides that make up the CGG region. Therefore, the protocol was modified as described above to allow for digestion with the methylation sensitive enzyme Fnu4HI (recognizes GCNGC) and PCR amplification was carried out with only the control workflow (FAM: no digestion) from the manufacturer. Briefly, 8 μL of diluted sample DNA was mixed with 2 μL control DNA. Four microliter of this mixture was incubated for 2 h at 37 °C with 3.7 μL Digestion Buffer and 0.3 μL Control Enzyme (FAM). Then 20 μL GC-Rich Amp Buffer, 0.1 μL GC-rich polymerase mix, and 1.9 μL FAM-Primers were added to each reaction and PCR was performed with the following cycles: 1 cycle of 95 °C for 5 min, 27 cycles involving 97 °C for 35 s, 62 °C for 35 s, and 72 °C for 4 min, 1 cycle of 72 °C for 10 min followed by cooling to 4 °C. Ten microliter of each PCR reaction was then mixed with 2 μL Gel Loading Dye, Purple (6X) (NEB) and run at 80 V for 20 min and 100 V for 60 min on a 1% agarose gel followed by staining with ethidium bromide solution for 15 min and visualization with Molecular Imager® Gel Doc™ XR+ (Bio-Rad). Quantification of band intensities was achieved with the Gel Analysis utility in ImageJ software.

**Viral integration site detection**. Viral integration sites were defined by following a pipeline developed by Ho et al.[135] with several key modifications. First, quality trimmed WGBS reads (from above) were aligned with bowtie2 v2.3.4.1 (--very-sensitive-local option) to custom FASTA files containing in silico bisulfite-converted sequences (CG to YG, C to T) of forward and reverse strands of the integration-capable lentiviral elements (between two LTRs) from all treatments for that particular cell line: dCas9 plasmids, gRNA3, or gRNAscr plasmids, and Cre plasmids. Notably, the sequence from the lentiviral dCas9 plasmid sequence was in silico recombined (deletion between loxP sites, leaving one loxP site) to mimic Cre action in the cells. Then samtools v1.3.1[136] was invoked to extract all aligned soft-clipped reads; these are reads that were clipped in order to align to the lentiviral sequences and therefore the clipped portion represents possible read-through into mouse genome (no difference from Ho et al.). We then ran a modified variant of the script published by Ho et al. (to allow for alignment to mouse bisulfite converted genomic sequences generated by Bismark rather than human unconverted genomic sequences) that used BLAST[137] to identify boundaries between viral and mouse sequences (Supplementary Software 1). All overlaps with dmCpGs were performed with BEDTools intersect v2.29.2[138].

**Statistics and data visualization**. All data involving simple statistical tests not described above in WGBS and ChIP-seq methodology (e.g. *T*-test, Mann–Whitney test, Pearson's *r*, Holm-Sidak correction for multiple testing) were calculated and graphed with Graphpad Prism 8 software.

**Reporting summary**. Further information on research design is available in the Nature Research Reporting Summary linked to this article.

## Data availability
The data that support this study are available from the corresponding author upon reasonable request. The whole genome bisulfite sequencing (WGBS) data generated in this study have been deposited in the Gene Expression Omnibus (GEO) database under accession code GSE162138. The chromatin immunoprecipitation sequencing (ChIP-seq)

data generated in this study have been deposited in the GEO database under accession code GSE174275. Mouse mm10 genome is available at https://www.ncbi.nlm.nih.gov/assembly/GCF_000001635.20/. Publicly available microarray data used for candidate gene selection for Supplementary Table 7 is in the GEO database under accession code GSE8374. Source data are provided with this paper.

## Code availability
Code used to find viral integration sites is available as Supplementary Software 1.

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

## Acknowledgements

This study was funded by the Canadian Institutes of Health Research (PJT159583). D.M.S was supported by fellowships from the McGill University Faculty of Medicine (Friends of McGill Fellowship; JP Collip Fellowship in Medical Research; James Frosst Fellowship). We thank the Next Generation Sequencing team of the Centre d'expertise et de services Génome Québec at McGill University for their sequencing services. We also thank Julien Leconte for performing cell sorting at the McGill University Flow Cytometry Core Facility and Andrew Bayne of McGill University for assistance in visualization of the CRISPR/Cas9 structure.

## Author contributions

D.M.S. and M.S. designed all experiments. D.M.S. performed all experiments and created all of the figures. D.M.S. and M.S. both contributed to data analysis and writing of the manuscript.

## Competing interests

D.M.S. and M.S. have no competing interests to declare.
