## [Peer Review File · Nature Communications]

Unraveling the functional role of DNA demethylation at specific promoters by targeted steric blockage of DNA methyltransferase with CRISPR/dCas9REVIEWER COMMENTS

Reviewer #1 (Remarks to the Author):

In this manuscript "Unraveling the functional role of DNA methylation using targeted DNA demethylation by steric blockage of DNA methyltransferase with CRISPR/dCas9", the authors DM Sapozhnikov and M Szyf have tested an important hypothesis, using the versatile CRISPR/dCas9 tool, that demethylation of highly methylated promoters in cell lines and primary cells results in re-expression of genes marked by DNA methylation at their promoter region.

These types of hypotheses are important to test, and was not possible before Crispr development because of the lack of tools which are devoid of confounding variables. Generally, in the DNA methylation field, inability to separate cause and consequence led to accumulation of correlative literature and it's only now, with the help of Crispr epigenetic editing that we can explore these questions with high precision.

The result they communicate is that CpG poor promoter demethylation does not result by default in gene transcription initiation. This is an important finding because we can start understanding tissue specific gene promoter methylation (which is the most variable in CpG poor promoters, while CpG rich promoters are generally devoid of DNA methylation in the normal tissue context, with the exception of germline genes and pathologically hypermethylated CpG rich promoters). This study differs from previous studies by the fact that they are using dCas9-gRNA alone to inhibit maintenance methylation to cause demethylation without other confounding variables. Thus, this study provides strong evidence to support the above conclusion.

Also, what this study is reporting and concluding is that using TET enzymes (which are often called DNA demethylases, although they only catalyze the first step in the process of demethylation by oxidising and converting 5-methylcytosine to 5-hydroxymethylcytosine) to study the direct effect of promoter DNA demethylation on gene expression is often inappropriate due to the TET enzymatic activity independent effects on inducing gene transcription.

These data are very useful for the community and the manuscript is well written, however I do have come comments that need to be addressed:

1. The major caveat in this study is that when studying the effect of promoter demethylation, this study is focusing mainly on CpG poor promoter genes. There is only one CpG rich promoter that is tested at the end of the manuscript, but unfortunately they could not measure DNA methylation before or after treatment due to technical reasons. A previous study, looking at a CpG rich promoter, demonstrated that the presence of DNA methylation interfered with transcriptional efficiency (and was oxidation/demethylation dependent in this case and increased the biological effect above the level of the catalytically inactive TET) and modulated thus the switch of cell identity (PMID: 31073172), work that should be acknowledged. This is strengthened by the authors own data in the FMN gene, the only CpG rich gene analysed in this study. There is, therefore, not sufficient evidence in this manuscript to conclude and generalise that demethylation of CpG rich promoters does not lead to gene re-expression. But they provide strong evidence that genes with CpG poor promoters are less inhibited, or sometimes not inhibited, by DNA methylation.

2. The title is not representative for the discoveries shown. First, the authors study the function of DNA demethylation and the consequences of it. They did not induce DNA methylation with a methyltransferase fused to dCas9 to study the effect of DNA methylation. The authors acknowledge the increasing number of previous studies using Crispr/dCas9 editing to study the functional impact of DNA methylation but not all of them fail to disentangle cause and effect. For example, a previous study reported on cause and functional consequence in promoter DNA methylation in a pathological context, using a transient epigenetic editing system in primary human cells (PMID: 29133799). Therefore, as it stands in the abstract is not accurate "Different strategies to address this question were developed; however, all are confounded and fail to disentangle cause and effect." At least in Saunderson et al. the study provided strong evidence that aberrant gain of DNA methylation prevented gene expression of p16 therefore primary breast cells avoided senescence entry. This sentence in the abstract needs amending.

Second, the title actually distracts from highlighting the significant finding that targeted removal of DNA methylation can be highly specific to chosen CpGs and because it is done without confounding

effects (as indeed the authors argue) further demonstrates a growing assertion that DNA demethylation does not result in gene re-expression by default, therefore the two cannot be equated – and this is the case particularly in CpG poor promoters as this study demonstrates. This was previously observed in an extreme case where cells that undergo global DNA demethylation but did not result in promiscuous increase in gene expression (PMID: 23850245). It is also being regularly observed in various studies and experiments where researchers expect gene reactivation following promoter DNA methylation loss, but in fact they find no gene reactivation. Therefore, this study is a direct demonstration that, except pathologically methylated promoters and possibly germline promoters, demethylation does not equal gene reexpression. Similarly, this sentence in the abstract “enabling examination of the role of DNA methylation per se in living cells.” should read “enabling examination of the role of DNA demethylation per se in living cells.”

3. The authors write “The promoter is highly methylated in NIH-3T3 cells” and Fig 1B shows the native methylation in NIH-3T3 cells of CpGs 9,10,11 (control cells) – can the authors show a figure with all the CpGs and their methylation levels in untreated, untransfected cells? This is important for all subsequent comparisons because in some cases lentiviral transduction (using scrambled gRNA) reduces methylation at CpGs 9,10,11 from ~80% to ~60% (Fig 4A-C) – particularly CpG9.

4. In Figure 1 (B,D,H-J) and subsequent figures too, are replicates technical or biological? A biological replicate would be separate transfection experiments. A technical replicate would be the same DNA from one experiment amplified/analysed separately.

5. Figure 1D shows gene expression data of transcription factors induced upon 5-aza treatment – these haven’t been measured at the protein level, how do the authors know that these TFs have increased presence of the respective proteins? See also comment 7.

6. The data in this paragraph is interesting but slightly difficult to follow: “First, it was immediately apparent that NIH-3T3 cells expressing only a scrambled, non-targeting guide (gRNAscr) and dCas9-TET were significantly more demethylated than counterparts expressing the same non-targeting gRNA and dCas9-deadTET (Figure 1E-G), particularly at CpGs 5, 10, and 11 (range from ~22 to 26 percent) but also to a smaller degree at all remaining evaluated CpGs.” The word “counterpart” makes it confusing. Maybe just simplify by saying that TET fusion compared to deadTET reduced methylation in CpG 5,10,11 as expected but was not very different from scrRNA and a site specific gRNA “indicative of a potential ubiquitous and dCas9-independent activity of the fused, over-expressed TET domain.”

7. Figure 1H : “ Third, when evaluating the transcriptional effects of the epigenetic editing system, we were surprised to discover that dCas9-deadTET paired with gRNA 1 or 2 (gRNA3 blocks the TSS and likely interferes with RNA polymerase binding [40]) resulted in strong transactivation of the IL33-002 transcript to levels comparable to dCas9-TET (Fig. 1H),” can this expression be compared to the level induced by 5-aza treatment? How are the global/genome-wide effects of dCas9-deadTet compare to the global effects of 5-aza? Maybe the overexpression seen by 5-aza is really down to IL33 promoter demethylation (i.e. no other global effects are relevant particularly in the absence of protein expression data associate with Fig 1D, which would make Fig 1D obsolete and slightly misleading). This is strengthened by the authors’ own observation in Fig 5D that 5-aza treatment does not further increase IL33-002 expression upon demethylation with dCas9 combined with gRNA3.

8. This is interesting indeed “Yet, we found that dCas9-VP64 co-expressed with all 3 IL33-002 gRNAs resulted in dramatic and broad demethylation of the IL33-002 promoter” Is demethylation happening passively across the locus then gene expression is induced subsequently or gene expression happens irrespective of demethylation? I suppose treating cells with a cell cycle inhibitor can test this? Moreover, this is a CpG poor promoter, can this be generalized to highly methylated CpG rich promoters? The cell cycle arrest could be tested for both CpGpoor and CpGrich promoter. These are important to address to claim that demethylation happens subsequent to transcriptional activation.

9. In the luciferase construct is the whole region in Fig1A inserted, can the authors add a scale or indicate from where to where the fragment goes? “The target DNA used for methylation was

a 1,015 bp fragment of the II33-002 promoter (Fig. 1A) inserted into an otherwise CpG-free luciferase reporter vector [45]". Also, how was transfection normalization done in this experiment in the absence of renilla luciferase?

10. Figure 4 experiments: The authors need to globally analyse off target DNA demethylation caused by their method using dCas9 and site specific gRNA compared to plasmid backbone transduced cells and scrRNA, which is lacking at the moment. It would be easy to argue that it is very unlikely that off target effects are not happening, but until the experiment is done, we don't really know. At the moment the assumption is, based on the three gRNAs, that there are close to none non-significant off-target effects. Figure 4A does show some off target effects (some of them are statistically significant although likely not biologically significant) but we don't know if there are genome wide effects. Similarly, looking at data in Fig 4 many scr treated cells demethylate, is this off target effect of the Cas9 in cells? The authors find "with the lone exception of a single CpG in one clone that displayed 39.5% methylation", the untransduced cells (in Fig 1 and see comment 3) are around ~80% methylation in these CpGs, what causes this demethylation if not off-target effect.

11. The precision could be further increased using targeted bisulfite sequencing, in order to make the claim below robust, by analyzing the DNA the authors might still have: "We conclude that we were able to produce cell lines with almost completely demethylated target CpGs with this approach (the small level of methylation detected in these clones is around the standard error for unmethylated controls in our pyrosequencing assay)."

12. dCas9 expression was measured as transcript (normalized to GAPDH as in figure legend) but a better correlation is with the Cas9 protein level by FACS (there are Cas9 specific antibodies available). "The clonal analysis suggests a clonal variation in the extent of demethylation by dCas9:gRNA. A plausible cause could be variation in the level of expression of either dCas9 or the gRNA. dCas9 expression did not correlate with methylation levels (Supp. Fig. S3D)"

13. Much of the data presented is very interesting so this point is just a comment on one of the observations. One main difference between epigenetic editing in primary cells and cell lines is that in the absence of the transiently transfected Cas9-DNMT3a fusions, cell lines lose the methylation induced, whereas primary cells seem not to. It is interesting that in this system methylation is not regained which is an important observation. "Interestingly, low levels of methylation persisted for at least 75 days after removal of dCas9 by Cre recombinase (Fig. 5B), indicating a lack of de novo methylation of this locus in these cells and the ability of this approach to modify DNA methylation in a stable manner despite elimination of dCas9." Can the authors hypothesise in the discussion how many such sites might be redundant in any given cell line?

14. Again the finding that dCas9 induced demethylation does not induce gene expression (might be statistically significant but it's small), is interesting and important. "We detected a small but significant ($p=0.0312$) increase in expression in NIH-3T3 cells treated with dCas9:gRNA3 and Cre recombinase as compared to dCas9:gRNAscr, but not in dCas9:gRNA1 or dCas9:gRNA2 cells (Figure 5C). This is consistent with our in vitro/transient transfection luciferase assays findings (Fig 2F); both approaches suggest that methylation of TSS CpGs 9, 10, and 11 silence the basal II33-002 promoter."

I wonder if this is because this promoter is a CpG poor promoter? Nevertheless, as mentioned above, since methylation of CpG poor promoters seems to be tissue specific, it's important to have this knowledge as precedence that methylation removal will not change gene expression in a significant way.

15. Figure 5F: Not sure if I missed it but what is, quantitatively, the difference in induction of IL33 in cells with highly methylated promoters (scr gRNA) and gRNA3 demethylated and subsequent CRE removed? In other words, how much does the promoter methylation inhibit gene expression? Both in the case of poly(I:C) and LPS?

16. SERPINB5 and Tnf demethylation data is also interesting but again these genes don't have a CpG island, correct? The authors say: "consistent with the conclusion that demethylation of the promoter is insufficient for its expression and demethylation of other regions, such as the depicted enhancer regions (Fig. 6A), is required for induction of SERPINB5". Have the authors tested this hypothesis that enhancers of these genes need to be demethylated? If not, why not?

17. The FMR gene seems to be the only gene which recapitulates the expected demethylation-re-expression pattern and coincidentally this gene has a CpG island at the promoter region. Question: "We obtained primary fibroblasts from a patient with Fragile X syndrome with approximately 700 CGG repeats exhibiting high methylation [69]" Are these primary cells from the same patient as in reference or a different patient. If a different patient, the authors need to first test the composition and methylation level of the gene promoter.

18. The authors say that they could not measure methylation levels in the promoter region of FMN due to technical reasons: "We were unable to map DNA methylation in the dCas9 demethylated pools due to technical challenges in amplification of such a highly repetitive and CG-dense region, despite attempting to do so with buffers and polymerases optimized for GC-rich templates, as well as bisulfite-free approaches in an effort to avoid the decreased sequence diversity caused by sodium bisulfite (C to T transitions)." Couldn't the authors replicate the methods used in the reference cited 69, looks like they submitted samples to <https://www.epigendx.com/> ? Alternatively, the authors could do MeDIP-qPCR with several primer pairs, or Nanopore? Perhaps they could confirm that there is increased expression of FMR1 after 5-aza treatment, to confirm that it is demethylation dependent?

19. This is quite unfortunate that the authors couldn't measure their DNA demethylation level on a promoter sequence that seems to be the only gene in this study truly responsive to DNA demethylation. This further strengthens the assertion that it's really CpG island promoter containing genes that are the most influenced by DNA methylation.

20. Demethylation of the HNF4 gene upon crispr editing of the DNA sequence is very interesting but can one broadly conclude that extensive promoter demethylation upon catalytically active Cas9 editing happens based on one case? Moreover, it is off-topic to some extent, not sure how is it related to the rest of the figures. This data is good as a preliminary look at possible demethylation events that occur after Cas9 editing. It looks like even the unedited clones in the gRNAHNF4A (4 of the strands shown in Figure 7C) are completely unmethylated, whereas in the scrambled ones they are majority methylated. Perhaps I misunderstand and all the gRNAHNF4A clones have been edited? Can the authors explain this data? Nevertheless, I find it off-topic and would be best to expand the dCas9 demethylation method on other CpG rich promoters, some pathological (many are known) or germline like Dazl for example.

Minor point:

Interesting data, but it would be good to have a figure to illustrate this: 'The results are in accordance with the crystal structure of the dCas9:gRNA:DNA triplex, which reveals minimal 5' protrusion of dCas9:gRNA beyond the 5' end of the gRNA target and more pronounced extension of both dCas9 protein and, to a larger degree, gRNA scaffold beyond the 3' end of the gRNA target sequence [48]' Similarly, the figure may help clarify this part: 'Interesting, the 3' footprint is smaller by at least 2bp (and at most 6bp) than in the strand interacting with the gRNA, as CpG 19 is not affected on the antisense strand.'

Reviewer #2 (Remarks to the Author):

In this manuscript, Sapozhnikov and Szyf reported to use a catalytically dead Cas9 to induce targeted DNA demethylation at several genomic loci in dividing cells. They first compared the demethylation effects by dCas9-Tet1 and dCas9-dead Tet1 at the I133 locus in NIH3T3 cells. Then they showed a dead Cas9 can prevent the occupied DNA being methylated in vitro and induced a moderate demethylation in cells at the I133 locus. The efficacy of induced demethylation can be enhanced by

passaging the targeted cells or subsequent cloning. They further showed that targeted demethylation by dCas9 at the I133, SERPINB5, Tnf loci only achieved a small or no elevation in gene expression. Only demethylation of the CGG repeats at the FMR1 locus induced a significant upregulation of FMR1 expression. Last, they described an untargeted DNA demethylation during CRISPR/Cas9 mediated gene knockout. Overall, the topic of this study is of great interest aiming to address the critical question on correlation vs. causality of DNA methylation at the promoter regions. However, some conclusions are either overstated or not supported by their current design and data. The following points need to be experimentally addressed to improve the quality of this manuscript before it can be recommended to be accepted for publication.

1. In Figure 1, the authors reported a surprise observation that dCas9-dead Tet1 was able to induce a significantly higher elevation of I133 expression compared to dCas9-Tet1. To support this conclusion, the expression levels of dCas9-Tet1 and dCas9-dTet1 in these experiments need to be carefully examined by qPCR and western blot.
2. It is odd to use Tet2, but not Tet1, to demonstrate Tet protein can induce DNA methylation independent transactivation in HEK293T cells. To be consistent with the previous experimental setting, Tet1 and NIH3T3 cells are needed to be used.
3. In Figure S3D and S3E, without a statistical analysis, it is not conclusive that sgRNA expression but not dCas9 expression level correlates with the demethylation level given the small number of samples (less than 10).
4. Regarding the potential off-target effect by dCas9, assaying four candidates is far away to be conclusive. A genome wide approach such as whole genome bisulfite sequencing is required to address this concern.
5. Based on the proposed mechanism for dCas9-induced DNA demethylation, post-mitotic cells such as neurons are not editable. At least one locus is needed to be tested by dCas9 in neurons such as the FMR1 locus.

Reviewer #3 (Remarks to the Author):

This is a novel study where the authors develop and test, from many angles, a new methodology to study in vitro and in vivo site specific CpG methylation/demethylation. They use an enzyme-free CRISPR/dCas9-based system for targeted methylation editing that achieves selective methylation in vitro and passive demethylation in cells through steric interference with DNA methyltransferase activity. They map the size of the region of interference, optimize the system for improved demethylation of targeted CpGs without detectable off-target effects within the locus, and analyze the transcriptional consequences of demethylation of different genes known to be highly activated by 5-Aza treatment. They perform, side by side, their method using dCas9, and the published method using dCas9-TET1 on the I133 locus to point to the improvements in specificity and the ability to methylate/demethylate single CpG sites.

In general, the claims are convincing and appropriately discussed in the context of previous literature and the work is technically sound. Only their claim that this model does not introduce demethylation on off-target regions would have been better supported by whole genome methylation analysis. They show however that with this method they modify the methylation of unique CpG sites on the I133 promoter without affecting nearby CpGs as a proof of specificity; this level of specificity was not achieved with the commonly used dCas9-TET1 approach used here as control.

The findings will be of great interest to others in the field of DNA methylation since, due to limitations in specificity, new methods to study methylation are continuously searched for. The paper will also inform the CRISPR/Cas9 field, since they show marked demethylation around the TSS of a CRISPR/Cas9 targeted gene that results in altered transcription of the mutated gene. So they suggest that care should be taken in the interpretation of results from all CRISPR/Cas9 editing experiments that target regions close to the TSS and consider whether the simple presence of the Cas9 protein on the DNA was blocking, by steric hindrance, the re-methylation around the targeting region resulting in unexpected results.

The manuscript is clearly written but they should remove some discussion from the Results part.

For most of the experiments the statistical analysis seems fine. There are some where the statistical significance is not marked on the panels (asterisks), and others where it is not clear which groups are

being compared in the statistical analysis. Some experiments in the supplemental material have an n=1. If they are not essential they should be removed from the manuscript or if they want to retain the information they should increase the n. The particular experiments are indicated in the Specific Comments.

They provide sufficient methodological detail that the experiments could be reproduced. However, they need to modify some information in methods to be more explicit about the structure of the plasmids used.

Specific Comments

Results

CRISPR/TET-based approaches confound the causal relationship of DNA methylation and transcription:

Paragraph 1. Does the treatment with 5'aza also reduce methylation of CpGs 1-5? Why only 9-11 are shown?

Paragraph 3. Although they say that the "differences in demethylation were significant", they do not indicate p values in the text or in the Figure 1E-G legend.

They also suggest that "TET1 would interact with several DNA targets across the genome by their cognate DNA binding domain in the absence of any additional targeting." However they do not consider that the gRNAs or the dCas9 itself could introduce a low degree of off-targeting effects.

Paragraph 4. It would have been more informative and would have allow a stronger conclusion about specificity if they would have tested the whole-genome changes in methylation patterns caused by dCas9-TET1 vs. their method with dCas9.

Paragraph 6. Why they use a pcDNA-TET2 plasmid to show the transcriptional activity of TET proteins when they have performed all the other studies with TET1.

Paragraph 7. Are there other publications which show that TET proteins can act across large genetic distances. A reference must be included.

A novel method for site-specific DNA methylation in vitro:

Paragraph 1. They state that "since dCas9 is a prokaryotic protein that is unlikely to interact with other proteins in the gene transcription machinery and has no enzymatic activity epigenetic or other, there should be limited confounding factors to DNA demethylation." They are not providing proof of this statement. So they need to include references.

Paragraph 2. Please, indicate that the CpG - free vector is referred as pCpGI, next to the reference.

Paragraph 4. "These data demonstrate the exquisite impact of site-specific methylation rather than just methylation density, and thus this assay appears to capture the sequence specificity of inhibition of promoter function by DNA methylation." Earlier work used to mutate individual CpG sites to evaluate the importance of each site. They should comment about the benefits of their method compared to individual CpG mutation.

The dCas9 system directs robust site-specific demethylation in living cells:

Paragraph 3. "... there was detectable significant downregulation of I133-001 under gRNA1." Have you check the methylation status of I133-001? Are there potential off-targets for gRNA1 close to I133-001? "We also assessed whether dCas9:gRNA3 caused demethylation of the top 5 predicted candidate off-target CpGs for gRNA3". Why only with gRNA3 and not the other 2? Were the corresponding targets affected in the dCas9-TET1 experiments?

The effect of site-specific demethylation on I133 gene expression:

Paragraph 1. Mislabeled Figure 3G change to S3G.

Paragraph 3. "One of the two base substitutions to render this dCas9 variant nuclease-dead (D10A, H840A) is different than the dCas9 used in previous experiments (D10A, N863A)." Why do they use a different dCas9 sequence? Is it to avoid potential cre cleavage of dCas9? Please explain.

"Chromatin immunoprecipitation of cells from Fig. S6E with antibody for dCas9 (anti-Flag) followed by qPCR using primers surrounding the I133-002 TSS." Please indicate that the dCas9 plasmid used has a 3xflag 5' to the dCas9 sequence.

"ChIP-qPCR demonstrated elevated dCas9 binding to the I133-002 promoter region only in cells stably

expressing dCas9 and gRNA3 but not in dCas9:gRNAsc cells regardless of Cre treatment (Supp. Fig. S6F). The figure legend indicates "dCas9 binding is expressed as percent input (n=1)." This ChIP is a critical control to support that the dCas9 is bound or not to the promoter region of these genes. So it should be performed more than once to assess its statistical significance.

"...low levels of methylation persisted for at least 75 days after removal of dCas9 by Cre recombinase (Fig. 5B), indicating a lack of de novo methylation of this locus in these cells and the ability of this approach to modify DNA methylation in a stable manner despite elimination of dCas9."

Is this finding generalizable to other loci. How long the demethylation persisted in the other experiments tested like in SERPINB5, Tnf and FMR1?

Paragraph 4. "It is possible that the small magnitude of induction of expression by demethylation of the TSS region can be explained by the presence of other methylated regulatory regions or other required trans-acting factors that need to be demethylated to facilitate larger changes in expression".

Have you checked if they have different binding of pIII?

Paragraph 7. "These data are consistent with the hypothesis that methylation silences basal promoter activity but does not affect inducibility." This concept is only relevant to this locus and it is not necessarily generalizable.

Discussion

Paragraph 11-12. "Indeed, we did not detect off target demethylation, in contrast to the dCas9-TET fusion construct." This is true only for the loci tested in this work. Because you do not test genome-wide changes in methylation, as you indicate in the next paragraph: "Though our off-target analyses could be augmented by more comprehensive analyses (whole-genome bisulfite sequencing), our initial observations suggest no off-target demethylation by this approach."

Methods

gRNA design and synthesis. unformatted reference

([dx.doi.org/10.17504/protocols.io.dwr7d5](https://doi.org/10.17504/protocols.io.dwr7d5))

Please indicate which elements are included in the EcoR1-flanked gRNA scaffold.

Supplementary Table S5

Very high CTs for most genes and n=1. It is difficult to interpret any fold changes.

Figure Legends

In all figure legends please indicate the n for each experiment.

Clarify which bars are being compared to determine p in all experiments.

Figure 1. Please indicate n for 1B,C and D

Are differences in %methylation significant in 1E,F and G? Please show asterisks on the figure.

Figure 3. There is a difference in the pattern of methylation between C and E corresponding to the different strands. Why do you consider this pattern symmetrical?

Figure 5. Why is the difference in n between C (n=3) and (D-E) (n=4-5) when the experiments in D-E were performed with cells from C? Are these biological or technical replicates?

How did you determine which were relevant in the maximal induction experiment?

In panel H how many n you selected for each maximal induction and methylation?

Figure 6. Which CpG sites are you targeting in Maspin?

What is the purpose to show the same data in two formats in B and C?

Please number the CpG sites 1-6 in the diagram in 6A.

Why do you subclone the lowly methylated gRNA maspin clones to measure expression?

gRNA-CGG is too GC reach. Would that cause more off-targets effects?

Figure S2A. Please re-label site 10 (it is mis-labeled 12).

Figure S3A. Please add significance asterisks.

Figure S6D. Indicate in the diagram that this dCas9 has a 3XFlag 5'. Without it is not clear why do you use anti-flag antibody for the ChIP and other analysis. (pLV hUbc-dCas9-T2A-GFP, Addgene #53191).

Reviewer #1 (Remarks to the Author):

In this manuscript "Unraveling the functional role of DNA methylation using targeted DNA demethylation by steric blockage of DNA methyltransferase with CRISPR/dCas9", the authors DM Sapozhnikov and M Szyf have tested an important hypothesis, using the versatile CRISPR/dCas9 tool, that demethylation of highly methylated promoters in cell lines and primary cells results in re-expression of genes marked by DNA methylation at their promoter region.

These types of hypotheses are important to test, and was not possible before Crispr development because of the lack of tools which are devoid of confounding variables. Generally, in the DNA methylation field, inability to separate cause and consequence led to accumulation of correlative literature and it's only now, with the help of Crispr epigenetic editing that we can explore these questions with high precision.

The result they communicate is that CpG poor promoter demethylation does not result by default in gene transcription initiation. This is an important finding because we can start understanding tissue specific gene promoter methylation (which is the most variable in CpG poor promoters, while CpG rich promoters are generally devoid of DNA methylation in the normal tissue context, with the exception of germline genes and pathologically hypermethylated CpG rich promoters). This study differs from previous studies by the fact that they are using dCas9-gRNA alone to inhibit maintenance methylation to cause demethylation without other confounding variables. Thus, this study provides strong evidence to support the above conclusion.

Also, what this study is reporting and concluding is that using TET enzymes (which are often called DNA demethylases, although they only catalyze the first step in the process of demethylation by oxidising and converting 5-methylcytosine to 5-hydroxymethylcytosine) to study the direct effect of promoter DNA demethylation on gene expression is often inappropriate due to the TET enzymatic activity independent effects on inducing gene transcription.

These data are very useful for the community and the manuscript is well written, however I do have come comments that need to be addressed:

We thank the reviewer for the exceedingly thorough and careful review of all the experiments. We very much appreciate the fact that the reviewer noted that the manuscript useful and well written. The reviewer revealed a number of shortcomings and flaws that we address with the point-by-point response below.

1. The major caveat in this study is that when studying the effect of promoter demethylation, this study is focusing mainly on CpG poor promoter genes. There is only one CpG rich promoter that is tested at the end of the manuscript, but unfortunately they could not measure DNA methylation before or after treatment due to technical reasons. A previous study, looking at a CpG rich promoter, demonstrated that the presence of DNA methylation interfered with transcriptional efficiency (and was oxidation/demethylation dependent in this case and increased the biological effect above the level of the catalytically inactive TET) and modulated thus the switch of cell identity (PMID: 31073172), work that should be acknowledged. This is strengthened by the authors own data in the FMN gene, the only CpG rich gene analysed in this study. There is, therefore, not sufficient evidence in this manuscript to conclude and generalise that demethylation of CpG rich promoters does not lead to gene re-expression. But they provide strong evidence that genes with CpG poor promoters are less inhibited, or sometimes not inhibited, by DNA methylation.

The reviewer is entirely correct in noting the limitations of this study. We want to stress the fact that we are not attempting to make conclusions as to the functional consequences of

CpG methylation of CpG-rich promoters or any classes of any elements and we believe that we did not make any such statements in the manuscript. Likewise, we are not making any conclusions even for CpG-poor promoters, even though these represent the majority of those analyzed in this study. The conclusions we made are limited strictly to the small number of genes – and in fact, really only the specific CpGs – analyzed in this study. We agree that we lack the comprehensive analysis to make any broad conclusions about any type of gene or element. The overall responses in the CpG-poor promoters were small in magnitude, but it would be unrealistic to predict that all genomic CpG-poor promoters have such responses, as even within our study the responses were qualitatively different. It is entirely possible that some number of both CpG-poor and CpG-rich promoters may display dramatic transcriptional changes in response to demethylation, and it would be fascinating to discover the mechanisms that control this. Unfortunately, we did not have the time and capacity yet to perform a comprehensive analysis of the great number of promoters that would be necessary to make these conclusions: the goal of this manuscript was to produce a method capable of doing so and evaluating that method in a number of ways. The goal of this manuscript was not to define how specific elements in the genome may respond to demethylation.

As an additional note, CpG-rich promoters are also more difficult to target with this method, as the size of the demethylation is fairly small and a multi-gRNA system needs to be made and optimized in order to achieve robust demethylation of larger CpG-rich regions. Of course, many studies have used multiple gRNAs with CRISPR/Cas9, so we do not anticipate this to be a major obstacle, but have not optimized this as of yet. At present, our method allows for the interrogation of individual CpGs or CpGs within a very small distance (e.g. CpGs 9,10,11 in IL33-002).

Again, we would like to state that we are not *per se* analyzing CpG-rich promoters or even CpG-poor promoters; rather, our method allows us to analyze CpGs or individual sets of CpGs within different contexts (such as a promoter). We spend some time evaluating specific CpGs, but a large portion of this manuscript is devoted to evaluating the method.

This is certainly something we wish to do, but we feel that it is an entirely different project, and that it is responsible to publish this technique so that the larger scientific community can sooner have knowledge of this technique and possess the ability to assess the functional effect of methylation at any and all genes of interest, and thus achieve this goal together. Some goals of this study were to demonstrate how this technique could be applied, determine the size of interference with methylation, analyze off-target effects, compare to dCas9-TET, and verify that it works in different genes and in different species and that different genes respond differently, which we believe we achieved.

2. The title is not representative for the discoveries shown. First, the authors study the function of DNA demethylation and the consequences of it. They did not induce DNA methylation with a methyltransferase fused to dCas9 to study the effect of DNA methylation. The authors acknowledge the increasing number of previous studies using Crispr/dCas9 editing to study the functional impact of DNA methylation but not all of them fail to disentangle cause and effect. For example, a previous study reported on cause and functional consequence in promoter DNA methylation in a pathological context, using a transient epigenetic editing system in primary human cells (PMID: 29133799). Therefore, as it stands in the abstract is not accurate “Different strategies to address this question were developed; however, all are confounded and fail to disentangle cause and effect.” At least in Saunderson et al. the study provided strong evidence that aberrant gain of DNA methylation prevented gene expression of p16 therefore primary breast cells avoided senescence entry. This sentence in the abstract needs amending.

We agree with the reviewer that Saunderson et al. provided evidence that that aberrant gain of DNA methylation prevented gene expression of p16. We have now discussed this

paper in the introduction on page 3: "DNMT3A /DNM3L has also been targeted with dCas9 to methylate and silence p16 in primary breast cells resulting in avoidance of senescence entry[30]." Yet, there remain some potential confounds at the genomic level brought about by the overexpression of dCas9 fused to a domain that has enzymatic activity and protein interaction activity, as we discuss in the paper and demonstrated in part by reference 39 (Galonska et al.).

We also agree with the reviewer that our paper focuses on the role of DNA demethylation in gene expression. The abstract was therefore revised, and we include the following new and revised sentences: "Although associations between DNA demethylation and gene expression were established four decades ago, the causal role of DNA demethylation in gene expression remains unresolved. Different strategies to address this question were developed; however, confounds involved in these approaches may complicate the disentangling of cause and effect."

Finally, we have renamed the title to be more reflective of the discoveries shown: "Unraveling the functional role of DNA demethylation at specific promoters by targeted steric blockage of DNA methyltransferase with CRISPR/dCas9".

Second, the title actually distracts from highlighting the significant finding that targeted removal of DNA methylation can be highly specific to chosen CpGs and because it is done without confounding effects (as indeed the authors argue) further demonstrates a growing assertion that DNA demethylation does not result in gene re-expression by default, therefore the two cannot be equated – and this is the case particularly in CpG poor promoters as this study demonstrates. This was previously observed in an extreme case where cells that undergo global DNA demethylation but did not result in promiscuous increase in gene expression (PMID: 23850245). It is also being regularly observed in various studies and experiments where researchers expect gene reactivation following promoter DNA methylation loss, but in fact they find no gene reactivation. Therefore, this study is a direct demonstration that, except pathologically methylated promoters and possibly germline promoters, demethylation does not equal gene reexpression. Similarly, this sentence in the abstract "enabling examination of the role of DNA methylation per se in living cells." should read "enabling examination of the role of DNA demethylation per se in living cells."

We agree with the reviewer's thoughts here and have now echoed them in the Discussion, citing the study recommended by the reviewer and one related study. We also agree with the recommendations here and have changed the title and several instances of similar terminology throughout the manuscript to more accurately reflect the fact that we are studying DNA demethylation rather than DNA methylation and have also corrected the specific sentence in question.

3. The authors write "The promoter is highly methylated in NIH-3T3 cells" and Fig 1B shows the native methylation in NIH-3T3 cells of CpGs 9,10,11 (control cells) – can the authors show a figure with all the CpGs and their methylation levels in untreated, untransfected cells? This is important for all subsequent comparisons because in some cases lentiviral transduction (using scrambled gRNA) reduces methylation at CpGs 9,10,11 from ~80% to ~60% (Fig 4A-C) – particularly CpG9.

We have included this suggested figure as Supplementary Figure S1A and have included references to it in the relevant text. We renamed the panels in Supplementary Figure S1 to accommodate this change and made corrections in the text accordingly.

4. In Figure 1 (B,D,H-J) and subsequent figures too, are replicates technical or biological? A biological replicate would be separate transfection experiments. A technical replicate would be the same DNA from one experiment amplified/analysed separately.

All replicates presented in this study are biological; while we did perform technical replicates (for example, duplicates or triplicates for luminometer readings, RT-qPCR, pyrosequencing, etc.) these were always averaged together to then represent a single biological replicate. For the luciferase assays, this means (as the reviewer correctly stated) that each replicate represents an independent well of cultured cells that was transfected independently and luciferase activity was measured independently. For all lentiviral experiments, each replicate represents independent cultured cells with independent infections with viral particles and this is now clarified in the methods section on top of page 26: "All replicates presented in this study are biological replicates. A technical replicate was performed for each of the assays presented in the paper and averaged per each biological replicate."

5. Figure 1D shows gene expression data of transcription factors induced upon 5-aza treatment – these haven't been measured at the protein level, how do the authors know that these TFs have increased presence of the respective proteins? See also comment 7.

The reviewer is correct in concluding that we did not confirm increased TF protein levels. However, we only meant to show increased TF expression at the transcript level – because that is the level that 5-aza acts on. By doing so, we illustrate a potential mechanism for possible trans-activation of IL33 gene expression in order to highlight the confounds of genomic demethylation by the global demethylating agent 5-aza. We do not by any means confirm this mechanism to be in play and in fact we would argue that it is not in play as we later demonstrate activation of IL33 by targeted demethylation to similar levels as 5-aza and failure of 5-aza to induce this TSS-demethylated gene further (Fig. 5D). It seems the reviewer agrees with us on this, as discussed in point 7. We therefore feel that measurements of the protein levels of these TFs is distracting from the point and we hope that the reviewer agrees. Perhaps, though, the reviewer might be interested in this example of induction of one of these TFs (Sp1) by 5-aza, confirmed by western blot, though in a different cell type (PMID: 24038389).

6. The data in this paragraph is interesting but slightly difficult to follow: "First, it was immediately apparent that NIH-3T3 cells expressing only a scrambled, non-targeting guide (gRNAscr) and dCas9-TET were significantly more demethylated than counterparts expressing the same non-targeting gRNA and dCas9-deadTET (Figure 1E-G), particularly at CpGs 5, 10, and 11 (range from ~22 to 26 percent) but also to a smaller degree at all remaining evaluated CpGs." The word "counterpart" makes it confusing. Maybe just simplify by saying that TET fusion compared to deadTET reduced methylation in CpG 5,10,11 as expected but was not very different from scrgRNA and a site specific gRNA "indicative of a potential ubiquitous and dCas9-independent activity of the fused, over-expressed TET domain."

Thank you for noticing this. We have tried to simplify the text on page 4 as recommended and hope the changes make the experiments more understandable:

"First, it was immediately apparent that even in the absence of targeting, NIH-3T3 cells expressing only a scrambled, non-targeting guide (gRNAscr) and dCas9-TET were significantly more demethylated than those expressing the same gRNAscr and dCas9-deadTET (Fig.1E-G). While dCas9-TET triggered a 22 to 26 percent demethylation as compared with dCas9-deadTET at CpGs 5 ($p<0.0001$), 10 ($p<0.0001$), and 11 ($p<0.0001$), dCas9-TET:gRNAscr that was not targeted to these sites also caused demethylation at these sites as well as all remaining evaluated CpGs. "

7. Figure 1H :” Third, when evaluating the transcriptional effects of the epigenetic editing system, we were surprised to discover that dCas9-deadTET paired with gRNA 1 or 2 (gRNA3 blocks the TSS and likely interferes with RNA polymerase binding [40]) resulted in strong transactivation of the IL33-002 transcript to levels comparable to dCas9-TET (Fig. 1H),” can this expression be compared to the level induced by 5-aza treatment?

We believe the reviewer will find the answer to this in Figure 5H: the levels of expression induced by demethylation of the TSS are comparable to those induced by 5-aza treatment.

How are the global/genome-wide effects of dCas9-deadTet compare to the global effects of 5-aza?

Though this is a very interesting question, we unfortunately did not have the resources to perform genome-wide expression/methylation analyses of both 5-aza and dCas9-deadTET. We would not, however, expect a great deal of overlap due to the dramatically disparate modes of action of 5-aza (covalent DNMT1 inhibition) and dCas9-deadTET (unclear, perhaps TF recruitment to random sequences?).

Maybe the overexpression seen by 5-aza is really down to IL33 promoter demethylation (i.e. no other global effects are relevant particularly in the absence of protein expression data associate with Fig 1D, which would make Fig 1D obsolete and slightly misleading). This is strengthened by the authors’ own observation in Fig 5D that 5-aza treatment does not further increase IL33-002 expression upon demethylation with dCas9 combined with gRNA3.

We are in complete agreement with the reviewer that our data suggests this in Fig. 5D, as also discussed in point 5, above. However, we do not believe that this makes Fig. 1D misleading or obsolete. The global effects of 5-aza are rarely considered in the literature when assessing the inducibility of a gene of interest and most authors thus conclude that activation by 5-aza indicates methylation sensitivity in *cis*, when, in reality, numerous trans-acting factors can also be induced. We believe this (Fig. 1D) to be a critical figure that highlights an oft overlooked issue, and one that needs to be considered for every gene of interest. To put it differently, this figure emphasizes the need for targeted demethylation strategies and, just because then the targeted demethylation strategy appears to rule it out in the specific case of IL33, it does not, in our minds, retroactively invalidate the need for targeted demethylation, a need demonstrated by this figure. We hope that the reviewer can agree with this justification, but if the reviewer still finds this to be misleading, this is certainly not our intent and we are prepared to remove it.

8. This is interesting indeed “Yet, we found that dCas9-VP64 co-expressed with all 3 IL33-002 gRNAs resulted in dramatic and broad demethylation of the IL33-002 promoter” Is demethylation happening passively across the locus then gene expression is induced subsequently or gene expression happens irrespective of demethylation? I suppose treating cells with a cell cycle inhibitor can test this?

The reviewer poses an interesting mechanistic question. Unfortunately, because our lentiviral strategy to insert dCas9-VP64 and gRNA requires antibiotic selection, cells need to divide for resistant clones to expand and by the time this has happened, demethylation would have already occurred, so we can’t address this question with the particular method used in this paper. A second problem is posed by the nature of cell-cycle inhibitors: they modify pathways that involve great numbers of transcription factors – both general and tissue-specific (PMID: 26825227) – and thus are prone to producing countless transcriptional confounds. However, we believe we have answered the reviewer’s question with an alternative experimental strategy, detailed below. To summarize, the answer is the second one posed by the reviewer: “gene expression happens irrespective of demethylation”, at least at a population level.

We have added Supp. Figure S1J (and adjusted other figure names). In this experiment, we performed a time course experiment in which we transiently transfected IL33 gRNA2 (transient in order to avoid lentiviral integration time/process as a variable) NIH-3T3 cells that were already stably expressing dCas9-VP64, and monitored IL33 expression and methylation at 24, 48, 72, and 96 hours. We found that IL33 induction had already occurred at 24 hours and subsided by 48 hours, despite there being no observable DNA methylation at this time, nor even up to 96 hours. Therefore, we show induction of IL33 in the absence of demethylation, suggesting, as the reviewer said, that gene expression can happen irrespective of DNA methylation and might be preceding and causing demethylation. This experiment is discussed on page 6 of the revised manuscript; "To further support this, we performed a time-course experiment in which we observed activation of transcription of IL33-002 under dCas9-VP64:gRNA2 24 hours after transient transfection but did not detect any demethylation at this time point nor at any time point up to 96 hours (Supp. Fig S1J) ."

However, we wish to caution that we cannot firmly conclude that gene expression is happening irrespective of DNA methylation based on these data. Because we are observing changes in populations of tens of thousands of cells, it remains possible that a very small number of cells (say 10) became demethylated and these cells exclusively are the source of the RNA transcript being measured. In this example, 10 demethylated cells out of >100,000 would be a 0.1% change in methylation, which is well beyond the sensitivity of our quantification. In order to truly conclude that RNA expression can happen in the presence of DNA demethylation, one could capture RNA polymerase actively engaged in transcription and analyze the DNA that it is bound to; only if that DNA is methylated can one conclude that expression can happen from methylated template DNA. However, as expanded upon in the comment below, the goal of our study was to develop a tool to avoid this scenario, and we believe that such investigations – though fascinating – are outside the scope of our aims.

Finally, the purpose of these experiments was to demonstrate that demethylation can occur when a transcription factor is targeted to a gene, even if that transcription factor has no demethylase activity. This was done in an effort to highlight the fact that dCas9-TET approaches are confounded in this manner. Again, while dCas9-VP64-induced expression/demethylation dynamics are fascinating and critical to understand, they are ancillary to the aims of the manuscript.

Moreover, this is a CpG poor promoter, can this be generalized to highly methylated CpG rich promoters? The cell cycle arrest could be tested for both CpG-poor and CpG-rich promoter. These are important to address to claim that demethylation happens subsequent to transcriptional activation.

We recognize the added scientific value that could be gained by the study of CpG-rich promoters, but as discussed in point 1, we did not study CpG-rich promoters (our method cannot demethylate more than a few close CpGs at present), nor did we study a great number of CpG-poor promoters. The limited number of CpGs (in promoters) assayed here prevent us from making such generalizations, and thus we warn against any conclusions beyond the exact promoters – in fact, even beyond the exact CpGs – studied herein and we hope that is clear throughout the manuscript.

The question of whether demethylation is a consequence or cause of transcription activation by dCas9-VP64 (or any TF for that matter) has been a longstanding and fascinating question in the field, and it appears that part of the answer is that it depends on the specific promoter being studied (see reference 10). We test this question, as suggested by the reviewer above: this is now described in Supplementary Figure S1J. Our new experiment presented in this revised manuscript as Figure S1J supports the conclusion that

transcription factors can induce transcription of a methylated gene promoter in a manner that precedes DNA demethylation.

9. In the luciferase construct is the whole region in Fig1A inserted, can the authors add a scale or indicate from where to where the fragment goes? "The target DNA used for methylation was a 1,015 bp fragment of the IL33-002 promoter (Fig. 1A) inserted into an otherwise CpG-free luciferase reporter vector [45]". Also, how was transfection normalization done in this experiment in the absence of renilla luciferase?

The 1,015 bp fragment spans from -844 to +171 relative to the IL33-002 transcription start site. To make this clearer, we have added the genomic positions into Fig. 1A at either end of the IL33-002 promoter fragment and we have also added this to the figure description.

We realize that the absence of co-transfection of a control reporter vector is a topic worth debate and consideration. Despite the apparent appeal of this strategy, there are known confounds introduced by the approach and the benefits may or may not outweigh these dangers [Farr, A. and Roman, A. (1992) A pitfall of using a second plasmid to determine transfection efficiency. *Nucleic Acids Res.* 20, 920.] Knowing these dangers, our laboratory opts to perform transfections of single reporters. In order to ensure consistent transfection between plasmids of different conditions, we take several critical and careful steps. First, all measurements of DNA quantity for luciferase assay experiments were performed with highly sensitive fluorescence-based readings (Qubit) rather than the more common but less accurate spectrophotometer approach in order to ensure that the same amounts of DNA were being transfected across conditions. Second, we use an excess (3:1) of transfection reagent to avoid the possibility of leftover/untransfected DNA. Third, we normalize luciferase activity to total protein content (measured by Bradford) in cell lysate to correct for slight variations in cell lysis efficacy and recovery. Fourth, we performed each biological replicate completely independently: i.e. instead of making a 3X master transfection mix and transfecting into 3 replicate wells of cells (per condition) simultaneously, we performed one transfection per condition per day, such that each of 3 biological replicates were performed on 3 different days, and therefore there is no systemic bias in transfection mix setup between conditions.

To ascertain that this leads to the transfection of equivalent copy numbers of luciferase plasmid between conditions, we have now performed the following experiment and we present the results in the new Supplementary Figure S1N: We transfected n=4 SV40-pCpGI plasmids, methylated and mock-methylated. Each stage of the experiment was done independently and on different days: each methylation or mock methylation, each plasmid DNA quantification, and each 50ng transfection. Then, instead of isolating cell lysate for luciferase assays (at 36h), we extracted DNA and measured by qPCR (no reverse transcription) the luciferase gene copy number relative to copy number of the genomic *Actb* gene. We found that there is no significant difference between the luciferase copy numbers when the vector is fully methylated or fully unmethylated. Therefore, we do not believe that transfection normalization with a second reporter is necessary under these conditions. These data have been added as Supplementary Figure S1N. The description of this experiment was added to the method section on page 26; "We validated that our transfection method results in equal copy numbers transfected for both methylated and unmethylated DNA by measuring copy number of transfected pCpGI 36 hours after transfection (Supp. Fig. S1N)". It was also added to the relevant section for Fig. 1 on page 6: "SV40-pCpGI copy number in cells is equivalent upon transfection of fully methylated or fully unmethylated DNA (Supp. Fig. S1N)."

10. Figure 4 experiments: The authors need to globally analyse off target DNA demethylation caused by their method using dCas9 and site specific gRNA compared to plasmid backbone transduced cells and scrRNA, which is lacking at the moment. It would be easy to argue that it is very unlikely that off target effects are not happening, but until the experiment is done, we don't really know. At the moment the assumption is, based on the three gRNAs, that there are close to none non-significant off-target effects. Figure 4A does show some off target effects (some of them are statistically significant although likely not biologically significant) but we don't know if there are genome wide effects. Similarly, looking at data in Fig 4 many scr treated cells demethylate, is this off target effect of the Cas9 in cells? The authors find "with the lone exception of a single CpG in one clone that displayed 39.5% methylation", the untransduced cells (in Fig 1 and see comment 3) are around ~80% methylation in these CpGs, what causes this demethylation if not off-target effect.

We have followed the reviewer's suggestion and have now analyzed off-target DNA demethylation globally using genome wide bisulfite sequencing of the dCas9 and dCas9-Tet conditions, each in a biological triplicate. This was a critical recommendation. We have now added an entire section titled "dCas9 off-target demethylation events and comparison to dCas9-TET" with a new Figure 6. Briefly, we found 643 significantly hypomethylated CpGs in cells treated with dCas9:gRNA3:Cre compared to dCas9:gRNAscr:Cre. Two of these were IL33-002 target CpGs 10 and 11. The remaining 641 off-target CpGs showed no considerable sequence similarity to gRNA3 and we conclude that they are thus likely not a consequence of off-target dCas9:gRNA3 binding.

Regarding the reviewer's additional comments that "scr treated cells demethylate" in Fig. 4, we would like to note that there is some fluctuation of genomic methylation during cell passaging (reference #66), though it is for the most part, stable. Highly methylated sites remain highly methylated and vice versa. We therefore would not consider these to be off-target effects.

In the case mentioned above - "the lone exception of a single CpG in one clone that displayed 39.5% methylation" - this is referring to a CpG that was less methylated in a single gRNAscr clone than in the remaining gRNAscr clones. We don't believe that this was an off-target effect but think that is more probably caused by clonal variation (bottleneck effects) which happens in cultured cells, because they are heterogenous, rather than off-target binding since gRNAscr has no potential off-target binding site in IL33. CRISPR/Cas9 binding is known to be reduced by several magnitudes by only a few mismatches, and that is with a catalytically active enzyme: for dCas9 to bind strongly to prevent DNMT1 activity through the replication fork with >7 mismatches as is the case here borders on the impossible.

We hope that our genome-wide off-target analysis resolves to some degree the question of the extent of off-target dCas9 activity, though future analyses could certainly benefit from investigating multiple gRNAs in different cell lines from different species.

11. The precision could be further increased using targeted bisulfite sequencing, in order to make the claim below robust, by analyzing the DNA the authors might still have: "We conclude that we were able to produce cell lines with almost completely demethylated target CpGs with this approach (the small level of methylation detected in these clones is around the standard error for unmethylated controls in our pyrosequencing assay)."

The reviewer is correct in noting that the depth of targeted sequencing would be higher. However, the fact that we were able to demethylate some CpGs from >90% methylation to <5% methylation is, to us, indicative of almost complete demethylation. To rephrase what was stated in the text, unmethylated plasmid control DNA commonly is detected at non-

zero values up to 3%. The fact that Pyrosequencing has a highly reliable read depth of in the 10⁶ range as well as the high cost of targeted sequencing led us to use this well accepted method.

12. dCas9 expression was measured as transcript (normalized to GAPDH as in figure legend) but a better correlation is with the Cas9 protein level by FACS (there are Cas9 specific antibodies available). "The clonal analysis suggests a clonal variation in the extent of demethylation by dCas9:gRNA. A plausible cause could be variation in the level of expression of either dCas9 or the gRNA. dCas9 expression did not correlate with methylation levels (Supp. Fig. S3D)"

We agree that our experimental approach was not sufficiently powered to completely rule out the possibility of dCas9 correlation to DNA demethylation efficiency, and the reviewer's suggested experiment could indeed lead to the discovery of some correlation. However, we did find a significant correlation to gRNA expression level, and that discovery led us to then pursue a strategy to increase gRNA expression levels by selection with high puromycin levels. This strategy resulted in effectively complete demethylation of IL33 target CpGs. Therefore, our experiments served their purpose: it is clear that the correlation to gRNA expression level was informative and useful. We are certain however, that dCas9 can be a limiting factor in other experimental setups, and this has been shown in many articles, (and this is why we looked at it at all); it just did not appear to be a factor under the expression levels we achieved with our specific lentiviral strategy.

13. Much of the data presented is very interesting so this point is just a comment on one of the observations. One main difference between epigenetic editing in primary cells and cell lines is that in the absence of the transiently transfected Cas9-DNMT3a fusions, cell lines lose the methylation induced, whereas primary cells seem not to. It is interesting that in this system methylation is not regained which is an important observation. "Interestingly, low levels of methylation persisted for at least 75 days after removal of dCas9 by Cre recombinase (Fig. 5B), indicating a lack of de novo methylation of this locus in these cells and the ability of this approach to modify DNA methylation in a stable manner despite elimination of dCas9." Can the authors hypothesise in the discussion how many such sites might be redundant in any given cell line?

This is indeed a very interesting and important point that cell lines lose methylation induced by dCas9-DNMT3A whereas primary cells apparently do not. This will be critical to keep in mind in any future experiments, and as such we have added the following statement to the discussion on page 23, as suggested by the reviewer:

"The ability of newly demethylated sites to stay demethylated can vary; we detected no increase in IL33-002 TSS methylation 75 days after removing dCas9 by Cre-mediated recombination in NIH-3T3 cells, but saw a small non-significant increase in SERPINB5 demethylated CpGs 45 days after Cre-mediated dCas9 removal. It is useful for research timescales that both sites largely stay demethylated, but the small variation between the two genes in the two cell lines suggests that the retention of unmethylated CpGs may vary as a factor of cell line (e.g. how much de novo methyltransferase activity a cell line has) or by specific CpG sites (e.g. in a growing cell population, how detrimental to cell growth is demethylation of a specific CpG, and will it be selected against?) and thus it will be important in future studies that use this technique to assay how long demethylation persists in the CpG and cell line contexts under examination to ensure that demethylation persists for the duration of the experiments. It is important to note in stem cells where de

novo DNMTs are expressed to a higher level, methylation might be regained after removal of dCas9"

14. Again the finding that dCas9 induced demethylation does not induce gene expression (might be statistically significant but it's small), is interesting and important. "We detected a small but significant ($p=0.0312$) increase in expression in NIH-3T3 cells treated with dCas9:gRNA3 and Cre recombinase as compared to dCas9:gRNAscr, but not in dCas9:gRNA1 or dCas9:gRNA2 cells (Figure 5C). This is consistent with our in vitro/transient transfection luciferase assays findings (Fig 2F); both approaches suggest that methylation of TSS CpGs 9, 10, and 11 silence the basal IL33-002 promoter."

I wonder if this is because this promoter is a CpG poor promoter? Nevertheless, as mentioned above, since methylation of CpG poor promoters seems to be tissue specific, it's important to have this knowledge as precedence that methylation removal will not change gene expression in a significant way.

We agree with reviewer comment. It may very well be because this is a relatively CpG-poor promoter and, as stated before, we are hesitant to make any conclusions as to the behavior of all CpG-poor promoters and hope that our method can be used at a larger scale to answer these interesting questions. We are happy that the reviewer finds this important and interesting.

15. Figure 5F: Not sure if I missed it but what is, quantitatively, the difference in induction of IL33 in cells with highly methylated promoters (scr gRNA) and gRNA3 demethylated and subsequent CRE removed? In other words, how much does the promoter methylation inhibit gene expression? Both in the case of poly(I:C) and LPS?

The reviewer did not miss this; it was our mistake to include these numbers in Figure 5F, but not discuss them in the text. The answer is that a demethylated TSS produces 1.48X more IL33-002 transcript with polyI:C and 2.02X more with LPS than methylated TSS. We have added the following sentences to the manuscript on page 15:

"TSS-demethylated dCas9:gRNA3 cells had a 1.48X higher level of IL33-002 expression than dCas9:gRNAscr counterparts at 8 hours ($p=0.0097$). However, the overall induction within each treatment group (poly(I:C) vs. control) was lower in gRNA3 cells (401X) than in gRNAscr cells (451X), because control-treated gRNA3 cells already have a higher baseline IL33-002 expression as demonstrated here (1.67X, $p=0.1354$) and in Fig. 5C-D."

For LPS treatment, this was discussed without the quantitative aspect, so we added the numbers in the context of the text that was already written.

16. SERPINB5 and Tnf demethylation data is also interesting but again these genes don't have a CpG island, correct? The authors say: "consistent with the conclusion that demethylation of the promoter is insufficient for its expression and demethylation of other regions, such as the depicted enhancer regions (Fig. 6A), is required for induction of SERPINB5". Have the authors tested this hypothesis that enhancers of these genes need to be demethylated? If not, why not?

Yes, the reviewer is correct in stating that neither SERPINB5 nor Tnf have CpG Islands. We have not at this time examined the hypothesis that enhancers must be demethylated in coordination with promoters. This is because the demethylation of large regions would require multiple gRNAs, and we have not yet fully developed a protocol that can allow multiplexing gRNAs for this purpose. Even with 4 gRNAs, which is the largest number of

gRNAs that we have been attempting to optimize, we expect to demethylate maximally ~200 bp, which is still considerably smaller than smallest reported enhancer in SERPINB5 (500bp) (and does not take into account the gRNASERPINB5 that is needed to demethylate the promoter). At present, our technique allows us to ask about the contribution to gene expression of individual CpGs, not large regions, like enhancers or CpG-islands, and as such we focused on individual CpGs in promoters.

17. The FMR gene seems to be the only gene which recapitulates the expected demethylation-re-expression pattern and coincidentally this gene has a CpG island at the promoter region. Question: "We obtained primary fibroblasts from a patient with Fragile X syndrome with approximately 700 CGG repeats exhibiting high methylation [69]" Are these primary cells from the same patient as in reference or a different patient. If a different patient, the authors need to first test the composition and methylation level of the gene promoter.

Yes, these are the same cells from the same patient as analyzed in the referenced paper, obtained directly from the supplier. We have edited the text to try to make this clearer.

18. The authors say that they could not measure methylation levels in the promoter region of FMN due to technical reasons: "We were unable to map DNA methylation in the dCas9 demethylated pools due to technical challenges in amplification of such a highly repetitive and CG-dense region, despite attempting to do so with buffers and polymerases optimized for GC-rich templates, as well as bisulfite-free approaches in an effort to avoid the decreased sequence diversity caused by sodium bisulfite (C to T transitions)." Couldn't the authors replicate the methods used in the reference cited 69, looks like they submitted samples to <https://www.epigendx.com/>?

Unfortunately, the strategies in this paper are not useful options for us. The EpigenDX assay that the reviewer is referring to here is a pyrosequencing assay for the PROMOTER of FMR1. The CGG repeats which we are targeting with our dCas9 method and CGG gRNA are located in the 5' UTR, far enough from the promoter that we don't expect any changes in the promoter when dCas9 is targeted to the CGG repeats (based on our distance analyses in Figures 3 and S2). The authors of the cited study do claim to analyze the methylation of the CGG repeats with the Asuragen FMR1 PCR kit, however, upon closer inspection, the method uses methylation-sensitive restriction enzymes whose target sequences lie outside of the CGG repeat region itself, making this approach also unfit for analyzing the methylation status of our targeted region.

Alternatively, the authors could do MeDIP-qPCR with several primer pairs, or Nanopore? Perhaps they could confirm that there is increased expression of FMR1 after 5-aza treatment, to confirm that it is demethylation dependent?

Thank you for these suggestions. Due to the limited sensitivity of MeDIP-qPCR, particularly in this hard-to-amplify region, and our inexperience with Nanopore technology (and knowledge of its limitations), we opted for several alternatives.

First, as suggested by the reviewer, we performed 5-aza treatment of these cells and observed a large but statistically insignificant (interestingly, the high variance is similar to that of the activation we saw with the dCas9 method) activation of FMR1 upon 5-aza treatment. We have added this figure as Supp. Fig. S10. We also included in the relevant passage of the manuscript references to five additional studies that demonstrate FMR1 activation by 5-aza. Second, we attempted to see if we could confirm FMR1 demethylation

by whole genome bisulfite sequencing, but unfortunately, we failed to obtain sufficient high-quality reads mapping to this repetitive region. Therefore, we are still unable to quantify methylation at the FMR1 locus.

19. This is quite unfortunate that the authors couldn't measure their DNA demethylation level on a promoter sequence that seems to be the only gene in this study truly responsive to DNA demethylation. This further strengthens the assertion that it's really CpG island promoter containing genes that are the most influenced by DNA methylation.

We agree with the reviewer that this is a likely possibility, but our data lacks the comprehensive analysis of probably hundreds of genes needed before such a conclusion can be made, but we hope the tool we present herein can be used in the future to begin to elucidate this possibility.

20. Demethylation of the HNF4 gene upon crispr editing of the DNA sequence is very interesting but can one broadly conclude that extensive promoter demethylation upon catalytically active Cas9 editing happens based on one case? Moreover, it is off-topic to some extent, not sure how is it related to the rest of the figures. This data is good as a preliminary look at possible demethylation events that occur after Cas9 editing. It looks like even the unedited clones in the gRNAHNF4A (4 of the strands shown in Figure 7C) are completely unmethylated, whereas in the scrambled ones they are majority methylated. Perhaps I misunderstand and all the gRNAHNF4A clones have been edited? Can the authors explain this data? Nevertheless, I find it off-topic and would be best to expand the dCas9 demethylation method on other CpG rich promoters, some pathological (many are known) or germline like Dazl for example.

Please note this is now Figure 8 due to addition of a new Figure 6.

We agree that the single case presented in this paper does not demonstrate that this applies to all or even most cases of Cas9 editing. We do not claim it to be extensive. However, it is still a demonstration that this can and does happen. Therefore, it serves as a warning for researchers to consider, given that the effects are the exact opposite of those intended in CRISPR knockouts. To reiterate, we do not claim that this is a constant phenomenon in Cas9 editing, only a possibility that ought to be considered, and we feel that we have provided an example that it can happen by showing it in this locus.

Yes, it was also interesting to us that even unedited strands of DNA are unmethylated. We believe that one explanation for this may be because they were indeed cut, but the other allele was used for HDR and they were correctly repaired.

We agree that the data is therefore preliminary, but we still consider it to be a direct logical step from the data presented in the rest of this paper – that the CRISPR/Cas9 system can cause demethylation by tight binding to DNA – and it is consistent with all of the experiments performed herein that demonstrated the demethylation potential of the CRISPR system. We also find that the consequences can potentially be very important, especially since CRISPR/Cas9 mediated gene editing is so popular and is considered a “clean” method for gene editing, and that this can produce the exact opposite of the intended affects.

Given that one other reviewer stated that this experiment was valuable, we are hesitant to remove this section from the paper. We hope this reviewer can agree with us about the importance of demonstrating this confound, even if other work is necessary to determine how extensive this may be.

At this time, evaluation of the impact of demethylation of an entire CpG-rich promoter will require further methodology development due to the fact that we have optimized a one-gRNA system that can demethylate a region of less than 50bp. We strongly agree with the reviewer that analyzing an entire CpG-rich promoter would be fascinating and something we very much look forward to developing the techniques for based on this method.

XXXX

Minor point:

Interesting data, but it would be good to have a figure to illustrate this: 'The results are in accordance with the crystal structure of the dCas9:gRNA:DNA triplex, which reveals minimal 5' protrusion of dCas9:gRNA beyond the 5' end of the gRNA target and more pronounced extension of both dCas9 protein and, to a larger degree, gRNA scaffold beyond the 3' end of the gRNA target sequence [48]' Similarly, the figure may help clarify this part: 'Interesting, the 3' footprint is smaller by at least 2bp (and at most 6bp) than in the strand interacting with the gRNA, as CpG 19 is not affected on the antisense strand.'

Thank you, for this suggestion. We have now added a figure (Supplementary Figure S2A) of the cryo-EM structure of Cas9:gRNA bound to target DNA and highlighted the nucleotides in the DNA complementary to the 5' and 3' ends of the gRNA target sequence. We hope this makes this point much clearer. Unfortunately, this figure cannot clarify the second part here because (1) the DNA strands are not resolved long enough into the 5' and 3' ends to be able to analyze this (in fact, we had to manually add two unresolved nucleotides to the 5' end of the target DNA strand and gRNA in order to predict where they would be located) and (2) the strand not interacting with the target strand (the non-target strand) was resolved poorly overall, with only a structure of the 3' end.

Reviewer #2 (Remarks to the Author):

In this manuscript, Sapozhnikov and Szyf reported to use a catalytically dead Cas9 to induce targeted DNA demethylation at several genomic loci in dividing cells. They first compared the demethylation effects by dCas9-Tet1 and dCas9-dead Tet1 at the II33 locus in NIH3T3 cells. Then they showed a dead Cas9 can prevent the occupied DNA being methylated in vitro and induced a moderate demethylation in cells at the II33 locus. The efficacy of induced demethylation can be enhanced by passaging the targeted cells or subsequent cloning. They further showed that targeted demethylation by dCas9 at the II33, SERPINB5, Tnf loci only achieved a small or no elevation in gene expression. Only demethylation of the CGG repeats at the FMR1 locus induced a significantly upregulation of FMR1 expression. Last, they described an untargeted DNA demethylation during CRISPR/Cas9 mediated gene knockout. Overall, the topic of this study is of great interest aiming to address the critical question on correlation vs. causality of DNA methylation at the promoter regions. However, some conclusions are either overstated or not supported by their current design and data. The following points need to be experimentally addressed to approve the quality of this manuscript before it can be recommended to be accepted for publication.

Thank you for reviewing the manuscript and for the helpful and critical comments that have certainly revealed and allowed us to address the shortcomings of the manuscript. Please find below a point-by-point response to every comment.

1. In Figure 1, the authors reported a surprise observation that dCas9-dead Tet1 was able to induce a significantly higher elevation of II33 expression compared to dCas9-Tet1. To support this conclusion, the expression levels of dCas9-Tet1 and dCas9-dTet1 in these experiments need to be carefully examined by qPCR and western blot.

Thank you – this is a critical suggestion. We performed the suggested RT-qPCR – which shows a 1.2X (statistically significant) increased expression of dCas9-deadTet compared to dCas9-Tet. On the other hand, western blot showed no significant difference between the levels of the two proteins.

Since the data is not particularly clear, we changed this wording to remove the emphasis on the relationship between dCas9-deadTet and dCas9-Tet. The important finding here is not that dCas9-deadTet activates a target gene more than dCas9-Tet, but that dCas9-deadTet can activate a target gene at all, and to a high level, despite lacking a catalytic capacity. This is important because it questions the idea that “active DNA demethylation” by Tet is causal to activation of gene expression.

The revised text reads on page 5: “Equally surprising was the fact that that dCas9-deadTET was also effective in transactivation of IL33-001 (Fig. 1I) ($p < 0.0001$ for all targeting gRNAs). The IL33-001 transcript was also significantly more expressed in dCas9-deadTET cells under gRNA1 ($p = 0.0091$) and gRNA3 ($p = 0.0033$) as compared to catalytically active dCas9-TET, though it may be caused by different level of expression of the constructs; although dCas9-deadTET expression levels were moderately higher than dCas9-TET by RT-qPCR (Supp. Fig. S1C), the protein levels as determined by a western blot analysis were not different (Supp. Fig. S1D-E).”

2. It is odd to use Tet2, but not Tet1, to demonstrate Tet protein can induce DNA methylation independent transactivation in HEK293T cells. To be consistent with the previous experimental setting, Tet1 and NIH3T3 cells are needed to be used.

Thank you for this suggestion. We have replaced this figure with TET1 data generated using Tet1, which had similar effects, as well as with mutant TET1. This TET2 data has been moved to supplemental (S1M) and all figure legends and methods have been updated accordingly.

3. In Figure S3D and S3E, without a statistically analysis, it is not conclusive that sgRNA expression but not dCas9 expression level correlates with the demethylation level given the small number of samples (less than 10).

Thank you. We have added additional details concerning statistical analysis into the text on page 11:

"dCas9 expression did not correlate with methylation levels ($r=0.1982$, $p=0.6091$, $n=9$) (Supp. Fig. S3D) whereas gRNA3 expression levels correlated negatively with methylation ($r= -0.7307$, $p<0.05$, $n=9$) (Supp. Fig. S3E)."

We have also modified the figures S3D, S3E, and S3G to include lines of best fit as well as the r-values and p-values to better illustrate the statistical analyses.

We hope this reaffirms that there are statistical analyses to support these conclusions. However, we would also like to note that we do not suggest that dCas9 expression cannot be a limiting factor: only that it was not in our experiments with the expression levels achieved by our lentiviral and antibiotic selection strategy. The fact that the sgRNA was the limiting factor in our experiments was further supported by the fact that when the antibiotic (puromycin) was increased in order to select for cells with increased sgRNA expression (with puromycin resistance gene) we observed dramatic increases in DNA demethylation efficiency (Figures 4F,4G). Therefore, even if the analyses were underpowered to completely rule out dCas9 correlation due to sample size, the significant correlation to gRNA expression was extremely useful in guiding our optimization and these experiments thus served their purpose.

4. Regarding the potential off-target effect by dCas9, assaying four candidates is far away to be conclusive. A genome wide approach such as whole genome bisulfite sequencing is required to address this concern.

Thank you, for this excellent suggestion. We have therefore now performed whole-genome bisulfite sequencing as recommended by this reviewer and other reviewers. We have now added an entire section, titled "dCas9 off-target demethylation events and comparison to dCas9-TET" with a new Figure 6. In addition to comparisons with dCas9-TET, this section includes exactly this: a genome-wide analysis of all hypomethylated CpGs in dCas9:gRNA3:Cre compared to dCas9:gRNAsc:Cre in an effort to evaluate the off-target concern. Briefly, we find that while there are 641 off-target hypomethylated CpGs, we find that they do not align to gRNA3 and are thus likely the result of other manipulations (lentivirus, passaging, etc.) that are not the dCas9:gRNA binding activity.

5. Based on the proposed mechanism for dCas9-induced DNA demethylation, post-mitotic cells such as neurons are not editable. At least one locus is needed to be tested by dCas9 in neurons such as the FMR1 locus.

We agree that the reviewer's suggestion will provide conclusive proof for the proposed mechanism, however we feel that this will be beyond the scope of the methodology developed in this manuscript and since we have already provided certain lines of evidence in the current study (detailed below). Applying the method developed and optimized in this paper to neurons will be particularly difficult since we require high gRNA and high dCas9 expression (lentiviral) that requires selection with antibiotics. Such selection would be impossible in nondividing cells.

In addition, the following points are consistent with our model:

- **dCas9 is a catalytically dead protein (and is widely used, so this is well-established) with no enzymatic activity**
- **dCas9 is known to sterically interfere with protein binding of DNA (e.g. RNA polymerase II: this is the basis of the technique called CRISPR interference)**
- **we can reconstitute *in vitro* the mechanism by adding dCas9 and gRNA to block methylation by the methyltransferase M.SssI of specific CpGs in a plasmid. This only works if dCas9 and gRNA are added before M.SssI (Fig. 2 A-E) – if added after, dCas9 and gRNA cannot remove methylation marks that have already been deposited (Fig. 2G)**

Reviewer #3 (Remarks to the Author):

This is a novel study where the authors develop and test, from many angles, a new methodology to study *in vitro* and *in vivo* site specific CpG methylation/demethylation. They use an enzyme-free CRISPR/dCas9-based system for targeted methylation editing that achieves selective methylation *in vitro* and passive demethylation in cells through steric interference with DNA methyltransferase activity. They map the size of the region of interference, optimize the system for improved demethylation of targeted CpGs without detectable off-target effects within the locus, and analyze the transcriptional consequences of demethylation of different genes known to be highly activated by 5-Aza treatment. They perform, side by side, their method using dCas9, and the published method using dCas9-TET1 on the I133 locus to point to the improvements in specificity and the ability to methylate/demethylate single CpG sites.

In general, the claims are convincing and appropriately discussed in the context of previous literature and the work is technically sound. Only their claim that this model does not introduce demethylation on off-target regions would have been better supported by whole genome methylation analysis. They show however that with this method they modify the methylation of unique CpG sites on the I133 promoter without affecting nearby CpGs as a proof of specificity; this level of specificity was not achieved with the commonly used dCas9-TET1 approach used here as control.

The findings will be of great interest to others in the field of DNA methylation since, due to limitations in specificity, new methods to study methylation are continuously searched for. The paper will also inform the CRISPR/Cas9 field, since they show marked demethylation around the TSS of a CRISPR/Cas9 targeted gene that results in altered transcription of the mutated gene. So they suggest that care should be taken in the interpretation of results from all CRISPR/Cas9 editing experiments that target regions close to the TSS and consider whether the simple presence of the Cas9 protein on the DNA was blocking, by steric hindrance, the re-methylation around the targeting region resulting in unexpected results.

The manuscript is clearly written but they **should remove some discussion from the Results part.**

For most of the experiments the statistical analysis seems fine. There are some where the statistical significance is not marked on the panels (asterisks), and others where it is not clear which groups are being compared in the statistical analysis. Some experiments in the supplemental material have an n=1. If they are not essential they should be removed from the manuscript or if they want to retain the information they should increase the n. The particular experiments are indicated in the Specific Comments.

They provide sufficient methodological detail that the experiments could be reproduced. However, they need to modify some information in methods to be more explicit about the structure of the plasmids used.

We thank the reviewer for this succinct summary and analysis and are truly excited to see that the reviewer has found this work novel, interesting, clear, and convincing. We are also very grateful for the constructive and well-thought out comments that have undoubtedly improved the quality of the manuscript. To that end, we have made some effort to remove discussion elements from the Results section as recommended above and have provided answers to the specific comments below.

Specific Comments

Results

CRISPR/TET-based approaches confound the causal relationship of DNA methylation and transcription:

Paragraph 1. Does the treatment with 5'aza also reduce methylation of CpGs 1-5? Why only 9-11 are shown?

Only CpGs 9-11 were originally shown because this was meant only to be a validation to confirm the drug was working and verify demethylation of the IL-33 transcription start site by 5-Aza. However, we appreciate the value of providing data as to the other studied CpGs and have now added this information to Figure 1B. The answer is yes, 1 μ M 5-Aza significantly reduces methylation of CpGs 1, 2, 3, and 5.

Paragraph 3. Although they say that the "differences in demethylation were significant", they do not indicate p values in the text or in the Figure 1E-G legend.

This is a good point. We have corrected Figure 1E-G to display all p-values and included a mention of these p-values in the relevant text.

They also suggest that "TET1 would interact with several DNA targets across the genome by their cognate DNA binding domain in the absence of any additional targeting." However they do not consider that the gRNAs or the dCas9 itself could introduce a low degree of off-targeting effects.

Yes, the reviewer is correct that we did not mention it in this section, largely because this is already well established and not part of our Results. Still, we agree that this needs to be mentioned, we therefore added to the text on page the following sentence that the reviewer suggested to the results section on page 5:" . It stands to reason that, in addition to the known off-target potential of gRNAs and the CRISPR system, overexpressed proteins would interact with several DNA targets across the genome by their cognate DNA binding domain in the absence of any additional targeting".

Paragraph 4. It would have been more informative and would have allow a stronger conclusion about specificity if they would have tested the whole-genome changes in methylation patterns caused by dCas9-TET1 vs. their method with dCas9.

We agree with the reviewer that this would be highly informative and, as such, we have performed the recommended whole-genome analysis by whole-genome bisulfite sequencing. These new data are presented in the new section in the results, titled "dCas9 off-target demethylation events and comparison to dCas9-TET" with a new Figure 6.

In brief, the WGBS results were consistent with the data for IL33-002/IL33-001 that the reviewer is referring to here: we provide evidence of global genomic hypomethylation in response to dCas9-TET expression. We also fail to find any statistically differentially methylated CpGs between dCas9-TET:gRNAsc and dCas9-TET:gRNA3. The dCas9 method, in contrast, produces levels of demethylation of IL33 that are significant even genome-wide

(the two most demethylated sites are targeted CpGs 10 and 11). We report 641 significantly hypomethylated CpGs that are "off-targets" of dCas9, but we also show that these sites have insufficient sequence similarity to be considered as potential sites of dCas9:gRNA3 binding and therefore likely result from other confounding variables, such as differential lentivirus integration or bottleneck effects during cell passaging.

Paragraph 6. Why they use a pcDNA-TET2 plasmid to show the transcriptional activity of TET proteins when they have performed all the other studies with TET1.

As the reviewer has requested, we have now repeated the study with a TET1 expression plasmid – including a TET1 variant with a catalytic mutation - and replaced this figure (Fig. 1J) with the new TET1 data, which had similar results, as well as with mutant TET1. This TET2 data has been moved to supplemental (S1M) and all figure legends and methods have been updated.

Paragraph 7. Are there other publications which show that TET proteins can act across large genetic distances. A reference must be included.

While we could not find examples of different dCas9-TET systems where the authors analyzed DNA methylation at a long distance from the gRNA binding site, it is important to note two things here. First, the TET family is known to bind enhancers and facilitate long-range chromatin interactions. We have added a sentence and references to state this, as the reviewer suggested on page 5 in the results section: "This is aggravated by the fact that the TET family of proteins is known to participate in enhancer regions and facilitate long-range chromatin interactions[41, 42]". Second, the long-range effects of TET may be completely independent of the gRNA binding, and therefore may not be long range interactions at all, but instead bound by dCas9-TET through the TET domain. This can be elucidated experimentally, but we believe it is beyond the scope of our aims here.

A novel method for site-specific DNA methylation in vitro:

Paragraph 1. They state that "since dCas9 is a prokaryotic protein that is unlikely to interact with other proteins in the gene transcription machinery and has no enzymatic activity epigenetic or other, there should be limited confounding factors to DNA demethylation." They are not providing proof of this statement. So they need to include references.

We agree with reviewer that there is no direct evidence that dCas9 does not interact with any eukaryotic protein. However, there are no reported interactions of dCas9 with any human proteins in the literature. We can reference the fact that the protein does not appear to bear any protein:protein interaction domains. Therefore, we have revised this sentence to reflect these suggestions of the reviewer on page 7 of the results section: " dCas9 is a prokaryotic protein with no documented protein:protein interaction with eukaryotic gene transcription machinery, the protein has no homology to known eukaryotic protein:protein interaction domains and has no enzymatic activity epigenetic or other [48]."

Paragraph 2. Please, indicate that the CpG - free vector is referred as pCpGI, next to the reference.

Thank you, we have added this.

Paragraph 4. "These data demonstrate the exquisite impact of site-specific methylation rather than just methylation density, and thus this assay appears to capture the sequence specificity of inhibition

of promoter function by DNA methylation." Earlier work used to mutate individual CpG sites to evaluate the importance of each site. They should comment about the benefits of their method compared to individual CpG mutation.

This is a good point; as the reviewer requested, we have added comments about the earlier mutagenesis strategy and the benefits of our method. However, we felt this was best placed in the Discussion on page 23:

"This method has advantage over earlier methods that protected individual CpGs from methylation by mutagenesis to non-CpG sequences[108], since mutagenesis can disrupt protein:DNA interactions by the sequence change rather than by the methylation difference. Our method alters the methylation per se without disrupting the genetic sequence."

The dCas9 system directs robust site-specific demethylation in living cells:

Paragraph 3. "... there was detectable significant downregulation of IL33-001 under gRNA1." Have you check the methylation status of IL33-001?

We have not examined the IL33-001 promoter by targeted bisulfite sequencing, but with WGBS there were no significant differences in the IL33-001 promoter.

Are there potential off-targets for gRNA1 close to IL33-001?

There are no particularly convincing off-targets for gRNA1 in IL33-001. Below, please find a table showing ranked (top=best) alignments of gRNA1 against +/- 1000bp from the IL33-001 TSS (which in this table is at 16,018). The table was generated with CCTop, as described in our manuscript in the new WGBS data section. We have also added this observation to the relevant text.

Chromosome	start	end	strand	mismatches	target_seq	PAM	alignment
IL33-001_temp	16602	16624	+	12	TTCAATAGATCTTCATGCGC	TGG	----- PAM
IL33-001_temp	16960	16982	-	10	CAAGCAAGGATGACTTGAGG	AAG	- - - - - - PAM
IL33-001_temp	16202	16224	-	8	AAGCAAATGTCATCATGAGT	TAG	- - - - PAM
IL33-001_temp	16887	16909	+	13	TACTGTGGTCACGCAGGAGC	CAG	- - - - - - - - PAM
IL33-001_temp	15092	15114	+	10	ACTACAGATTTTATTGTCC	TGG	---- - - - - PAM
IL33-001_temp	15519	15541	+	12	AAGGTGAGGGGCAGAGGAGC	TGG	- - - - - - - PAM
IL33-001_temp	16296	16318	+	12	GCCGTGAAACTTTACAAAGC	AAG	- - - - - - PAM
IL33-001_temp	15404	15426	+	12	ATGAAGGTTGCAAATAAGC	CAG	-- - - - - - - PAM
IL33-001_temp	15704	15726	-	13	CTTGCCTTAGCCTCCAGGGC	TGG	---- - - - - - - PAM
IL33-001_temp	16979	17001	-	12	CAGTGGTGGACCACAGTAGC	AAG	- - - - - - - PAM

"We also assessed whether dCas9:gRNA3 caused demethylation of the top 5 predicted candidate off-target CpGs for gRNA3". Why only with gRNA3 and not the other 2?

During the course of this study, we placed emphasis on gRNA3 because it was the gRNA that demethylated the IL33-002 transcription start site and had minor effects on gene expression and inducibility by 5-aza (Figure 5). Because of its consequences on expression, we used only gRNA3-treated cells for many analyses, such as poly:IC experiments, ChIP, etc. We recognize that a more thorough off-target analysis of this system would use multiple gRNAs, across multiple cell lines and species, and analyze this genome-wide, and hope to do this in the future. We have added to our Discussion that this is a limitation on page 22 of the discussion section:

“We also find that there appear to be no off-target DNA demethylation events as a consequence of gRNA:dCas9 off-target sequence similarity, but we recognize that further work is needed to identify the biological origin of DMRs and to evaluate off-target effects of multiple gRNAs across multiple cell lines and species.”

Were the corresponding targets affected in the dCas9-TET1 experiments?

We did not analyze dCas9-TET with gRNA3 at these candidate off-target sites by targeted bisulfite pyrosequencing. So, in an effort to answer this question, applied our WGBS data to see if (1) these top predicted off-targets were affected and (2) all predicted off-targets (up to 4 mismatches and 1 gap) were affected.

None of the top predicted off-target sites were DMRS (significantly hypomethylated, $q < 0.01$, methylation difference $> 25\%$, 5X minimum coverage in all samples) in either dCas9-TET:gRNA3 or dCas9-TET:gRNAscr.

However, of all predicted off-targets of up to 4 mismatches and 1 gap (100bp around cut site), 21 and 5 were within 100bp of DMRs in dCas9-TET:gRNA3 and dCas9-TET:gRNAscr, respectively.

We have added the following revised text into the manuscript in order to provide this information:

“Of all regions containing predicted gRNA3 off-targets of up to 4 mismatches and 1 gap (100bp around cut site), 21 and 5 were within 100bp of DMRs in dCas9-TET:gRNA3 and dCas9-TET:gRNAscr, respectively.”

The effect of site-specific demethylation on Il33 gene expression:

Paragraph 1. Mislabeled Figure 3G change to S3G.

Thank you, we corrected this and another nearby similar instance.

Paragraph 3. “One of the two base substitutions to render this dCas9 variant nuclease-dead (D10A, H840A) is different than the dCas9 used in previous experiments (D10A, N863A).” Why do they use a different dCas9 sequence? Is it to avoid potential cre cleavage of dCas9? Please explain.

In these experiments, we had to introduce by lentivirus a dCas9 sequence that was flanked by loxP sites so that it could be excised by Cre recombinase. Rather than adding loxP sites ourselves to the original lentiviral dCas9 plasmid, we opted to purchase from Addgene a popular dCas9 plasmid with loxP sites. It so happened that one of the two mutations used to turn Cas9 into catalytically inactive dCas9 by this group of authors was different from those used on our original un-floxed dCas9 plasmid. This has no known functional impact on the ability of dCas9 to bind target DNA, and both dCas9 variants clearly worked to demethylate targets. There is no underlying rationale for the different mutations, nor did we have any strong reason to include this fact other than to accurately report our experiments and perhaps to show that different dCas9 variants can be used in this methodology. We also performed these experiments prior to the ones with dCas9 from *S. aureus*, which is an entirely different protein from a different species but is still able to block methyltransferases, again showing that many forms of dCas9 can be used.

"Chromatin immunoprecipitation of cells from Fig. S6E with antibody for dCas9 (anti-Flag) followed by qPCR using primers surrounding the I133-002 TSS." Please indicate that the dCas9 plasmid used has a 3xflag 5' to the dCas9 sequence.

Thank you, we added this clarification to the figure, the figure description, and the Methods section.

"ChIP-qPCR demonstrated elevated dCas9 binding to the I133-002 promoter region only in cells stably expressing dCas9 and gRNA3 but not in dCas9:gRNAsc cells regardless of Cre treatment (Supp. Fig. S6F). The figure legend indicates "dCas9 binding is expressed as percent input (n=1)." This ChIP is a critical control to support that the dCas9 is bound or not to the promoter region of these genes. So it should be performed more than once to assess its statistical significance.

We agree that this is critical to the data and that n=1 was not sufficient. We have repeated the experiment for a total of n=3, with the same results, now with statistical significance and revised Figure S6F and its legend accordingly .

"...low levels of methylation persisted for at least 75 days after removal of dCas9 by Cre recombinase (Fig. 5B), indicating a lack of de novo methylation of this locus in these cells and the ability of this approach to modify DNA methylation in a stable manner despite elimination of dCas9."

Is this finding generalizable to other loci. How long the demethylation persisted in the other experiments tested like in SERPINB5, Tnf and FMR1?

Since Tnf was demethylated in the same NIH-3T3 cells as IL33 and since we were unable to accurately quantify FMR1 methylation, we analyzed whether DNA demethylation remained in human MDA-MB-231 clones with demethylated SERPINB5. We have added the data as Supplementary Figures S8B-C and have added the following sentence to the text in the section about SERPINB5:

"Methylation levels of 3 gRNAsc clones and 3 gRNASERPINB5 clones remained constant for at least 45 additional days of passaging, though there appeared to be a small non-significant trend of increasing methylation in demethylated gRNASERPINB5 clones (Supp. Fig S8B-C)."

Paragraph 4. "It is possible that the small magnitude of induction of expression by demethylation of the TSS region can be explained by the presence of other methylated regulatory regions or other required trans-acting factors that need to be demethylated to facilitate larger changes in expression". Have you checked if they have different binding of polII?

While we did not check polII binding, we did address the question of whether other methylated regions in cis or trans are involved in expression of IL33 outside the region that was targeted by using a global demethylating agent 5-aza and saw no additional demethylation, suggesting that there no other regions that are controlled by DNA methylation and silence the expression of IL33. Still, it is formally possible that other methylated transcription factors that were not demethylated by 5-aza are required for expression of IL33. To address this important point of the reviewer the following paragraph was revised on page 22 in the discussion section:

"First, the fusion of TET to dCas9 is flexible and may allow access to DNA in a wider region, perhaps inducing demethylation in other regulatory regions with putative polII binding sites that are methylated or at methylated transcription factors that are required for more robust expression (Fig. 1 E-G). However, treating cells that have been demethylated at the I133-002 TSS CpG sites 9-11 with 5-2' deoxy-azacytidine doesn't further induce the gene,

while cells that were methylated at 9-11 sites are induced to a level similar to the levels achieved by dCas9. This suggests that the main regulation by DNA methylation occurs at CpGs 9-11 but that the gene is further induced by DNA methylation independent mechanisms"

Paragraph 7. "These data are consistent with the hypothesis that methylation silences basal promoter activity but does not affect inducibility." This concept is only relevant to this locus and it is not necessarily generalizable.

We agree with this comment and have modified this sentence according to this suggestion:

"The data are consistent with the hypothesis that at this locus methylation can silence basal promoter activity but not affect inducibility."

Discussion

Paragraph 11-12. "Indeed, we did not detect off target demethylation, in contrast to the dCas9-TET fusion construct." This is true only for the loci tested in this work. Because you do not test genome-wide changes in methylation, as you indicate in the next paragraph: "Though our off-target analyses could be augmented by more comprehensive analyses (whole-genome bisulfite sequencing), our initial observations suggest no off-target demethylation by this approach."

Thank you, this is true. Indeed, as the reviewer has requested, we performed a whole genome bisulfite sequencing and discuss the results in a new section. However, we have now added the whole-genome analysis as we mentioned in a previous comment. Still, we decided to delete this entire sentence because it leads to some slight confusion. Instead, we replaced it in the next paragraph with the following discussion of our off-target results:

"Since inhibition of DNA methylation is dependent on tight binding of dCas9 which is also dependent on gRNA target and quality, the risk for nontargeted demethylation is low. Accordingly, we find that there appear to be no off-target DNA demethylation events as a consequence of gRNA:dCas9 off-target binding in WGBS data and in targeted sequencing of 5 candidate off-target regions, but we recognize that further work is needed to identify the biological origin of the DMRs that were detected in WGBS analysis and to evaluate off-target effects of multiple gRNAs across multiple cell lines and species."

Methods

gRNA design and synthesis. unformatted reference
([dx.doi.org/10.17504/protocols.io.dwr7d5](https://doi.org/10.17504/protocols.io.dwr7d5))

Thank you, we corrected the formatting of this reference.

Please indicate which elements are included in the EcoR1-flanked gRNA scaffold.

Thanks for pointing out this unclear description, we corrected the description for further clarification on page 26 of the methods section; "Briefly, 455bp double stranded DNAs containing the human U6 promoter, gRNA sequence, gRNA scaffold, and termination signal were ordered as gBlock Gene Fragments (IDT). These were re-suspended, amplified with Taq Polymerase (Thermo Fisher Scientific) according to manufacturer protocol and using

primers listed in Supplementary Table S2, extracted from an agarose gel with the QIAEX II Gel Extraction Kit (QIAGEN), and inserted into pCR®2.1-TOPO (Thermo Fisher Scientific) by incubating for 30 minutes at room temperature. The gRNA scaffold was now flanked by EcoRI sites from the vector. A lentiviral backbone was obtained from Addgene (pLenti-puro, Addgene #39481) and the CMV promoter was removed to prevent aberrant transcription by digesting the plasmid with ClaI-HF and BamHI-HF (NEB), gel extracting, removing DNA overhangs with the Quick Blunting™ Kit (NEB), and circularization with T4 Ligase (Thermo Fisher Scientific) for 1 hour at 22°C. The resulting promoterless pLenti-puro plasmid was then digested with EcoRI and the 5' phosphates were removed with Calf Intestinal Alkaline Phosphatase (Thermo Fisher Scientific) to facilitate efficient ligation of the EcoRI-flanked gRNA scaffold.”

Supplementary Table S5

Very high CTs for most genes and n=1. It is difficult to interpret any fold changes.

Note: this table is now Supplementary Table S10.

This data is actually presented as averages for triplicate drug treatments (n=3). We failed to state this and have now made changes to the table and description to convey this information. We agree that many Ct values are high, but the purpose of this experiment was not to catalogue responses of various genes to 5-Aza-2'-deoxycytidine, rather to identify potential candidates that display strong responses to the drug. Ultimately, we concluded from this data that the *Tnf* gene is one such candidate. This gene had acceptable Ct values and we replicated the strong response to the drug over many independent experiments throughout the course of this work. Therefore, this preliminary search was effective in identifying a candidate gene. We do agree that it is difficult to interpret most of the fold changes that we did not explore further. We felt that these data are sufficiently important to include as they demonstrate the process that we undertook to chose *Tnf* as a target for dCas9-based demethylation. All genes with high Cts were not explored further.

Figure Legends

In all figure legends please indicate the n for each experiment.

We have added n to figure descriptions where they were missing.

Clarify which bars are being compared to determine p in all experiments.

We have added line symbols or text in the legend to indicate which bars are being compared in all experiments.

Figure 1. Please indicate n for 1B,C and D

Thank you, we have added this to the figure description.

Are differences in %methylation significant in 1E,F and G? Please show asterisks on the figure.

Thank you, we have added the asterisks in the figure. We have also now mentioned some of the significant differences in our Results.

Figure 3. There is a difference in the pattern of methylation between C and E corresponding to the different strands. Why do you consider this pattern symmetrical?

Yes, we agree that the pattern is different. The term "symmetrical" was meant to reflect that both the target and non-target strands of the DNA are protected from methylation (as opposed to just one strand being protected) not necessarily that the exact pattern was identical on both strands. It was an attempt to be concise, but we see the confusion that may arise from this. We have changed the title of the section from "Blocking of methylation by dCas9 is limited to its binding site and is symmetrical" to "Blocking of methylation by dCas9 is limited to its binding site and affects both DNA strands". We hope this term is more appropriate.

Figure 5. Why is the difference in n between C (n=3) and (D-E) (n=4-5) when the experiments in D-E were performed with cells from C? Are these biological or technical replicates?

For Figures D-E we used additional gRNA:dCas9 pools that were not assayed in C. All data presented in this manuscript is based on biological replicates. For these experiments, this involves both dCas9/gRNA infection and independent treatments with the drug. There were technical replicates done at the RT-qPCR level, which were averaged and presented as one data point for each biological replicate. This is now clarified in the revised methods section on page 28 in the end of the "Cell culture" section.

How did you determine which were relevant in the maximal induction experiment?

In this study we used gRNA3 cells because we were specifically asking the question whether demethylation at the TSS enables or augments *IL33* induction by extrinsic inducers.

In panel H how many n you selected for each maximal induction and methylation?

The n for expression is 3-4, and the n for methylation is 3-6. Panel H is a summary figure of the data presented for each treatment in the relevant sections, so the n for each can be found in the figures for that particular experiment. We have modified the figure description to clarify this point.

Figure 6. Which CpG sites are you targeting in Maspin?

We believe that there is some confusion stemming from our inconsistent terminology: maspin is the name of the protein produced from the *SERPINB5* gene. The *SERPINB5* targeted CpG sites are shown in Figure 6A. We unfortunately used the two interchangeably and have now revised the figure and the manuscript to replace all instances of "maspin" with "*SERPINB5*", including a renaming of gRNAmaspin to gRNASERPINB5.

Please note this is now Figure 7 due to addition of a new Figure 6.

What is the purpose to show the same data in two formats in B and C?

We believe that it is important to show both formats. The first format in Figure 6B more clearly demonstrates that the treatment itself (lentiviral dCas9 + lentiviral gRNA) results in significant demethylation of the *SERPINB5* promoter in MDA-MB-231 cells. The second format in Figure 6C demonstrates the variability of the treatment at a clonal level and also

shows that there is a fraction of clones that are nearly completely demethylated, despite the population average in Figure 6B being significantly higher.

Please note this is now Figure 7 due to addition of a new Figure 6.

Please number the CpG sites 1-6 in the diagram in 6A.

This is a good suggestion and we added this to the diagram.

Please note this is now Figure 7 due to addition of a new Figure 6.

Why do you subclone the lowly methylated gRNA maspin clones to measure expression?

Thank you for pointing out that the rationale for this experiment is not explained in the text. The purpose of the initial round of cloning is to isolate subpopulations of gRNA maspin cells with sufficient levels of dCas9/gRNA expression to drive considerable demethylation. However, if silencing of maspin (generally considered a tumor suppressor) is necessary for the survival and proliferation of MDA-MB-231 cancer cells, there may be a selection within these clones that favors continued silencing of maspin expression (for example, by histone modifications or transcription factors) and leads to greater levels of proliferation of cells that sporadically limit maspin expression. This has been previously shown in MDA-MB-231 cells, for example by Beltran et al, *Oncogene*, 2007. By subcloning, it may be possible to isolate the slowly growing cells in order to maximize the chance for finding subpopulations of cells with demethylated maspin promoters and also high maspin expression, potentially revealing the direct effects of demethylation without the confound of selection. Of course, this strategy did not result in the identification of such subclones and, as such, is merely hypothetical. It may still be possible that DNA demethylation could activate maspin expression when other silencing factors are absent, but the effect of selection is so strong that these cells simply do not survive.

To address the reviewer comments and clarify this point we revised the text on page 17. "In this case, increasing puromycin had a mild effect in increasing the frequency of unmethylated promoters and even the highest puromycin concentrations (40 µg/mL) resulted in demethylation of only 20% (Supp. Fig S8). We reasoned that perhaps there is a strong selection against cells expressing serpinB5 resulting in overgrowth of highly methylated SerpinB5 cells. Therefore, we turned to the previously described clonal isolation strategy. We picked approximately 20 clones from each of the two treatments (gRNAscr and gRNASERPINB5) and evaluated methylation by pyrosequencing, which revealed a significant demethylation in gRNASERPINB5 MDA-MB-231 clones on average compared to gRNAscr clones (Fig. 7B)."

gRNA-CGG is too GC reach. Would that cause more off-targets effects?

Yes, we agree that the lack of sequence complexity and GC-richness of this sgRNA likely increases its propensity for off-target effects, especially as this gRNA was uniquely designed to bind this specific CGG-repeat target, rather than other gRNAs we typically use whose sequences are chosen more carefully by algorithms that rank on-target efficacy against off-target potential. We obtained this gRNA plasmid via Addgene directly from the article "Rescue of Fragile X Syndrome Neurons by DNA Methylation Editing of the FMR1 Gene" published in Cell by Liu et al (reference #94). In a section of this article, titled the "The Off-Target Effect of dCas9-Tet1 Is Minimal", the authors performed an investigation of its off-target effects by an assessment of genome-wide binding (genome-wide ChIP seq), the methylation status of top 6 candidate methylated loci that overlapped with the ChIP-

seq data, a ChIP-BS-Seq experiment, and RNA-seq data. For example, they pulled down DNA bound to dCas9-TET1/CGG (anti-Cas9 antibody) and analyzed the methylation levels of the bound DNA: "29 loci showed a change of methylation larger than 10%, with the FMR1 locus displaying the most significant methylation decrease (85%)". We hope this answers your question.

Figure S2A. Please re-label site 10 (it is mis-labeled 12).

Thank you. We have made this correction.

Figure S3A. Please add significance asterisks.

Thank you. We have added significance asterisks to this figure.

Figure S6D. Indicate in the diagram that this dCas9 has a 3XFlag 5'. Without it is not clear why do you use anti-flag antibody for the ChIP and other analysis. (pLV hUbc-dCas9-T2A-GFP, Addgene #53191).

Thank you. We have made this clearer in the diagram, in the figure description and in the Methods section by noting that dCas9 bears a 3xFLAG tag. We also repeated this experiment with n=3 (it was n=1 before), as per other reviewers' comments.

REVIEWER COMMENTS

Reviewer #1 (Remarks to the Author):

The authors have addressed all of my concerns point by point, argued their position well and amended the manuscript where needed. Importantly, they are providing data analysing off-target demethylation effects of dCas9 fusions, with appropriate controls and replicates and this will be useful for the wider research community interested in these editing approaches.

The abstract somehow still generalises the conclusions beyond these few genes analysed and the specific sites analysed, although the authors repeatedly wish to restrict their conclusions as such. But the data is clear and thorough, nevertheless, well presented and will add value to the current knowledge in the epigenetic editing and DNA demethylation.

Minor comment regarding the decision to include one of the studies this reviewer mentioned, that tried to disentangle cause-consequence relationship between aberrant DNA methylation and physiological consequence. Citing the study in line 78 is misplaced since the authors talk about Tet based editing: "DNMT3A/DNM3L has also been targeted with dCas9 to 79 methylate and silence p16 in primary breast cells resulting in avoidance of senescence entry[30]."

Discussing this study was meant to counterbalance and add to the challenge highlighted by the authors in Line 54/55: "Yet, these studies also exemplify a fundamental challenge in the field: the persistent inability to

55 attribute causality to a particular instance of aberrant DNA methylation."

I would suggest removing the citation from the section where it's misplaced, in order to maintain the focus around DNA demethylation.

Reviewer #2 (Remarks to the Author):

In the revised manuscript by Sapozhnikov and Szyf, a few new experiments were performed to attempt to address substantial concerns raised by reviewers. However, the new data was not appropriately interpreted and presented. Most importantly, some conclusions remain overstated without experimental support. Below are the specific points.

1. Regarding the off-target effect by dCas9, the authors performed whole genome bisulfite sequencing (WGBS) and identified that "When comparing all treatment conditions to untreated controls, there were 54 DMRs in dCas9:gRNAsc, 338 in dCas9:gRNA3, 3,940 in dCas9-TET:gRNAsc, and 6,286 in dCas9-TET:gRNA3". In the abstract, they concluded that "We find no evidence of off-target DNA demethylation as a consequence of gRNA binding". In fact, they only checked some predicted off-target sites. The dCas9:sgRNA3 binding sites can be experimentally detected by ChIP-seq and the methylation status of these sites should be examined using their WGBS data. Without such an analysis, the WGBS data alone did not support their strong conclusion on no off-target evidence.

2. Another major caveat in this manuscript is all the examined promoters, except the FMR1 promoter, do not contain a CpG island (high density of CpGs). This concern was pointed out by other reviewers as well. For the only CpG rich FMR1 gene, the authors were not able to measure its DNA methylation level due to their technical limitation. In fact, the methylation level of the CGG repeats at the FMR1 locus can be examined with a commercially available kit (<https://asuragen.com/portfolio/genetics/amplidex-mpcr-fmr1/>), and the CpG island at the FMR1 locus can be measured by Pyro-seq (see the cited reference 95 in the manuscript). In addition, the examination of FMR1 expression level in Figure 7J lacked a positive control. It has been reported and confirmed by multiple research groups that Fragile X syndrome patient cells do not express FMR1 at all. Normalization of FMR1 expression in dCas9 cells to the untreated cells with literally zero expression of FMR1 is misleading. A positive control, such as fibroblasts with normal CGG repeats, should be included to calculate the percentage of FMR1 expression in dCas9 cells. In

summary, failure to examine the DNA methylation level at the targeted FMR1 locus and lack of a positive control in FMR1 expression analysis do not support their conclusion "targeting demethylation to the pathologically silenced FMR1 gene targets robust induction of gene expression". This caveat prevents the generalization of their key point stated in the title "Unraveling the functional role of DNA demethylation at specific promoters by targeted steric blockage of DNA methyltransferase with CRISPR/dCas9", and significantly weakened the significance of this study.

3. The nomenclature in this manuscript is chaotic. For instance, the "deltaCas9(dCas9):gRNA" needs to be consistent with the widely used format "dead Cas9:sgRNA" in the field. Another example is DMR. The authors confused about differentially methylated regions (DMRs) vs. differentially methylated CpGs (DMGs). These are two different concepts. When they used DMRs, what they referred to are DMGs.

Reviewer #3 (Remarks to the Author):

The authors have addressed all my concerns in detail, and have performed additional experiments where necessary.

REVIEWER COMMENTS

Reviewer #1 (Remarks to the Author):

The authors have addressed all of my concerns point by point, argued their position well and amended the manuscript where needed. Importantly, they are providing data analysing off-target demethylation effects of dCas9 fusions, with appropriate controls and replicates and this will be useful for the wider research community interested in these editing approaches.

The abstract somehow still generalises the conclusions beyond these few genes analysed and the specific sites analysed, although the authors repeatedly wish to restrict their conclusions as such. But the data is clear and thorough, nevertheless, well presented and will add value to the current knowledge in the epigenetic editing and DNA demethylation.

We apologize if the abstract is still too general – we attempted to make additional changes in order to accurately reflect the conclusions on only the genes under study. Please find below the relevant excerpt from the abstract with additions highlighted and deletions highlighted and struck-through:

Using this new method, ~~we show that in several~~ probe a small number of inducible promoters and find that in these instances, the main effect of DNA methylation is silencing basal promoter activity. Thus, the effect of demethylation of the promoter region in these tested genes is small, while induction of gene expression by ~~different~~ several pharmacological inducers is large and independent of DNA methylation ~~independent~~. In contrast, targeting demethylation to the pathologically silenced FMR1 gene ~~targets~~ elicits more robust induction of gene expression. We also found that standard CRISPR/Cas9 knockout generates a broad unmethylated region around the deletion, which might confound interpretation of CRISPR/Cas9 gene depletion studies. ~~In summary, this~~ Once applied to a comprehensive set of genomic regions, this new method could be used to reveal the true extent, nature, and diverse contribution of DNA methylation to gene regulation. ~~of DNA methylation at different regions.~~

Minor comment regarding the decision to include one of the studies this reviewer mentioned, that tried to disentangle cause-consequence relationship between aberrant DNA methylation and physiological consequence. Citing the study in line 78 is misplaced since the authors talk about Tet based editing: "DNMT3A/DNM3L has also been targeted with dCas9 to

79 methylate and silence p16 in primary breast cells resulting in avoidance of senescence entry[30]."

Discussing this study was meant to counterbalance and add to the challenge highlighted by the authors in Line 54/55: "Yet, these studies also exemplify a fundamental challenge in the field: the persistent inability to

55 attribute causality to a particular instance of aberrant DNA methylation."

I would suggest removing the citation from the section where it's misplaced, in order to maintain the focus around DNA demethylation.

We agree with the reviewer that the reference is misplaced and that the study by Saunderson et al is a strong proof of causality, but are hesitant to categorically present it elsewhere as an unconfounded example of causality due to the facts that the study's reported off-target effects and the potential protein interactions of the employed DNMT3A and DNMT3L domains may contribute to observed transcriptional effects. Therefore, we have done as the reviewer suggested and removed this reference.

Reviewer #2 (Remarks to the Author):

In the revised manuscript by Sapozhnikov and Szyf, a few new experiments were performed to attempt to address substantial concerns raised by reviewers. However, the new data was not appropriately interpreted and presented. Most importantly, some conclusions remain overstated without experimental support. Below are the specific points.

1. Regarding the off-target effect by dCas9, the authors performed whole genome bisulfite sequencing (WGBS) and identified that "When comparing all treatment conditions to untreated controls, there were 54 DMRs in dCas9:gRNAscr, 338 in dCas9:gRNA3, 3,940 in dCas9-TET:gRNAscr, and 6,286 in dCas9-TET:gRNA3". In the abstract, they concluded that "We find no evidence of off-target DNA demethylation as a consequence of gRNA binding". In fact, they only checked some predicted off-target sites. The dCas9:sgRNA3 binding sites can be experimentally detected by ChIP-seq and the methylation status of these sites should be examined using their WGBS data. Without such an analysis, the WGBS data alone did not support their strong conclusion on no off-target evidence.

The reviewer recommended here to perform ChIP-seq for dCas9 binding and analyze these sites for differential methylation in order to support the conclusion of "no evidence of off-target DNA demethylation as a consequence of gRNA binding". This is an excellent suggestion and we have now completed this analysis and it is presented in new Figures 6G-I. To summarize here, we performed ChIP-seq using anti-Flag to capture FLAG-tagged dCas9 in triplicate NIH3T3 cells expressing either gRNA3 or gRNAscr (same cells with demethylated *IL33* used in Figure 5 with no Cre treatment). Comparing gRNA3 to gRNAscr, we found the highest enrichment at *IL33* promoter but found 151 significantly differentially enriched regions, some of which were predicted off-targets and most of which showed sequence complementarity to gRNA3. None of these regions were significantly differentially methylated according to WGBS data. We therefore find that the experiment

recommended by the reviewer provides support to our conclusions. Please find below the new paragraphs in the manuscript that describe the results and methods:

In the Results section on page 15 lines 566-577, we added the following paragraph:

“To test this hypothesis, we performed chromatin immunoprecipitation sequencing (ChIP-seq) with an anti-FLAG antibody in cells expressing FLAG-tagged dCas9 and either gRNA3 or gRNAsc. We found 151 significantly differentially enriched regions (DERs) of dCas9:gRNA3 (FDR <0.05) and 44 DERs in dCas9:gRNAsc (Fig. 6G, Supp. Table S8). The most enriched locus (by fold change) was the targeted IL33 promoter and the summit of the peak (highest fragment pileup and predicted binding spot) was within the gRNA3 target sequence (Fig. 6H). Other DERs included 6 of the 4,436 off-target sites predicted above. Manual analysis of the top 5 gRNA3 DERs (sorted by FDR) revealed considerable sequence similarity to gRNA3, with 100% alignment of a 10-11 bp seed region and PAM (Fig. 6I). Importantly none of the 150 (excluding the IL33-002 DER) gRNA3 DERs overlapped with the 641 DMGs from the WGBS data: the only region demethylated and bound by dCas9 was IL33-002, supporting the specificity of this approach.”

In the Methods section on pages 39-30, lines 1176-1203 we added the following section:

“Chromatin Immunoprecipitation Sequencing and Analysis

For ChIP-seq experiments, cells were prepared as for ChIP and IP was performed with the same anti-FLAG antibody (above) on NIH-3T3 expressing FLAG-tagged dCas9-GFP (selected by FACS) and gRNA (selected under high puromycin); these are the same cells depicted in Figure 5 (transduced with empty vector instead of Cre recombinase). All cross-linking and immunoprecipitation steps were performed with the ChIP-IT High Sensitivity® Kit (Active Motif) according to manufacturer’s instructions using 30 µg input chromatin as quantified by NanoDrop. Sonication was performed as above. Eluted DNAs were sent to Centre d'expertise et de services Génome Québec at McGill University for library preparation and sequencing. Fragmented DNA from 12 samples (three replicates each of gRNAsc anti-FLAG, gRNA3 anti-FLAG, gRNAsc input, and gRNA3 input) was quantified using 2100 Bioanalyzer (Agilent Technologies). Libraries were generated robotically with the maximum volume provided by the client of fragmented DNA (range 100-300 bp) using the NEBNext Ultra II DNA Library Prep Kit for Illumina (New England BioLabs), as per the manufacturer’s recommendations. Adapters and PCR primers were purchased from Integrated DNA Technologies (IDT). Size selection was carried out using SparQ beads (Qiagen) prior to PCR amplification (12 cycles). Libraries were quantified using the Kapa Illumina GA with Revised Primers-SYBR Fast Universal kit (Kapa Biosystems). Average size fragment was determined using a LabChip GX (PerkinElmer) instrument. The libraries were normalized and pooled and then denatured in 0.05N NaOH and neutralized using HT1 buffer. The pool was loaded at 200pM on a Illumina NovaSeq S4 lane using Xp protocol as per the manufacturer’s recommendations. The run was performed for 2x100 cycles (paired-end mode). A phiX library was used as a control and mixed with libraries at 1% level. Base calling was performed with RTA v3.4.4 . Program bcl2fastq2 v2.20 was then used to demultiplex samples and generate fastq reads. Paired-end FastQ files were trimmed for adapters and quality scores using TrimGalore v0.6.4_dev [130] under default settings. Alignments to the mm10 genome were performed using bowtie2 v2.3.4.1 [131] under default settings and peak calling for each sample was performed with the macs2 v2.2.7.1 [132] callpeak function (--g mm --nomodel --extsize 204 --SPMR) after first running the predictd script and establishing --extsize

204 according to the macs2 manual. Alignments were passed to the DiffBind R package to identify significantly differentially enriched regions under default parameters.

2. Another major caveat in this manuscript is all the examined promoters, except the FMR1 promoter, do not contain a CpG island (high density of CpGs). This concern was pointed out by other reviewers as well. For the only CpG rich FMR1 gene, the authors were not able to measure its DNA methylation level due to their technical limitation. In fact, the methylation level of the CGG repeats at the FMR1 locus can be examined with a commercially available kit (<https://asuragen.com/portfolio/genetics/amplidex-mpcr-fmr1/>), and the CpG island at the FMR1 locus can be measured by Pyro-seq (see the cited reference 95 in the manuscript).

The reviewer raises an important point in that demethylation of the CGG repeats were not measured in the previous version of the manuscript. The neighboring CpG island is not particularly relevant for our methodology because we have shown that the demethylation by dCas9 is limited to its binding site (here, the CGG repeat region only) and therefore any changes in methylation status of CpG island would not be a direct consequence of this method. Though the Asuragen kit methodology is proprietary, according to publications provided by the company it does not in fact measure the CGG repeats themselves – it relies on methylation sensitive restriction enzyme sites (HpaII – recognizes CCGG) that sit just outside of the CGG repeat. Note that CCGG is not a recurring motif in the expanded CGG repeat region. There are instances in the literature where the ability of this kit to measure CGG repeat methylation has been misrepresented. Therefore, since both approaches do not directly measure the methylation of the region that is being modified, we were initially hesitant to use any of these techniques.

However, we have now in this revised version of this manuscript developed a modified version of the Asuragen kit protocol in which – prior to PCR using the kit protocol – we instead digest the DNA with Fnu4HI, which recognizes GCNGC (a repeating motif in the CGG repeat region) and is therefore more relevant to the study. The experimental workflow is outlined in Supplementary Figure S10B and data are presented in Supplementary Figures S10C-D. We demonstrate the functionality of this approach in the easier to amplify and unmethylated HEK293 CGG repeat region. Applied to Fragile X patient fibroblasts expressing dCas9 and either gRNA-CGG or gRNA_{scr}, the method shows a decrease in amplification of methylated CGG repeat region with gRNA-CGG, indicative of demethylation. We therefore provide evidence that this CG-dense region has reduced methylation with the dCas9 demethylation approach.

To reflect this, we have added the following sentence to the Results section on page 18 lines 694-695: “...we observed a reduction in the methylated CGG repeat fraction in the gRNA-CGG condition (Supp. Fig. S10B-D)...”

In addition, the examination of FMR1 expression level in Figure 7J lacked a positive control. It has been reported and confirmed by multiple research groups that Fragile X syndrome patient cells do not express FMR1 at all. Normalization of FMR1 expression in dCas9 cells to the untreated cells with literally zero expression of FMR1 is misleading. A positive control, such as fibroblasts with normal CGG repeats, should be included to calculate the percentage of FMR1 expression in dCas9 cells. In summary, failure to examine the DNA methylation level at the targeted FMR1 locus and lack of a positive control in FMR1 expression analysis do not support their conclusion “targeting demethylation to the pathologically silenced FMR1 gene targets robust induction of gene expression”. This caveat prevents the generalization of their key point stated in the title “Unraveling the functional role of DNA demethylation at specific promoters by targeted steric blockage of DNA methyltransferase with CRISPR/dCas9”, and significantly weakened the significance of this study.

The reviewer has suggested here that a positive control such as fibroblasts with normal CGG repeats should be included to calculate the percentage of FMR1 expression in treated cells.

We have now performed this analysis: we obtained age-matched control fibroblasts from the same supplier (as the Fragile X patient fibroblasts) and expression data for FMR1 induction upon treatment with dCas9:gRNA-CGG and Cre recombinase in Figure 7J is now presented as a fraction of expression in these control fibroblasts, referred to here as wild-type expression.

We thank the reviewer for this key recommendation and have now modified the results section from the previous version, reproduced below, to the current version, reproduced below that. We would also like to note that the restoration of wild-type levels of expression (which we had not claimed) is a separate concept from the robust induction (that we claim) where, in the latter case, we are comparing the large magnitudes of induction of FMR1 gene expression within the Fragile X patient fibroblast model to the magnitudes of induction we reported in the genes we had previously examined (e.g. *Il33*). We have tried to make this clearer in the text.

Previous manuscript version:

“After application of our optimized dCas9-demethylation protocol using gRNA-CGG or gRNAsc (20 µg/mL puromycin) we measured expression levels of the FMR1 gene in 6 independent pools and found significant upregulation of FMR1 gene expression ($p=0.0087$) up to a maximum of 110-fold in one pool, as well as other pools with 9-, 10-, 21-, and 70-fold inductions (Fig. 7J), all of which are vastly higher than the induction following TSS demethylation observed in *Il33* and suggestive of the fact that in this case DNA methylation of the repeat region has a large effect on gene expression. We were unable to map DNA methylation in the dCas9 demethylated pools due to technical challenges in amplification of such a highly repetitive and CG-dense region, despite attempting to do so with buffers and polymerases optimized for GC-rich templates, as well as bisulfite-free approaches in an effort to avoid the decreased sequence diversity caused by sodium bisulfite (C to T transitions).”

New manuscript version on page 18 lines 692-706 (additions highlighted, deletions highlighted and struck-through):

“After application of our optimized dCas9-demethylation protocol using gRNA-CGG or gRNAsc (20 µg/mL puromycin) we observed a reduction in the methylated CGG repeat fraction in the gRNA-CGG condition (Supp. Fig. S10B-D) and ~~measured expression levels of the FMR1 gene in 6 independent~~

pools and found significant upregulation of FMR1 gene expression ($p=0.0087$) up to a maximum of 110-fold in one pool, as well as other pools with 9-, 10-, 21-, and 70-fold inductions (Fig. 7J), characterized by an increase from a mean 0.7% of wild-type expression in gRNAsc cells to a mean of 27% in gRNA-CGG cells and as much as a 110-fold induction in one cell line corresponding to 81% of wild-type FMR1 levels. The magnitudes of induction of FMR1 gene expression all of which are vastly higher larger than the induction following TSS demethylation observed in II33 and are suggestive of the fact that in this case DNA methylation of the repeat region has a large effect on gene expression. We were unable to map DNA methylation in the dCas9 demethylated pools due to technical challenges in amplification of such a highly repetitive and CG-dense region, despite attempting to do so with buffers and polymerases optimized for GC-rich templates, as well as bisulfite-free approaches in an effort to avoid the decreased sequence diversity caused by sodium bisulfite (C to T transitions).

3. The nomenclature in this manuscript is chaotic. For instance, the “deltaCas9(dCas9):gRNA” needs to be consistent with the widely used format “dead Cas9:sgRNA” in the field. Another example is DMR. The authors confused about differentially methylated regions (DMRs) vs. differentially methylated CpGs (DMGs). These are two different concepts. When they used DMRs, what they referred to are DMGs.

Thank you for bringing this to our attention – we have removed all instances of “delta Cas9” and replaced them with “nuclease-dead Cas9” or “dCas9” where appropriate. Regarding the use of the term “gRNA” as opposed to “sgRNA”, it is our experience that both are commonly used terms in the field and as gRNA is a more encompassing term (includes crRNA and tracrRNA, which we did employ at one point in the manuscript) we would prefer to retain this nomenclature if that is fine with the reviewer.

It is also our experience that differentially methylated genes are referred to as DMGs, not differentially methylated CpGs, so we are hesitant to use the term suggested by the reviewer. However, we agree with the reviewer that DMR is not an accurate term, and we have now replaced all instances of DMRs in the manuscript and figures with dmCpGs, a term found widely throughout the literature.

Reviewer #3 (Remarks to the Author):

The authors have addressed all my concerns in detail, and have performed additional experiments where necessary.

REVIEWERS' COMMENTS

Reviewer #2 (Remarks to the Author):

In this revised manuscript by Sapozhnikov and Szyf, they performed additional experiments, described the new results, and interpreted their data in an appropriate manner compared to the previous manuscript. My scientific concerns were addressed. A minor suggestion to consider. The 150 differentially enriched regions identified by CHIP showed no significant differences on DNA methylation by WGBS. The authors could consider to generate one additional panel based on this result in Figure 6 to highlight this important aspect of DNA methylation editing.

Reviewer #2 (Remarks to the Author):

In this revised manuscript by Sapozhnikov and Szyf, they performed additional experiments, described the new results, and interpreted their data in an appropriate manner compared to the previous manuscript. My scientific concerns were addressed. A minor suggestion to consider. The 150 differentially enriched regions identified by ChIP showed no significant differences on DNA methylation by WGBS. The authors could consider to generate one additional panel based on this result in Figure 6 to highlight this important aspect of DNA methylation editing.

Thank you for this suggestion. We agree that it is important to emphasize this point by adding a panel. We have now added a volcano plot (Figure 6J) demonstrating methylation changes and associated statistical values in differentially enriched regions.

In addition to the Figure Description, we have also added the following descriptive text, in regard to this new panel, to the main text of the manuscript (lines 571-575):

“Moreover, restricting differential methylation analysis to only the 549 CpGs (with a minimum coverage of 5X in all 6 samples) located within the 150 off-target DERs bound by dCas9:gRNA3 revealed no statistically significant differentially methylated sites and no otherwise apparent trend that favors nonsignificant hypomethylation over hypermethylation (Fig. 6J).”